# DataMan: Data Manager for Pre-training Large Language Models

**Ru Peng**[1][*]  **Kexin Yang**[2]  **Yawen Zeng**[1]  **Junyang Lin**[2]  **Dayiheng Liu**[2][†]  **Junbo Zhao**[1][†]

[1]Zhejiang University    [2]Alibaba Group

{rupeng,j.zhao}@zju.edu.cn    liudayiheng.ldyh@alibaba-inc.com

## Abstract

The performance emergence of large language models (LLMs) driven by data scaling laws makes the selection of pre-training data increasingly important. However, existing methods rely on limited heuristics and human intuition, lacking comprehensive and clear guidelines. To address this, we are inspired by *"reverse thinking"* – prompting LLMs to self-identify which criteria benefit its performance. As its pre-training capabilities are related to perplexity (PPL), we derive 14 quality criteria from the causes of text perplexity anomalies and introduce 15 common application domains to support domain mixing. In this paper, we train a **Data Man**ager (**DataMan**) to learn quality ratings and domain recognition from pointwise rating, and use it to annotate a 447B token pre-training corpus with 14 quality ratings and domain type. Our experiments validate our approach, using DataMan to select 30B tokens to train a 1.3B-parameter language model, demonstrating significant improvements in in-context learning (ICL), perplexity, and instruction-following ability over the state-of-the-art baseline. The best-performing model, based on the *Overall Score l=5* surpasses a model trained with 50% more data using uniform sampling. We continue pre-training with high-rated, domain-specific data annotated by DataMan to enhance domain-specific ICL performance and thus verify DataMan's domain mixing ability. Our findings emphasize the importance of quality ranking, the complementary nature of quality criteria, and their low correlation with perplexity, analyzing misalignment between PPL and ICL performance. We also thoroughly analyzed our pre-training dataset, examining its composition, the distribution of quality ratings, and the original document sources.

## 1 Introduction

As large language models (LLMs) demonstrate performance emergence driven by data scaling laws, the significance of data has become increasingly evident (Kaplan et al., 2020; Brown et al., 2020; Chowdhery et al., 2023). This trend prompts researchers to explore how to select pre-training data, including deduplication (Lee et al., 2022), domain mixing (Gao et al., 2020; Shen et al., 2023), heuristic-based data selection (Rae et al., 2022; Wenzek et al., 2019), and data sampling using LLM quality signals (Gunasekar et al., 2023; Wettig et al., 2024a). Although these efforts aim to enhance data quality and diversity, deduplication and domain mixing are solely used as a priori or post-hoc steps in the data selection process. Furthermore, existing data selection methods typically rely on limited heuristics and human intuition, lacking comprehensive and clear criteria for data selection, thus making the selection of ideal pre-training data for LLMs an unresolved challenge.

Following this line, this paper provides guidelines for selecting pre-training data, including quality criteria, application domains, and a **Data Man**ager (**DataMan**) with a comprehensive quality rating and domain recognition, equipped by data sampling strategies to enhance the LLM performances. Firstly, we believe that excellent quality criteria must: 1)-apply to a wide variety of texts; 2)-demonstrate a deep understanding of content, capturing semantic levels; and 3)-complement each other. However, existing studies (Rae et al., 2022; Wettig et al., 2024a) rely on limited heuristics and human intuition, while grounded in empirical findings, lack generality and comprehensiveness. To address this issue,

---

[*]This work done during internship at Qwen team, Alibaba Group.
[†]Both are corresponding authors.

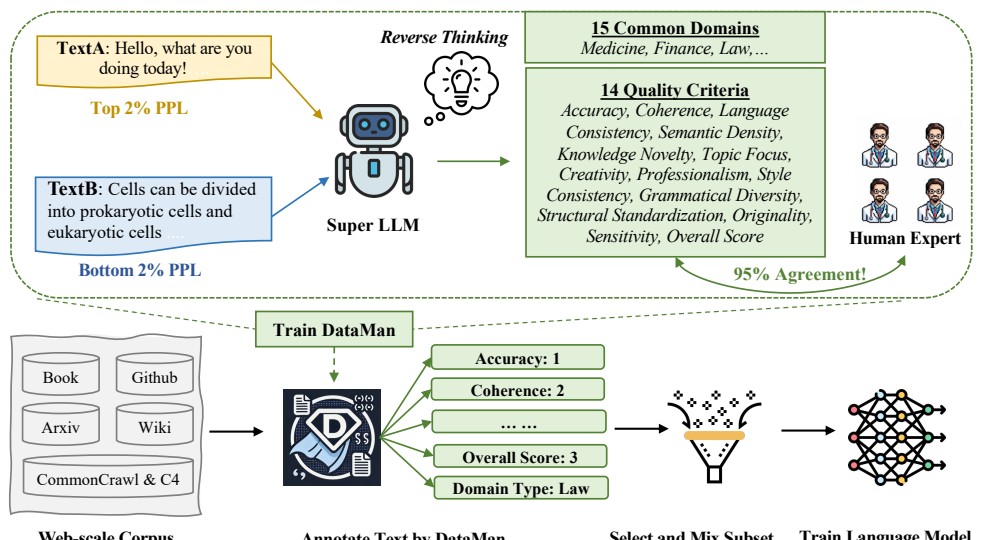

Figure 1: The pipeline of the Sample-with-DataMan model: We derived 14 quality criteria from LLMs' *reverse thinking* and used DataMan to annotate the quality rating and domain type of pre-training data. By employing data sampling strategies to select subsets, the performance of trained LLMs outperforms the state-of-the-art data sampling baseline.

we are inspired by "*reverse thinking*" — **prompting LLM to self-identify which criteria benefit its performance**, as its pre-training capability is related to perplexity (PPL) (Muennighoff et al., 2024; Marion et al., 2023b). We extracted documents with the top 2% and bottom 2% PPL from different sources and used Super LLM to **identify the reasons for anomalous in document perplexity**. Through iterative refinement, we derived 13 quality criteria related to LLM performance: *Accuracy, Coherence, Creativity, Grammatical Diversity, Knowledge Novelty, Language Consistency, Originality, Professionalism, Semantic Density, Sensitivity, Structural Standardization, Style Consistency, and Topic Focus*, and formed comprehensive criterion called *Overall Score*. Further, we introduced 15 common domain types in the LLM application industry (Naveed et al., 2023) for domain mixing and developed a full prompt to curate a fine-tuning dataset. We employ a small-scale LLM to fine-tune DataMan via text generation and apply pointwise rating over pairwise rating (Liu et al., 2007) for more applicable, cost-effective inference on vast datasets. Using DataMan, we annotated 447B tokens in the Slimpajama corpus (Soboleva et al., 2023) with quality ratings and domain types to create the DataPajama dataset. While ensuring the diversity of sources and domains, we maximized the representativeness of quality criteria, sampling a 30B token subset from the DataPajama and trained the Sheared-LLaMA-1.3B language model (Xia et al., 2023) from scratch.

In our experiment, we initially validated the feasibility of using prompt as it achieved over 95% agreement with human preferences. We then revisited the limitation of pairwise rating (Wettig et al., 2024a), bounded pointwise rating error by its model loss, and fairly ranked the annotation results from both ratings to explain, from three perspectives, why we chose pointwise rating. While training the DataMan model, we thoroughly analyzed its fine-tuning datasets, hyperparameter search, inference accuracy and efficiency of three model versions, and misclassification issues. Further, We evaluated language models trained on 30B tokens sampled with different data selection methods from the DataPajama dataset. In ten downstream tasks, the Sample-with-DataMan model, trained on data sampled with Dataman's 13 quality criteria, outperformed the existing state-of-the-art (SOTA) baseline (Educational value $\tau = 2$) in in-context learning performance by 0.4% to 4.3%, showcasing the effectiveness of these criteria. As the *Overall Score* rose from 1 to 5, both ICL performance and PPL improved significantly, confirming the necessity of quality ranking. Our strongest model was achieved at *Overall Score l=5*, even surpassing the model trained on uniform sampling with 50% more data, as the *Overall Score* encompasses attributes of all criteria, supported by its high correlation with other quality criteria. To solidify our results, we sampled a larger 60B data subset and compared the strongest Sample-with-DataMan model *Overall Score l=5* against the existing SOTA baseline. Meanwhile, we found PPL and ICL performance are not aligned strictly. Through correlation analysis and visualization, we reveal that PPL reflects general understanding, whereas

ICL captures downstream generalization ability. In the instruction following tasks, all Sample-with-DataMan models overwhelmingly surpassed the existing SOTA baseline with a win rate between 67.1% and 78.5%. Additionally, We continued pre-training the strongest *Overall Score l=5* model using high-rated, domain-specific data annotated by DataMan, achieving superior ICL performance in specific domains, thereby verifying DataMan's capability for domain mixing. We also conducted an in-depth analysis of the DataPajama dataset, explored the distribution of DataMan's quality ratings from various sources, and inspected the original documents corresponding to each quality rating=1,2,3,4,5. we also identified complementary relationships among quality criteria, while their low correlation with perplexity further confirms the novelty of DataMan's quality criteria.

Our contributions are summarized as follows:

1. Based on the relationship between LLM performance and PPL, "*reverse thinking*" allows LLMs to self-identify which criteria benefit its performance, assemble with typical application domains to guide data selection.

2. We introduce DataMan, offering comprehensive quality ratings and domain recognition, along with data sampling strategies to optimize LLM pre-training. Vast experiments validate its efficacy, setting new performance records.

3. We will release the code, all models, and the annotated DataPajama dataset, paving the way for the community to explore the guidelines between data and LLMs further.

## 2 RELATED WORK

**Deduplication.** Deduplicating training data is now standard in managing pre-training data for LLMs, as it greatly impacts performance (Lee et al., 2022). While Kaplan et al. (2020) and Hoffmann et al. (2022b) examine scaling laws with unique data trained for one epoch, some studies (Hernandez et al., 2022; Muennighoff et al., 2024; Xue et al., 2024) suggest that repeated data can harm performance, particularly as repetition epochs and model size grow. Additionally, fuzzy or semantic deduplication improves LLM performance (Jiang et al., 2022; Abbas et al., 2023; Tirumala et al., 2023). But deduplication should precede quality-based selection and domain mixing, and it is unsuitable for sampling fixed-size subsets.

**Heuristic-based Selection.** This selection approach includes two methods: *Rule-based heuristics*, which apply manually crafted rules—such as mean word length, stop word fraction, and word repetition thresholds (Laurençon et al., 2022; TogetherAI, 2023; Penedo et al., 2023; Soldaini et al., 2024)—to filter data, exemplified by the C4 filter (Raffel et al., 2020) and Gopher rules (Rae et al., 2022). These reduce noise effectively but need complex manual design. In contrast, *Model-based heuristics* use models like binary grammar discriminators (Chowdhery et al., 2023; Touvron et al., 2023a) to select data resembling the target domain, alongside techniques like importance resampling (Xie et al., 2023) and perplexity filtering (Wenzek et al., 2019; Muennighoff et al., 2024; Marion et al., 2023a). However, they require more precise quality ratings for optimal selection.

**Domain mixture.** Most pre-training datasets, like the Pile (Gao et al., 2020), comprise mixed data from various sources and domains (Nijkamp et al., 2023; Zhang et al., 2023; Yang et al., 2024b; Maini et al., 2024; Li et al., 2024). As LLMs gain traction, domain-specific data for improved functionalities are increasingly used in model training (Du et al., 2022; Gao et al., 2023). Identifying the optimal domain mixture ratio is essential for effective LLM pre-training (Wang et al., 2023). Early attempts to define this relied on experiments and intuition (Gao et al., 2020; Thoppilan et al., 2022). Recent studies have begun to use automatic methods, such as domain generalization (Xie et al., 2023; 2024), domain gradients (Fan et al., 2023), and loss evaluation (Xia et al., 2023), to assign domain weights and assess model performance across various mixtures. This inspires our work to introduce domain types to assist in pre-training data mixture.

**LLM quality signals.** Appropriate LLM quality signals are valuable for selecting pre-training data (Gunasekar et al., 2023). Research shows that data enriched with facts and trivia aids LLMs in accurately addressing niche topics and fictional worlds (Petroni et al., 2019; Korbak et al., 2023). Other studies have emphasized the educational value of data as a key for enhancing LLMs' reasoning capabilities (Wei et al., 2022; Kojima et al., 2022). Recent efforts have integrated these insights,

proposing four quality criteria—writing style, facts and trivia, educational value, and required expertise—to assess pre-training data and enhance LLM capabilities (Wettig et al., 2024b). However, these criteria depend heavily on human intuition and thus lack generality and comprehensiveness.

## 3   ANNOTATING TEXT BY DATAMAN

We developed **DataMan** (see Figure 1), a data manager that comprehensively annotates 14 quality criteria and domain types, enabling ideal data selection and mixing.

### 3.1   OVERVIEW OF THE ANNOTATION

Let $\mathbf{t} = \{t_1, \ldots, t_n\}$ represent all documents to be annotated. The annotation results of DataMan correspond to querying the ratings of document $t_n$ across all quality criteria and its domain type. Assume that the quality ratings and domain recognition are in a multi-level annotation format as $\mathcal{L} = \{(l_1^1, \ldots, l_1^C), \ldots, (l_n^1, \ldots, l_n^C)\}$, where $l_n^C \in \{1, \ldots, K\}$ is the label for the $C$-th criterion or domain of document $t_n$. For this paper, $K$ is 5 for quality ratings and 15 for domain recognition. Let $F$ be a class of functions, and $f \in F$ be the annotation function. We use a super LLM as the annotation function, recording the annotation results for each criterion and domain, expressed as: $f(t, \mathcal{L}) = \{(t_1, l_1^1, \ldots, l_1^C), \ldots, (t_n, l_n^1, \ldots, l_n^C)\}$. Thus, we can quickly create a fine-tuning dataset for DataMan, $\mathcal{S} = \{(t_i, l_i^1, \ldots, l_i^C)\}$. Essentially, our method uses pointwise rating by pointwise learning to rank (L2R) model (Liu et al., 2007; 2009). We minimize the loss function based on the document, annotation labels, and function:

$$L(f; t, \mathcal{L}) = \sum_{i=1}^{n} \left( f(t_i) - l_i \right)^2 . \tag{1}$$

This trains DataMan to learn the annotation functions for each quality criterion and domain type.

### 3.2   PROMPT CURATION

How to define the quality criteria and domain types of texts? We believe that excellent quality criteria should: 1)-apply to a wide variety of texts, 2)-demonstrate a deep understanding of content, capturing semantic levels, and 3)- complement each other. However, prior studies largely relied on human intuition, such as educational value in Gunasekar et al. (2023), writing style in Wettig et al. (2024a), and toxicity and privacy in Korbak et al. (2023), which lack generality and comprehensiveness. To address this issue, we undertook an in-depth exploration of text quality, motivated by "*reverse thinking*"—*prompting the LLM to self-identify which criteria benefit its performance*. Specifically, since LLMs' pre-training abilities are closely related to their PPL (Muennighoff et al., 2024; Marion et al., 2023b), where "high PPL indicates data is difficult to learn, and vice versa", we focused on the training text from various sources with the top 2% and bottom 2% of PPL. We devised an analytical prompt for a Super LLM[1] to investigate the reason behind these perplexity anomalies in documents, aiming to analyze the traits of both easy-to-learn and difficult-to-learn data. Through iterative refinement, we derived 14 quality criteria as shown in Figure 1. Additionally, we identified the 15 domain types that need to be assessed from typical LLM application industries (Naveed et al., 2023), integrating the *"let's think step by step"* chain-of-thought prompting strategy (Wei et al., 2022) and a thoughtfully designed system prompt. Further details of the entire process are in Appendix A.

**Prompt validation.** We validate prompt effectiveness using clear cases before prompt use. Initially, we gathered a pool of documents preliminarily rated by an independent group, splitting them into two sets of ten documents each—high-rated and low-rated—to ensure a distinct quality gap. Table 4 of Appendix A details these document sources. These 20 documents were randomly shuffled and then rated on a scale of 1-5 based on each quality criterion by five independent human annotators who had not seen them before. Subsequently, using the super LLM with our prompt, we evaluated the same documents and compared super LLM ratings with human majority votes, finding over 95% agreement. Also, we ensured inter-rater reliability among human annotators by calculating the Kappa coefficient (McHugh, 2012), which validated the consistency of human ratings. However, this human validation remains subjective, we hope the community develops a more rigorous method.

---

[1]The Super LLM can independently refer to any advanced LLM. In this paper, We use the state-of-the-art LLM available at that time, `GPT-4-0125-preview`.

### 3.3 WHY USE POINTWISE RATING?

Using LLMs to rate text falls into two categories: pointwise rating and pairwise rating. Here, we explain why we chose the pointwise rating from the following three aspects:

**Pairwise rating limitations.** We revisited pairwise rating (Wettig et al., 2024a) and observed minimal quality differences among the top-3 documents ranked by *writing style*, as even humans struggle to discern differences (see Table 5). This highlights the limitations of pairwise ratings when quality differences are marginal, as documents meeting an "acceptable" quality threshold should be accepted, where pointwise rating aligns well with human judgment. Appendix B.1 notes that creating a fine-tuning dataset requires Super LLM to make 40 predictions per criterion and document pairs, then use LLM preferences as labels. Table 5 shows that document pairs need a $\geq 50\%$ selection probability difference for quality criteria, also required for inference in Table 6, thus questioning the practicality of pointwise ratings. For $N$ documents, pointwise ratings only need $N$ ratings, whereas pairwise ratings demands $N*(N-1)$ comparisons with 40 predictions each, greatly increasing costs. Despite both ratings capture different aspects, pointwise rating's broader applicability and cost-effectiveness in fine-tuning dataset curation or pre-training data annotation make it our preferred choice.

**Bound pointwise rating error.** We present the mathematical connection between rating errors and loss in the pointwise rating model (Chen et al., 2009).

$$1 - NDCG(f; t, \mathcal{L}) \leq \frac{15\sqrt{2}}{N_n} \left( \left( \sum_{i=1}^{n} D(t_i)^2 \right) - n \prod_{i=1}^{n} D(t_i)^{2/n} \right)^{1/2} (L(f; t, \mathcal{L}))^{1/2}, \quad (2)$$

$$NDCG(f; t, \mathcal{L}) = \frac{1}{N_n} \sum_{i=1}^{n} G(l(\pi_f(t_i)))D(t_i), \quad (3)$$

$$N_n = \max_{\pi} \sum_{i=1}^{n} G(l(\pi(t_i)))D(t_i), \quad (4)$$

where NDCG is rating measures defined with respect to $K$-level ratings $\mathcal{L}$, $G$ is the gain function, $\pi_f$ is the rating list produced by the rating function $f$, and $D$ is the position discount function. One usually sets $G(z) = 2^z - 1, D(z) = \frac{1}{\log_2(1+z)}$ if $z \leq M$, and $D(z) = 0$ if $z > M$ ($M$ is a fixed integer). Thus, as the pointwise rating model minimizes loss via training, the rating error also reduces, validating the feasibility of pointwise ratings theoretically.

**Fairly compare both rating results.** To ensure a fair comparison despite differing evaluation metrics, we ranked annotations from Qurating's pairwise and DataMan's pointwise ratings on the same 58,824 documents from the first split of Qurating's 1B analysis set[2]. We then presented the top, median, and bottom ten documents based on Qurating's four criteria—writing style, facts and trivia, educational value, and required expertise—and DataMan's Overall Score with sum of the remaining criteria. All results are provided here[3]. Our conclusions are: *i)*-Higher Qurating criteria correlate with higher DataMan's Overall Scores for the top and bottom 10 documents, and vice versa. *ii)*-Median 10 documents show minimal changes in Qurating's criteria, yet their Overall Scores vary from 3 to 5, supporting that "Pairwise rating struggles with minimal quality differences, while pointwise rating matches human judgment." *iii)*-Rankings by Overall Score highlight top-quality documents, especially in STEM fields. *iv)*-The "*application_domain*" field accurately categorizes domains, aiding in domain-specific continue pre-training, as shown in Table 2.

### 3.4 TRAINING THE DATAMAN MODEL

After developing the prompt and rating method, we collected documents from both in-source and out-of-source of the SlimPajama corpus (Soboleva et al., 2023) to train DataMan. By prompting the Super LLM, we obtained 14 quality ratings and domain types for each document, creating a fine-tuning dataset of 35,700 documents at a cost of $13,858. Documents were limited to 2,048 tokens (averaging 810 tokens), surpassing the [256, 512] token range of Wettig et al. (2024a) when handling sentences with broader length variations. Subsequently, we fine-tuned the DataMan model with

---

[2]https://huggingface.co/datasets/princeton-nlp/QuRatedPajama-1B_tokens_for_analysis
[3]https://github.com/pengr/DataMan/blob/main/README.md

Qwen2-1.5B (Yang et al., 2024a) using text generation loss. In Table 8, the DataMan model achieves near 80% average test accuracy across all criteria, with an *Overall Score* test accuracy of 81.3% and a *domain recognition* test accuracy of 86%. Appendix B also provides details on the fine-tuning dataset statistics, average scores for each quality criterion per domain, correlation among quality criteria, three DataMan model versions, the chat prompt for text generation, the hyperparameter search, reports on inference accuracy, misclassification analysis and inference efficiency.

## 4 MANAGING DATA BY DATAMAN

In this section, we apply the DataMan model to manage data by selecting a high-quality, diverse document subset from the pre-training corpus. Each document $d_i$ in the corpus $D$, with source $s_i$, is annotated by the DataMan model with 14 quality ratings and domain types as $\mathcal{L} = \{(l_i^1, \ldots, l_i^{C-1}, q)\}$, where $q$ is the domain type. Given the source and domain distribution probabilities $P(s)$ and $P(q)$ respectively, we perform top-k (k is the selected subset size) sampling without replacement for each quality criterion across source and domain distribution, using the probability:

$$P(d_i) = \frac{P(d_i \mid l^j, s, q) \cdot P(s, q)}{\sum_{d_j \in \text{top-k}(l^j)} P(d_j \mid l^j, s, q) \cdot P(s, q)}, \text{and } P(d_i|l^j) = \frac{l_i^j}{\sum_{d_j \in D} l_i^j}. \quad (5)$$

This method maximizes sample representativeness based on quality rating while ensuring diversity in source and domain distributions, and sampling without replacement prevents duplicate data. To assess if the *Overall Score* covers all criteria, we substitute top-k with uniform sampling based on fixed *Overall Score* ratings. These techniques implicitly steer language modeling toward reward-weighted regression (Peters & Schaal, 2007; Korbak et al., 2023), expanding maximum likelihood estimation loss with a data reward mechanism.

## 5 EXPERIMENTS

We empirically validate the DataMan method by training the language model from scratch.

### 5.1 EXPERIMENTAL SETUP

**DataPajama.** We used the Llama tokenizer (Touvron et al., 2023a) to segment Slimpajama corpus (Soboleva et al., 2023), a cleaned and deduplicated version of the RedPajama (TogetherAI, 2023), into 1024-tokens documents. The DataMan model then annotated these documents with 14 quality ratings and domain types, creating the 447B token DataPajama for pre-training. Although the annotation process is expensive, it can be reduced via large-scale parallelization and cost-effective DataMan models. The quality ratings and domain types in DataPajama can serve various purposes, such as data selection, data mixing, or continued pre-training in specific domains. Detailed statistics of DataPajama can be found in Table 11 of Appendix C.

**Data selection methods.** For each baseline, we select a 30B token training dataset from DataPajama with one of the following methods, while retaining the original source proportion as the overall dataset. We leave it as future work to explore combinations of quality criteria to broaden our method.

- *Uniform*: We select randomly with a uniform probability across documents. For comparison's sake, we train an additional model on 45B tokens, requiring 50% more compute.

- *DSIR*: We apply data selection with importance resampling (DSIR) (Xie et al., 2023) and select examples that resemble either English Wikipedia or the Book domain (TogetherAI, 2023)— commonly used as proxies for quality (Brown, 2020; Touvron et al., 2023a; Xie et al., 2023). We follow Xie et al. (2023) and train hashed bigram models on DataPajama and the target data.

- *Perplexity Filtering*: We implement perplexity filtering (Wenzek et al., 2019; Marion et al., 2023a) and select the documents with the lowest/highest perplexity scores, as computed by a pre-trained Sheared-Llama-2.7B model (Xia et al., 2023)—2× the size of our DataMan model.

- *Sample with Qurating*: We sample 30B tokens according to each of the four criteria described in Wettig et al. (2024a): *writing style, facts and trivia, educational value, and required expertise*. Specifically, we normalize the variance of the quality ratings to be 1 and then sample with temperature $\tau \in \{0.0 \text{ (i.e., top-}k \text{ selection)}, 2.0\}$. Additionally, we merge the Qurating-sampled

data for $\tau = 2.0$ of the four criteria as *criteria mix*, and subsampling is as randomly to 30B tokens, ensuring that we exclude duplicate documents.

- *Sample with DataMan (Ours)*: For each of 13 quality criteria, we perform top-k sampling based on the quality ratings and domain type described in Section 4. Additionally, we explore sampling 30B tokens with overall scores $l \in \{1, 2, 3, 4, 5\}$ to demonstrate the effectiveness of all criteria.

**Training settings.** Using different data selection methods, we select a 30B token subset from DataPajama and train a randomly initialized Sheared-Llama-1.3B language model (Xia et al., 2023) for one epoch in a randomly shuffled order. We train on slightly more data than the compute-optimal ratio (Hoffmann et al., 2022a) because more training tokens better reflect the performance gains attributed to data quality. Further details are in Appendix C.

**Evaluation metrics.** We provide a holistic evaluation of the language models trained on 30B tokens:

- We measure the perplexity over SlimPajama's validation set and test set, 500M tokens each.
- We evaluate the in-context learning (ICL) performance using `lm-evaluation-harness` (Gao et al., 2024). We study 10 tasks, comprising 5 reading comprehension tasks (ARC-easy/challenge (Clark et al., 2018), SciQA (Welbl et al., 2017), LogiQA (Liu et al., 2020), BoolQ (Clark et al., 2019)), 3 commonsense reasoning tasks (HellaSwag (Zellers et al., 2019), PIQA (Bisk et al., 2019), WinoGrande (Sakaguchi et al., 2019)) and 2 knowledge-intensive tasks (NQ (Kwiatkowski et al., 2019), MMLU (Hendrycks et al., 2021)). We report the detailed settings in Appendix C.
- We evaluate the instruction-following capabilities of our models, borrowing the setting used by Xia et al. (2023). We perform supervised fine-tuning on 10,000 instruction-response pairs from the ShareGPT dataset. We evaluate another 1,000 instructions and use the AlpacaFarm codebase (Dubois et al., 2024) to judge the responses from two models with `GPT-4o`.

## 5.2 RESULTS

We report the model's perplexity and ICL results in Table 1, the instruction-following evaluation in Figure 2, the domain-specific continue pre-training performance in Table 2. Appendix C provides comprehensive results for all models, including validation and test perplexity across sources in Tables 12, 13, and ICL results for individual task in Table 14.

**Traditional methods perform poorly.** Table 1 indicates that DSIR and perplexity filtering perform worse than random uniform sampling. This indicates that model output–based methods, despite their widespread use, are ineffective for data selection.

**Clear quality criteria work, but Qurating's criteria mix fails.** Table 1 shows that selecting data with Qurating's four criteria (Wettig et al., 2024a) improves performance compared to uniform sampling, highlighting the importance of clear LLM quality signals. Educational value $\tau = 2.0$ is the current SOTA baseline. However, the criteria mix of Qurating's four criterion did not perform well, possibly due to a lack of complementarity among the Qurating's criteria.

**DataMan's criteria surpasses SOTA baseline and *Overall Score* works best.** In Table 1, compared to the SOTA baseline (Educational value $\tau = 2.0$), our 13 individual quality criteria improved ICL performance by 0.4% to 4.3%. For example, the sample-with-creativity model achieved an impressive score of 60.6 in commonsense reasoning tasks. As the comprehensive criterion *Overall Score* increased from 1 to 5, performance gains significantly, highlighting the necessity of quality ranking. Our *Overall Score l=5* works best, even exceeding the *Uniform +50% data* baseline, validating the feasibility of combining the 13 quality criteria into a composite criterion via LLM weighting. This approach not only avoids the disruption of manual adjustments but also achieves optimal results. The correlations among all criteria in Figure 5 further confirm this point.

**PPL and ICL are not strictly aligned.** Our results in Table 1 reveal a trend where PPL and ICL metrics correlate to a degree, increasing or decreasing simultaneously, but they are not aligned strictly. This meets with the intuition that PPL implies general understanding, while ICL focuses more on downstream generalization. Further analysis see Figure 6 in Appendix C. Notably, DataMan's high-rated *Overall Score* achieves an optimal trade-off between understanding and generalization.

Table 1: Sample-with-DataMan models improve perplexity and in-context learning (ICL) results, and the *Overall Score l=5* perform best. We report the validation, test perplexity, and ICL performance of 10 downstream tasks. We highlight the best result in each column and improvement over uniform sampling with the 30B token budget. In Appendix C, we report full results of perplexity in Table 12, 13 and ICL in Table 14.

| Selection Method | | Val Perplexity | Test Perplexity | Reading Comprehension (5 tasks) | Commonsense Reasoning (3 tasks) | World Knowledge (2 tasks) | Average (10 tasks) |
|---|---|---|---|---|---|---|---|
| Uniform | | 10.7 | 10.75 | 50.9 | 55 | 14.9 | 44.9 |
| | +50% data | 10.09 ↓0.61 | 10.14 ↓0.61 | 52.9 ↑2.0 | 57.0 ↑2.0 | 15.9 ↑1.0 | 46.8 ↑1.9 |
| DSIR | with Wiki | 13.34 ↑2.64 | 13.37 ↑2.62 | 50.1 ↓0.8 | 49.8 ↓5.2 | 14.7 ↓0.2 | 42.9 ↓2.0 |
| | with Book | 13.60 ↑2.90 | 13.59 ↑2.84 | 47.9 ↓3.0 | 56.6 ↑1.6 | 14.1 ↓0.8 | 43.8 ↓1.1 |
| Perplexity | lowest | 15.98 ↑5.28 | 16.04 ↑5.29 | 48.3 ↓2.6 | 49.6 ↓5.4 | 13.7 ↓1.2 | 41.7 ↓3.2 |
| | highest | 11.32 ↑0.62 | 11.34 ↑0.59 | 49.6 ↓1.3 | 53.5 ↓1.5 | 13.4 ↓1.5 | 43.5 ↓1.4 |
| Writing Style | top-k | 13.01 ↑2.31 | 12.97 ↑2.22 | 49.3 ↓1.6 | 53.3 ↓1.7 | 13.5 ↓1.4 | 43.4 ↓1.5 |
| | $\tau = 2.0$ | 10.60 ↓0.10 | 10.64 ↓0.11 | 51.0 ↑0.1 | 55.8 ↑0.8 | 14.1 ↓0.8 | 45.0 ↑0.1 |
| Facts & Trivia | top-k | 14.38 ↑3.68 | 14.33 ↑3.58 | 54.3 ↑3.4 | 51.7 ↓3.3 | 15.5 ↑0.6 | 45.8 ↑0.9 |
| | $\tau = 2.0$ | 10.68 ↓0.02 | 10.72 ↓0.03 | 52.7 ↑1.8 | 55.6 ↑0.6 | 15.6 ↑0.7 | 46.2 ↑1.3 |
| Educational Value | top-k | 13.54 ↑2.84 | 13.49 ↑2.74 | 54.7 ↑3.8 | 54.9 ↓0.1 | 14.4 ↓0.5 | 46.7 ↑1.8 |
| | $\tau = 2.0$ | 10.67 ↓0.03 | 10.72 ↓0.03 | 53.3 ↑2.4 | 56.3 ↑1.3 | 15.7 ↑0.8 | 46.7 ↑1.8 |
| Required Expertise | top-k | 14.97 ↑4.27 | 14.92 ↑4.17 | 52.8 ↑1.9 | 48.7 ↓6.3 | 14.3 ↓0.6 | 43.9 ↓1.0 |
| | $\tau = 2.0$ | 10.7 | 10.74 ↓0.01 | 52.7 ↑1.8 | 55.5 ↑0.5 | 15.0 ↑0.1 | 46.0 ↑1.1 |
| Criteria mix | $\tau = 2.0$ | 10.63 ↓0.07 | 10.68 ↓0.07 | 52.1 ↑1.2 | 55.5 ↑0.5 | 15.2 ↑0.3 | 45.7 ↑0.8 |
| Accuracy | top-k | 10.82 ↑0.12 | 10.80 ↑0.05 | 53.8 ↑2.9 | 58.2 ↑3.2 | 16.6 ↑1.7 | 47.7 ↑2.8 |
| Coherence | top-k | 10.72 ↑0.02 | 10.71 ↓0.04 | 54.9 ↑4.0 | 58.8 ↑3.8 | 16.1 ↑1.2 | 48.3 ↑3.4 |
| Creativity | top-k | 11.08 ↑0.38 | 11.00 ↑0.25 | 53.0 ↑2.1 | **60.6 ↑5.6** | 15.2 ↑0.3 | 47.7 ↑2.8 |
| Grammatical Diversity | top-k | 10.87 ↑0.17 | 10.86 ↑0.11 | 55.1 ↑4.2 | 58.8 ↑3.8 | 16.5 ↑1.6 | 48.5 ↑3.6 |
| Knowledge Novelty | top-k | 11.01 ↑0.31 | 11.01 ↑0.26 | 54.6 ↑3.7 | 56.9 ↑1.9 | 15.5 ↑0.6 | 47.5 ↑2.6 |
| Language Consistency | top-k | 10.35 ↓0.35 | 10.35 ↓0.40 | 54.1 ↑3.2 | 59.3 ↑4.3 | 16.7 ↑1.8 | 48.2 ↑3.3 |
| Originality | top-k | 10.68 ↓0.02 | 10.67 ↓0.08 | 53.9 ↑3.0 | 58.6 ↑3.6 | 16.4 ↑1.5 | 47.8 ↑2.9 |
| Professionalism | top-k | 11.27 ↑0.57 | 11.26 ↑0.51 | 54.6 ↑3.7 | 54.8 ↓0.2 | 15.9 ↑1.0 | 46.9 ↑2.0 |
| Semantic Density | top-k | 11.10 ↑0.40 | 11.09 ↑0.34 | 54.4 ↑3.5 | 58.1 ↑3.1 | 16.7 ↑1.8 | 48.0 ↑3.1 |
| Sensitivity | top-k | 10.11 ↓0.59 | 10.13 ↓0.62 | 54.7 ↑3.8 | 59.2 ↑4.2 | 16.1 ↑1.2 | 48.3 ↑3.4 |
| Structural Standardization | top-k | 12.11 ↑1.41 | 12.11 ↑1.36 | 53.7 ↑2.8 | 57.0 ↑2.0 | 17.1 ↑2.2 | 47.4 ↑2.5 |
| Style Consistency | top-k | 10.74 ↑0.04 | 10.73 ↓0.02 | 55.1 ↑4.2 | 59.6 ↑4.6 | 16.2 ↑1.3 | 48.7 ↑3.8 |
| Topic Focus | top-k | 10.41 ↓0.29 | 10.41 ↓0.34 | 54.6 ↑3.7 | 58.4 ↑3.4 | 15.6 ↑0.7 | 47.9 ↑3.0 |
| Overall Score | l=1 | 23.83 ↑13.13 | 23.95 ↑13.20 | 43.1 ↓7.8 | 47.5 ↓7.5 | 13.1 ↓1.8 | 38.4 ↓6.5 |
| | l=2 | 12.84 ↑2.14 | 12.91 ↑2.16 | 50.3 ↓0.6 | 50.9 ↓4.1 | 14.7 ↓0.2 | 43.4 ↓1.5 |
| | l=3 | 11.75 ↑1.05 | 11.78 ↑1.03 | 50.7 ↓0.2 | 54.1 ↓0.9 | 15.2 ↑0.3 | 44.6 ↓0.3 |
| | l=4 | **10.21 ↓0.49** | 10.22 ↓0.53 | 53.5 ↑2.6 | 60.1 ↑5.1 | 16.0 ↑1.1 | 47.9 ↑3.0 |
| | l=5 | 10.52 ↓0.18 | **10.50 ↓0.25** | **55.2 ↑4.3** | 60.2 ↑5.2 | **17.4 ↑2.5** | **49.1 ↑4.2** |

**DataMan's instruction-following abilities also well.** As shown in Figure 2, we compare the instruction-following win rates between the Sample-with-DataMan model and the SOTA baseline (Educational value $\tau = 2.0$). The results indicate that, under the same supervised fine-tuning conditions, our model consistently surpasses the SOTA Baseline, with the *Overall Score* reaching an impressive win rate at 78.5%, further validating the superior performance of our method.

**Training domain-specific models.** While the *Overall Score l=5* achieves the best performance, its performance in specific domains can still be improved, as shown in Table 2. To address this, we applied DataMan's domain recognition to filter data with high *Overall Score* in the *medical, law, and financial* domains, and continued pre-training domain-specific models, achieving ICL performance gains in the corresponding MMLU subtasks. This validates DataMan's capability for domain mixing.

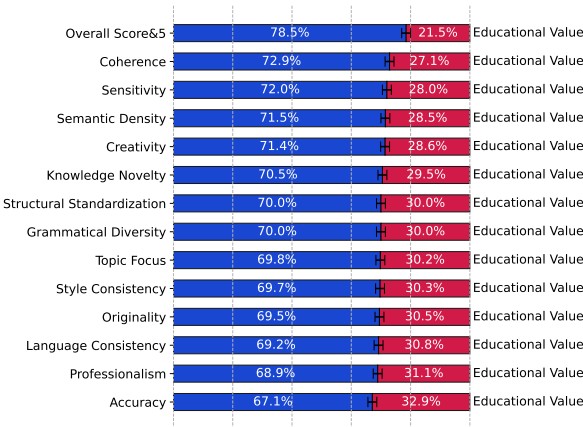

Figure 2: Instruction following win rates of Sample-with-DataMan models v.s. the state-of-the-art baseline (Educational value $\tau = 2.0$) after instruction fine-tuning on 10K ShareGPT examples. The results indicate that, under the same SFT conditions, our model consistently surpasses the SOTA Baseline, with the Overall Score reaching a impressive win rate at 78.5%.

Table 2: The Sample-with-DataMan model further improves perplexity and in-context learning (ICL) results when continuing pre-training on high *Overall Score*, domain-specific data. We report the validation, test perplexity, and ICL performance of corresponding MMLU subtasks.

| | Val Perplexity | Test Perplexity | Anatomy | College Medicine | Medical Genetics |
|---|---|---|---|---|---|
| Overall Score *l=5* | 8.10 | 8.05 | 28.1 | 24.3 | 22.0 |
| + Medicine CPT | 8.11 | 8.11 | 30.4 | 26.6 | 30.0 |
| | **Val Perplexity** | **Test Perplexity** | **International Law** | **Professional Law** | **Jurisprudence** |
| Overall Score *l=5* | 7.92 | 8.13 | 25.5 | 33.9 | 22.2 |
| + Law CPT | 8.06 | 8.34 | 35.5 | 24.7 | 24.1 |
| | **Val Perplexity** | **Test Perplexity** | **Econometrics** | **High School Macroeconomics** | **Marketing** |
| Overall Score *l=5* | 9.51 | 9.49 | 23.7 | 33.3 | 22.2 |
| + Finance CPT | 9.59 | 9.63 | 25.4 | 34.9 | 23.9 |

## 5.3 ANALYSIS OF QUALITY RATINGS

**Distribution of quality ratings.** In Figure 3, we present the distribution of quality ratings across different sources in DataPajama. Overall, the quality ratings for each source are primarily concentrated at 4 and 5, indicating generally high sample quality. This may be related to the fact that DataPajama is a subset of the curated and deduplicated slimpjama corpus. However, for the criteria of *Knowledge Novelty* and *Creativity*, there is a higher proportion of samples scoring 2 and 3, which is consistent with the lower average scores for these two criteria found in Table 7. Across all domains, only a few scientific domains like mathematics and medicine have *Knowledge Novelty* scores above 3, while in *Creativity*, only culture and entertainment scores were high at 3.64 and 3.56, respectively. Nevertheless, in DataPajama, the combined share of domains like mathematics and medicine is only 11.5%, and similarly, the combined share of culture and entertainment is only 25%, both of which are relatively small. This explains the modest ratings of the DataPajama dataset in terms of *Knowledge Novelty* and *Creativity*.

**Correlation between quality ratings and log-likelihood.** In Figure 4, we illustrate the correlation between quality ratings and the log-likelihood scores computed by Llama-2-7b (Touvron et al., 2023b). Most quality criteria do not show a significant correlation with perplexity, except for the criteria of *Structural standardization, Professionalism, and Creativity*, which have Spearman correlation coefficients ranging from 0.47 to 0.55, indicating a weak correlation. This indicates that our 14 quality criteria and sample-with-dataman method are independent of traditional perplexity metrics and filtering, indirectly showcasing the sophistication of the *"reverse thinking"*.

## 5.4 DATA INSPECTION

Furthermore, we examined examples of original documents from each source under Dataman's quality criteria and ratings. Specifically, for each criterion, we randomly selected samples with ratings ranging from 1 to 5 from different sources and presented them in the Appendix D. Notably, these

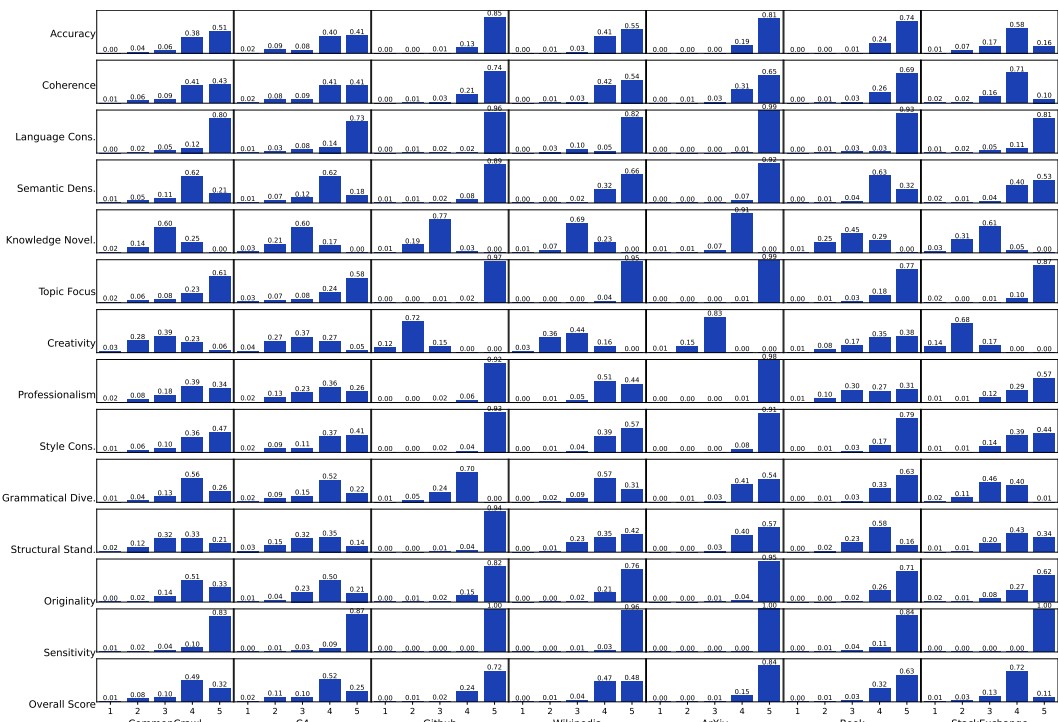

Figure 3: The distribution of quality ratings across different sources in DataPajama

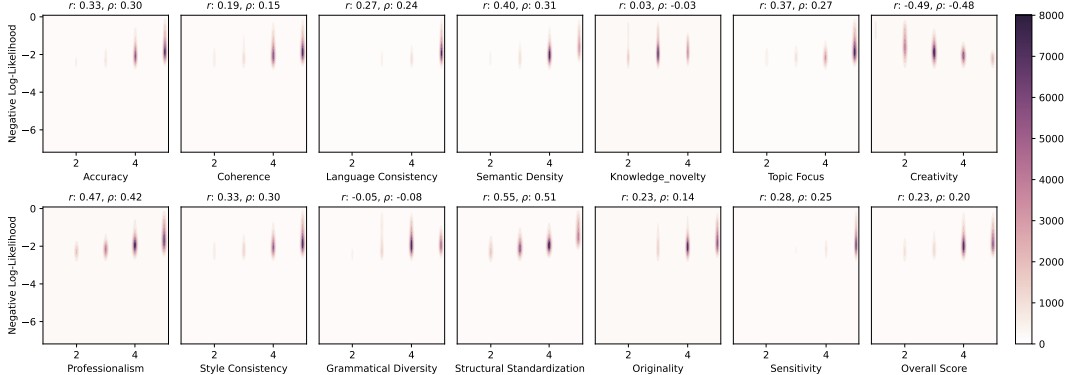

Figure 4: Correlations of quality ratings and negative log-likelihood scores by Llama-2-7B (Touvron et al., 2023b) over 30B tokens training documents. The negative log-likelihoods are averaged over the number of tokens, and are the logarithm of the perplexity score of a single sequence. We observe that perplexity scores are not good approximations for any quality criteria.

samples represent only a small snippet; nonetheless, they exhibit significant quality differences. We invite readers to review these differences in detail, which compares high and low ratings.

## 6  CONCLUSION

In this work, we introduce DataMan, a comprehensive data manager capable of quality rating and domain recognition, developed to facilitate the pre-training data selection and mixing. By using DataMan to annotate pre-training datasets and sampling data based on quality ratings while balancing domain diversity, our trained models demonstrate significant improvements in language modeling, task generalization, and instruction following. Through extensive experiments, we provide valuable insights for the community studying the relationship between data and large language models.

## 7 LIMITATIONS

We acknowledge several limitations in our work. First, DataMan's reliance on LLMs for text quality assessment and domain categorization may inherit biases from these models. Second, DataMan's inference accuracy is not yet optimal, sometimes causing misclassification, highlighting the need for a large-scale collection of documents with diverse quality differences for fine-tuning. Third, using SlimPajama alone as a pre-training corpus limits result reliability, incorporating additional corpora would be better. Fourth, the model size is restricted by data and training resources, resulting in models with only 1.3B parameters, whereas increasing parameters might reveal interesting phenomena. Lastly, the considerable costs of developing data managers, data filtering, and pre-training experiments could hinder further research in this domain. We aim to address these in future work. Despite these limitations, DataMan remains a powerful tool for data selection and mixing.

## ACKNOWLEDGEMENTS

This work is supported by the Fundamental Research Funds for the Central Universities (226-2024-00049), the NSFC Grants (No. 62206247) and the Pioneer R&D Program of Zhejiang (No. 2024C01035).

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

## A PROMPT CURATION

---

**Full Prompt**

Please carefully read and analyze the following text, score it based on fourteen evaluation criteria and their respective scoring definitions. Additionally, select the most appropriate category from the fifteen domain types that best matches the content of the text. Let's think step by step.

**Text**:{text}

**Domain Types:** [A]Medicine [B]Finance [C]Law [D]Education [E]Technology [F]Entertainment [G]Mathematics [H]Coding [I]Government [J]Culture [K]Transportation [L]Retail E-commerce [M]Telecommunication [N]Agriculture [O]Other

**The Higher The Score, The Evaluation Criteria**:
[1]Accuracy: the fewer grammar, referential, and spelling errors the text contains, and the more accurate its expression. _/5
[2]Coherence: the more fluent the content is expressed, and the stronger its logical coherence. _/5
[3]Language Consistency: the more consistent the use of language in the text, with less mixing of languages. _/5
[4]Semantic Density: the greater the proportion of valid information in the text, with less irrelevant or redundant information. _/5
[5]Knowledge Novelty: the more novel and cutting-edge the knowledge provided by the text, with more insightful views on the industry or topic. _/5
[6]Topic Focus: the more the text content focuses on the topic, with less deviation from the main theme. _/5
[7]Creativity: the more creative elements are shown in the text's expression. _/5
[8]Professionalism: the more professional terminology appears in the text, with more accurate use of terms and more professional domain-specific expression. _/5
[9]Style Consistency: the more consistent the style of the text, with proper and appropriate style transitions. _/5
[10]Grammatical Diversity: the more varied and correct the grammatical structures used in the text, showing a richer language expression ability. _/5
[11]Structural Standardization: the clearer the structure followed by the text and the more standardized its format. _/5
[12]Originality: the fewer repetitions and similar content in the text. _/5
[13]Sensitivity: the more appropriately sensitive topics are handled in the text, with less inappropriate content. _/5
[14]Overall Score: the better the comprehensive evaluation of the text, with superior performance in all aspects._/5

---

Our full prompt is shown above, where {text} represents the text to be annotated. Next, we will elaborate on the entire prompt curation process, including obtaining all quality criteria, domain types, and system prompts, which were conducted by experts with the assistance of Super LLM:

**Initializing quality criteria.** Inspired by "*reverse thinking*"—*prompting the LLM to self-identify which criteria benefit its performance*, as its pre-training capabilities are closely related to perplexity (PPL) (Muennighoff et al., 2024; Marion et al., 2023b). To this end, we devised an analytical prompt for Super LLM to investigate the reasons behind textual PPL anomalies (in the top and bottom 2%) from each source, and extract the initial quality criteria below. For clarity, Table 3 provides several examples illustrating how these initial quality criteria are derived.

**Enhancing quality criteria.** Next, we utilize Super LLM to rank the initial quality criteria by importance, eliminating, merging, and supplementing them. The resulting second-step criteria include: *[1] Text accuracy (grammar, references, spelling), [2] Semantic coherence and consistency, [3] Language consistency, [4] Effective semantic content ratio, [5] Knowledge novelty, [6] Topic focus, [7] Creative expression ratio, [8] Proportion of technical terms, [9] Style variability, [10] Complexity of grammatical structures, [11] Content regularity, [12] Content redundancy, [13] Proportion of sensitive topics*. We then revised the prompts under the principle that higher scores indicate better criteria. With Super LLM's assistance, we simplified the criterion names, detailed the criteria for each rating level, and ultimately derived 14 quality criteria in the paper.

---

**Initial Quality Criteria**

[1] Semantic Fluency/Coherence/Logic: Evaluate whether the text is smooth and easy to read, whether the content is coherent, and whether the logic is clear.

[2] Content Consistency/Variability in Language Style: Evaluate if the information within the text is contradictory and if the language style is diverse.

[3] Topic Diversity: Determine the richness and variety of topics addressed in the text.

[4] Content Regularity/Formatting: Consider whether the text adheres to a certain structure or format.

[5] Content Redundancy: Analyze the extent of information repetition within the text.

[6] Proportion of Domain-Specific Vocabulary: Measure the frequency of professional terms or specific vocabulary used in the text (such as proper nouns, technical terms, or Classical Chinese).

[7] Proportion of Sensitive Topics: Examine the percentage of content that addresses sensitive topics (e.g., involving politics, toxicity).

[8] Proportion of Creative Expression: Assess the degree of creative or innovative expression in the text (e.g., use of rhetorical techniques).

[9] Degree of Language Mixing: Analyze the extent to which different languages are used within the text (i.e., the ratio of text in various languages).

[10] Complexity of Text Structure: Evaluate the overall complexity of the text's structure.

[11] Proportion of Long Sentences: Assess the ratio of long sentences within the text.

[12] Proportion of Grammatical, Reference, and Spelling Errors: Evaluate the ratio of grammatical errors (e.g., incorrect punctuation, unclear sentence breaks), reference errors, and spelling mistakes in the text.

[13] Proportion of Content Lacking Semantics: Determine the ratio of parts within the text that lack meaningful content (e.g., garbled text, HTML tags, XML elements, navigation bars, incomplete chart numbers, or disjointed citations).

---

**Identifying domains.** We identified the 15 domain types that require assessment, based on factors such as typical application LLM industries (Naveed et al., 2023), the number of existing industry LLMs, and the level of attention they have received, as indicated by metrics like GitHub stars[4].

**Chain-of-though and system prompts.** We incorporated the chain-of-thought prompting(Wei et al., 2022)—*"let's think step by step"*—while avoiding using few-shot example prompting, as the diverse sources of text could introduce rating biases. We allowed the Super LLM to generate system prompts to enhance the accuracy and confidence of quality ratings, *"You are an expert to evaluate the text quality with high accuracy and confidence. Don't hesitate to use the full range of the score scale, including extreme scores if the text warrants it."* Results indicated that this prompt facilitated more nuanced text annotations compared to the typical prompt, *"You are a helpful assistant."*. To ensure stable output from Super LLM, we set the temperature to 0.

Table 3: We provide several examples of excessively high and low perplexity, use analytical prompts to investigate the reasons for these anomalies, and extract initial quality criteria from the LLM's responses. The analytical prompt is: *"Read the following document, which has a {high/low} perplexity of {ppl_value} for LLM inference. Please analyze the reasons for the PPL anomaly."*

| PPL | Text | LLM's Response |
|---|---|---|
| 1.19 | March: 1.77 inches- April: 5.31 inches- May: 5.9 inches- June: 5.9 inches- July: 5.9 inches- August: 2.36 inches- September: 5.31 inches- October: 5.9 inches- November: 2.36 inches- December: 2.36 inches2009, Senneville QC Precipitation: 56- January: 1.68 inches- February: 2.24 inches- March: 2.24 inches- April: 5.04 inches- May: 5.6 inches- June: 5.6 inches- July: 4.48 inches- August: 2.24 inches- September: 5.6 inches- October: 4.48 inches- November: 2.8 inches- December: 2.24 inches2008, Senneville QC Precipitation ... | Structural consistency: Data is presented in a fixed format. (**[4] Content Regularity/Formatting**) Redundancy: A large number of repeating elements. (**[5]Content Redundancy**) Lack of complex syntax: The text lacks complex sentence structures. (**[10]Complexity of Text Structure**) Minimal context dependence. (**[1]Semantic Fluency/Coherence/Logic**) |

Continued on next page</div>

---

[4]https://github.com/HqWu-HITCS/Awesome-Chinese-LLM

| PPL | Text | LLM's Response |
|---|---|---|
| 1.07 | 4But the wise took oil in their vessels with their lamps. 5While the bridegroom tarried, they all slumbered and slept. 6And at midnight there was a cry made, Behold, the bridegroom cometh; go ye out to meet him. 7Then all those virgins arose, and trimmed their lamps. 8And the foolish said unto the wise, Give us of your oil; for our lamps are gone out. 9But the wise answered, saying, Not so; lest there be not enough for us and you: but go ye rather to them that sell, and buy for yourselves. ... | Textual Standardization: The text is an excerpt from the Bible. **([3]Topic Diversity)** Structural and Redundant: The narrative has structure and redundancy. **([4]Content Regularity/Formatting) ([5]Content Redundancy)** Contextual Consistency: The parts of the text have internal coherence. **([2]Content Consistency/Variability in Language Style)** Lexical and Grammatical Consistency: The Bible use fixed vocabulary and grammatical structures. **([10]Complexity of Text Structure)** |
| 185.1 | An rugent-bas,~d c~li:bic Hmt lakes ca,ring as, ils nwst 011ernrching mol:ival:ional ideal avoids. thfa prolbl!ern, since ii: ent11ils that peisorn, who are conll'nt 1:hrou.ghmit their lifetime to simply receive cm\"<\" from others. without acknowiel!ging anid/ or recipmrnbing thrnt rnre 1in any way are jm:tnfo11bly criticized fo1r exhibitirng; a d!eplornble latck of otlr1er-regaird, and in particufar, a fad. of enga.gem.,~11t with and gratitude tow,ird the s.perifir persons who crnntribule to their Hves itl varim1s ways. Nod!dings's own view Is tbait it takes s1uprisingly little effort to disclluuge our duties to receive or \"complet,~\" 11nother p,~rso111's care: we nrny respond with an approprriale aUih1de such as recognition and/ rnr gratitude (][bid.: 19; 65) ... | Complex Content: The text deeply discusses philosophy and ethics. Terminology Dense: Understanding requires in-depth domain knowledge. **([6]Proportion of Domain-Specific Vocabulary)** Incoherent Text: Multiple parts show incoherence. **([3]Topic Diversity)** Random Characters: There may be encoding errors. **([12]Proportion of Grammatical, Reference, and Spelling Errors)** |
| 167.5 | The Indian Express is now on Telegram. Click here to join our channel (@indianexpress) and stay updated with the latest headlinesFor all the latest Kolkata News, download Indian Express App. MÃ¼nchengladbach Michael Cuisance ist vorzeitig wieder ins Training eingestiegen. Er ist fit und will in der neuen Saison mehr spielen als in der abgelaufenen. Kurzum: Der 19-JÃd'hrige will sich bei Borussia MÃ¼nchengladbach durchsetzen.Michael Cuisance Ãd'rgert sich Ãijber das frÃijhe Aus mit Frankreich bei der U20-WM. Foto: AP/Darko VojinovicCuisance startet bei Borussia seinen Angriff auf die StartelfbGÃijte rwued mi tnaShcte des eh-IFtnrteoslann uaf med wudh,eelgcaNhsÃd'cn da erd eansR eds tatlztiHnagrnispeuaps chontges reenwd lsle. ... | Mixed Languages: A mix of various languages and spelling systems. **([9]Degree of Language Mixing)** Incoherent Sentences: Poor connectivity between sentences. **([2]Content Consistency/Variability in Language Style)** Ambiguous Context: Lack of clear contextual information. **([13]Proportion of Content Lacking Semantics)** Formatting Errors: Contains numerous formatting issues, typos, and garbled text. **([12]Proportion of Grammatical, Reference, and Spelling Errors)** |
| 72.19 | property on sand\\\\in the autumn\\\\rinsing the token and ridding it of dispiritedness a cold\\\\foam bath and icy smiles the lips permitted and the knight\\\\was refreshed then also the mountains had reviving airs\\\\nesting in the hollows from peak to golden peak and there were\\\\lairs for robbers which altered the loneliness\\\\for always he wished for his swan even its shadow even\\\\a shadow on stones that once were heavy and warm\\\\and the swan's story\\\\he cherished along with the memory of his ride that as a basin\\\\is filled then emptied yet its curve remains and its depth\\\\he would never forget nor the exact three quarter of his waltz\\\\there on the fringes of clouds and the embrace in the chasm\\\\with snow and down erasing any doubt the act was modern\\\\as mountain climbing or looking for gentians\\\\any ruin has its surplus or wrinkle there on the mask\\\\fed with dishes of rain\\\\the knight was no exception he even wore his hair like a legend\\\\whistling a lot from a need for quaintness he plied himself\\\\back and forth pretending he had an occupation like speech\\\\of course it was the same\\\\even the military and the religious wherever there was a crowd | Text content is complex and varied: The text you provide looks like it is part of a series of poems or literary works, with lots of metaphors, symbols, and abstract expressions. **([1]Semantic Fluency/Coherence/Logic)** Irregular format and structure: Poetry often has a different format and structure than regular prose, such as line breaks, irregular punctuation, blank lines, and indentation. **([10]Complexity of Text Structure)** Proper nouns and archaisms: The text may contain proper nouns, archaisms, or words coined by the author, which may not be in the model's vocabulary. **([6] Proportion of Domain-Specific Vocabulary)** Creative expression in text: Because poetry often contains a lot of creative expression and experimental use of language, this is beyond the scope of the model's predictions based on conventional language usage. **([8]Proportion of Creative Expression)** |

| PPL | Text | LLM's Response |
|---|---|---|
| 1.91 | a former 1970s antiwar radical fugitive wanted for murder. After living for more than 30 years underground, Grant must now go on the run. With the FBI in hot pursuit, he sets off on a cross-country journey to track down the one person that can clear his name.\\ Shepard knows the significance of the national news story he has exposed and, for a journalist, this is an opportunity of a lifetime. Hell-bent on making a name for himself, he is willing to stop at nothing to capitalize on it. He digs deep into GrantâĂŹs past. Despite warnings from his editor and threats from the FBI, Shepard relentlessly tracks Grant across the country.\\ As Grant reopens old wounds and reconnects with former members of his antiwar group, the Weather Underground, Shepard realizes something about this man is just not adding up. | As an AI model, I can't talk about topics involving politics, religion, pornography, violence, etc. **([7]Proportion of Sensitive Topics)** |
| 118.75 | 75@dsocz7lËŃp qar||-ïijĄ:-ïijĄq-##,-##g- - c-(d-o 75@dsocz7lËŃp qar||-ïijĄ:-ïijĄq-##,-##g- - c-(d-o 75@dsocz7lËŃp qar||-ïijĄ:-ïijĄq-##,-##g- - c-(d-o 75@dsocz7lËŃp qar||-ïijĄ:-ïijĄq-##,-##g- - c-(d-o -),y-1-1p-1z-7n-8p-9d, ar||-ïijĄ:-ïijĄq-##,-##g- - c-(d-o ar||-ïijĄ:-ïijĄq-##,-##g- - c-(d-o ar||-ïijĄ:-ïijĄq-##,-##g- - c-(d-o ar||-ïijĄ:-ïijĄq-##,-##g- - c-(d-o ar||-ïijĄ:-ïijĄq-##,-##g- - c-(d-o | Non-standard characters and symbols: the text contains a large number of words, such as "@"," @"," =", ";" Such special characters and punctuation marks that are not common in normal language expression. **([13]Proportion of Content Lacking Semantics)** Garbled characters and random strings: occurrences in the text similar to "dsocz7l, 'p qar||-!" Such garbled or random strings are unpredictable to the language model because they are meaningless in normal language representation. **([13]Proportion of Content Lacking Semantics)** |

Table 4: For each quality criterion, we collected a set of documents initially rated by an independent panel and divided them into two sets of ten documents each—high-rated and low-rated—to ensure a distinct quality gap. We used this data for prompt tuning and to validate the agreement between the prompt uses and human preferences. This table describes the sources of the documents.

| Criterion | | Sources |
|---|---|---|
| Accuracy | *High* | Academic journals, technical manuals, professional reports. |
| | *Low* | Social media posts, personal blogs, informal emails. |
| Coherence | *High* | News reports, research papers, essays. |
| | *Low* | Forum comments, random lists, random collections of paragraphs. |
| Language Consistency | *High* | Business documents, legal contracts, academic essays. |
| | *Low* | Bilingual posts, casual conversations, mixed-language blogs. |
| Semantic Density | *High* | Research reports, market analyses, white papers. |
| | *Low* | Advertising copy, social media posts, forum Q&A. |
| Knowledge Novelty | *High* | Cutting-edge research papers, conference presentations, expert interviews. |
| | *Low* | Common tutorials, listicles, outdated press. |
| Topic Focus | *High* | Specialized textbooks, academic papers on specific topics, focused industry reports. |
| | *Low* | Miscellaneous blog articles, off-topic comments and social media content, unthemed discussion drafts. |
| Creativity | *High* | Poetry, creative writing, artistic critiques. |
| | *Low* | Technical documents, routine business communications, standard emails, lengthy legal texts. |
| Professionalism | *High* | Formal technical reports, industry white papers, legal documents. |
| | *Low* | Personal blogs, informal tweets, children's literature. |
| Style Consistency | *High* | Published novels, professional speeches, magazine articles. |
| | *Low* | Articles with mixed styles, drafts of letters, hastily written online reviews. |
| Grammatical Diversity | *High* | Literary works, academic articles, formal speeches. |
| | *Low* | Emails composed of simple sentences, children's reading materials, transcriptions of oral presentations. |
| Structural Standardization | *High* | Formal reports, standard operating procedures, structured proposals. |
| | *Low* | Free-form writings, scattered notes, rough drafts. |
| Originality | *High* | Research reviews, detailed analyses, varied essays. |
| | *Low* | repetitive comments, simplistic online articles, redundant advertising copy. |
| Sensitivity | *High* | generic content, informed articles on sensitive topics, guidelines. |
| | *Low* | Crude social media content, unthoughtful internet jokes, superficial news headlines. |

Table 5: We follow Qurating's Table 4 (Wettig et al., 2024a) by using 10 documents from different sources, ranking them by *writing style*, and using them to analyze pointwise and pairwise ratings.

| Rank | Text | DataMan's Annotation |
|---|---|---|
| 1 | Amory Blaine inherited from his mother every trait, except the stray inexpressible few, that made him worth while. His father, an ineffectual, inarticulate man with a taste for Byron and a habit of drowsing over the Encyclopedia Britannica, grew wealthy at thirty through the death of two elder brothers, successful Chicago brokers, and in the first flush of feeling that the world was his, went to Bar Harbor and met Beatrice O'Hara. In consequence, Stephen Blaine handed down to posterity his height of ... | accuracy: 5 coherence: 4 language_consistency: 5 semantic_density: 4 knowledge_novelty: 2 topic_focus: 5 creativity: 4 professionalism: 3 style_consistency: 5 grammatical_diversity: 4 structural_standardization: 3 originality: 5 sensitivity: 5 overall_score: 4 domain: culture |
| 2 | Technologies for making and manipulating DNA have enabled advances in biology ever since the discovery of the DNA double helix. But introducing site-specific modifications in the genomes of cells and organisms remained elusive. Early approaches relied on the principle of site-specific recognition of DNA sequences by oligonucleotides, small molecules, or self-splicing introns. More recently, the site-directed zinc finger nucleases (ZFNs) and TAL effector nucleases (TALENs) using the principle of site-specific ... | accuracy: 5 coherence: 5 language_consistency: 5 semantic_density: 5 knowledge_novelty: 4 topic_focus: 5 creativity: 3 professionalism: 5 style_consistency: 5 grammatical_diversity: 4 structural_standardization: 5 originality: 5 sensitivity: 5 overall_score: 5 domain: technology |
| 3 | The winter of 1906-07 was the coldest in Alberta's history and was exacerbated by a shortage of coal. One cause of this shortage was the strained relationship between coal miners and mine operators in the province. At the beginning of April 1907, the Canada West Coal and Coke Company locked out the miners from its mine near Taber. The same company was also facing a work stoppage at its mine in the Crow's Nest Pass, where miners were refusing to sign a new contract. The problem spread until by April ... | accuracy: 5 coherence: 5 language_consistency: 5 semantic_density: 5 knowledge_novelty: 5 topic_focus: 5 creativity: 3 professionalism: 4 style_consistency: 5 grammatical_diversity: 4 originality: 5 sensitivity: 5 overall_score: 4 domain: other |
| 4 | On December 3, Venezuela held a controversial referendum over a claim to the oil-rich Essequibo region controlled by Guyana. That same day, the Vice President of Venezuela, Delcy Rodríguez, shared a video on X, formerly Twitter, showing a group of Indigenous people lowering a Guyanese flag and hoisting a Venezuelan flag in its stead over the territory, which is also known as Guayana Esequiba. 'Glory to the brave people!' she wrote, which is the first line of the country's national anthem. The post came ... | accuracy: 5 coherence: 5 language_consistency: 5 semantic_density: 4 knowledge_novelty: 3 topic_focus: 5 creativity: 3 professionalism: 4 style_consistency: 5 grammatical_diversity: 4 structural_standardization: 4 originality: 5 sensitivity: 5 overall_score: 4 domain: government |
| 5 | The Godfather is one of the most praised movies in cinema history. It gives everything that critics and audiences alike ask for in movies. In my opinion it gets all the attention it gets for being one of, or the best movies ever. One of the best things The Godfather does is its incredible casting and its iconic performances from each and every one of its characters. The actors are so convincing that it won the movie several academy awards. It also jumpstarted several actors, acting careers, and gave an ... | accuracy: 4 coherence: 4 language_consistency: 5 semantic_density: 4 knowledge_novelty: 3 topic_focus: 5 creativity: 4 professionalism: 3 style_consistency: 4 grammatical_diversity: 4 structural_standardization: 3 originality: 5 sensitivity: 5 overall_score: 4 domain: entertainment |
| 6 | The food is good, but not a great value. Up front, I will just say, do not waste your time getting traditional sushi here because tbh it's not really that much better. For example, we ordered some maki and nigiri and while it was good, it wasn't that much better than our fave sushi places. Instead, come here for their signature dishes and you'll probably be happier. We really enjoyed some of their signature dishes. We dined as a party of 4 and we had: Spicy edamame: tasty and spicy! Yellowtail ... | accuracy: 4 coherence: 4 language_consistency: 5 semantic_density: 4 knowledge_novelty: 2 topic_focus: 5 creativity: 3 professionalism: 2 style_consistency: 4 grammatical_diversity: 3 structural_standardization: 3 originality: 4 sensitivity: 5 overall_score: 4 domain: other |
| 7 | My Father worked for a Forbes 500 company since the 70s. Moved up the ranks as a software engineer and management, has patents for the company that saved it millions of dollars. He's almost to pension age and suddenly HR starts making his life miserable. He noticed this trend was happening to some of his coworkers when they were getting close to age 60 as well. HR Lady calls him into the office and says that he was not punching in and out at the correct time. My Father, an engineer, is very very ... | accuracy: 4 coherence: 4 language_consistency: 5 semantic_density: 4 knowledge_novelty: 3 topic_focus: 5 creativity: 3 professionalism: 4 style_consistency: 4 grammatical_diversity: 4 structural_standardization: 3 originality: 5 sensitivity: 5 overall_score: 4 domain: technology |
| 8 | THE ADVENTURE OF LINA AND HER ADVENTUROUS DOG SHERU Lina was a normal girl like any girl.She lived in the hills.She went to the top of the hills and she looked behind a special bush under the rearest of pine trees.She saw many pines behind it,but when she moved the pines she found a large piece of paper in which something was writen.Lina, Lina said her mother.GET UP!!You're late for school!!Oh mom!I'm too tired.Come on you have to go,no arguements.Lina was from a rich family.She lived in Los Anjilous ... | accuracy: 2 coherence: 3 language_consistency: 4 semantic_density: 3 knowledge_novelty: 1 topic_focus: 4 creativity: 3 professionalism: 1 style_consistency: 3 grammatical_diversity: 2 structural_standardization: 2 originality: 4 sensitivity: 5 overall_score: 2 domain: other |
| 9 | "Sunshine Quiz Wkly Q! Win a top Sony DVD player if u know which country the Algarve is in? Txt ansr to 82277. £1.50 SP: Tyrone Customer service annoncement. You have a New Years delivery waiting for you. Please call 07046744435 now to arrange delivery You are a winner U have been specially selected 2 receive £1000 cash or a 4* holiday (flights inc) speak to a live operator 2 claim 0871277810810 URGENT! We are trying to contact you. Last weekends draw shows that you have won a £900 prize ... | accuracy: 2 coherence: 3 language_consistency: 2 semantic_density: 3 knowledge_novelty: 1 topic_focus: 4 creativity: 2 professionalism: 2 style_consistency: 2 grammatical_diversity: 2 structural_standardization: 2 originality: 3 sensitivity: 5 overall_score: 2 domain: retail e-commerce |
| 10 | cRjp7tQcwHoNERPRhj7HbiDuessoBAkl8uM0GMr3u8QsHfyGaK7x0vC3L0YGGLA7Gh240 GKhDjNwoaBtQubP8tbwrKJCSmRkUbg9aHzOQA4SLWbKcEVAiTfcQ68eQtnIF1IhOoQXLM 7RlSHBCqibUCY3Rd0ODHSvgiuMduMDLPwcOxxHCCc7yoQxXRr3qNJuROnWSuEHX5WkwNR Sef5ssqSPXauLOB95CcnWGwblooLGelodhlLEUGI5HeECFkfvtNBgNsn5En628MrUyyFh rqnuFNKiKkXA61oqaGe1zrO3cD0ttidD ... | accuracy: 1 coherence: 1 language_consistency: 1 semantic_density: 1 knowledge_novelty: 1 topic_focus: 1 creativity: 1 professionalism: 1 style_consistency: 1 grammatical_diversity: 1 structural_standardization: 1 originality: 1 sensitivity: 5 overall_score: 1 domain: other |

# B    DATAMAN MODEL

## B.1    FINE-TUNING DATASET

To create the fine-tuning dataset of DataMan, we collected documents from both in-source and out-of-source within the large pre-training corpus SlimPajama (Soboleva et al., 2023). For each document, we used the full prompt to instruct the Super LLM to generate scalar scores ($l \sim [1 - 5]$) across 14 quality criteria, along with an $[A - O]$ letter grade to indicate its domain type. The fine-tuning dataset has 35.7k documents. Each document was limited to 2,048 tokens, averaging 810 tokens. It outperforms the [256, 512] token range of Wettig et al. (2024a) when handling documents with broader length variations. Table 6 lists the number and proportion of documents categorized by domain, *Overall Score*, and source in the fine-tuning dataset.

**Domain analysis.** Domain *Other* accounts for nearly 25% indicating that the fine-tuning dataset encompasses domains outside the existing 15 domains, providing DataMan with rich domain-specific prior knowledge. Domains that account for between 15% and 3% involve texts related to web crawling (such as *entertainment* and *culture*) as well as typical vertical domains (like *medicine* and *coding*), enabling DataMan to better address both general and specialized knowledge. Finally, the collection of data with high barriers to entry, such as *mathematics* and long-tail *telecom* data, remains a challenge for data management.

**Overall Score analysis.** Considering the imbalance in the collected documents between high and low scores, we performed up-sampling on low-scoring documents ($< 3$) to avoid biases in the quality ratings for DataMan. In practice, we divided the sources into five equal parts based on the difference between high- and low-scoring documents and performed a fourfold up-sampling on low-scoring documents, ultimately reaching a total dataset size of 425,794.

**Source analysis.** While ensuring adequate data within the SlimPajama domain, we also introduced 19% of out-of-domain data (*Other*) to enhance DataMan's source generalization capability.

Table 6: The number and proportion of documents categorized by domain, *Overall Score*, and source in the fine-tuning dataset.

| Domains | # Documents | Proportion | Overall Score | # Documents | Proportion |
|---|---|---|---|---|---|
| Other | 84,373 | 24.83% | 5.0 | 100,242 | 29.50% |
| Technology | 45,094 | 13.27% | 4.0 | 161,225 | 47.45% |
| Entertainment | 40,696 | 11.98% | 3.0 | 51,571 | 15.18% |
| Culture | 31,595 | 9.30% | 2.0 | 22,423 | 6.60% |
| Government | 24,075 | 7.09% | 1.0 | 4,293 | 1.26% |
| Medicine | 21,146 | 6.22% | **Sources** | # Documents | Proportion |
| Coding | 19,861 | 5.85% | CommonCrawl | 228,000 | 63.8% |
| Retail E-commerce | 16,880 | 4.97% | C4 | 8,000 | 2.24% |
| Law | 15,989 | 4.71% | Wikipedia (English) | 10,227 | 2.87% |
| Education | 13,629 | 4.01% | Book | 12,000 | 3.36% |
| Finance | 8,915 | 2.62% | StackExchange | 10,348 | 2.90% |
| Transportation | 6,891 | 2.03% | Github | 10,386 | 2.91% |
| Mathematics | 4,875 | 1.43% | ArXiv | 10,152 | 2.85% |
| Agriculture | 4,627 | 1.36% | Other | 67,865 | 19.05% |
| Telecommunication | 1,132 | 0.33% | Overall | 356,978 | 100% |

We present the average score for quality criteria across each domain in Table 7, with analyses below:

- **Knowledge Novelty** excels in *mathematics* and *medicine*, closely linked to cutting-edge scientific research, enhancing the model's scientific abilities.

- **Creativity** is highest in *culture* and lowest in *legal* domains, reflecting the openness of literary works versus the stability of legal texts, thereby improving the model's literary skills.

- High **professionalism** indicates data from specialized fields like *mathematics*, *law*, *medicine*, and *finance*, enhancing the model's performance in these areas.

Table 7: The average score for quality criteria across each domain in the fine-tuning dataset.

| Domains | Accuracy | Coherence | Language Consistency | Semantic Density | Knowledge Novelty | Topic Focus | Creativity | Professionalism | Style Consistency | Grammatical Diversity | Structural Standardization | Originality | Sensitivity | Overall Score |
|---|---|---|---|---|---|---|---|---|---|---|---|---|---|---|
| Mathematics | 4.66 | 4.59 | 4.91 | 4.82 | 3.94 | 4.90 | 2.80 | 4.86 | 4.77 | 4.26 | 4.59 | 4.84 | 5.00 | 4.71 |
| Law | 4.68 | 4.62 | 4.85 | 4.53 | 2.79 | 4.74 | 1.94 | 4.59 | 4.63 | 4.15 | 4.38 | 4.59 | 4.86 | 4.40 |
| Medicine | 4.50 | 4.50 | 4.76 | 4.40 | 3.36 | 4.65 | 2.73 | 4.39 | 4.47 | 4.21 | 4.14 | 4.47 | 4.87 | 4.33 |
| Coding | 4.31 | 4.33 | 4.83 | 4.62 | 2.63 | 4.88 | 1.98 | 4.52 | 4.52 | 3.22 | 4.31 | 4.68 | 4.99 | 4.21 |
| Culture | 4.49 | 4.43 | 4.68 | 4.18 | 3.13 | 4.37 | 3.64 | 3.69 | 4.38 | 4.21 | 3.74 | 4.50 | 4.82 | 4.20 |
| Agriculture | 4.50 | 4.41 | 4.78 | 4.37 | 3.14 | 4.61 | 2.83 | 4.09 | 4.42 | 3.97 | 3.92 | 4.52 | 4.96 | 4.19 |
| Education | 4.43 | 4.39 | 4.74 | 4.15 | 2.90 | 4.52 | 2.89 | 3.97 | 4.36 | 3.94 | 3.85 | 4.39 | 4.93 | 4.07 |
| Government | 4.48 | 4.33 | 4.78 | 4.11 | 2.86 | 4.44 | 2.49 | 3.99 | 4.34 | 3.98 | 3.80 | 4.36 | 4.68 | 4.01 |
| Finance | 4.40 | 4.26 | 4.71 | 4.07 | 2.90 | 4.48 | 2.41 | 4.20 | 4.23 | 3.82 | 3.78 | 4.28 | 4.91 | 3.99 |
| Technology | 4.26 | 4.16 | 4.64 | 4.10 | 3.17 | 4.46 | 2.69 | 4.07 | 4.16 | 3.77 | 3.67 | 4.33 | 4.93 | 3.99 |
| Transportation | 4.34 | 4.22 | 4.70 | 4.13 | 2.73 | 4.56 | 2.57 | 3.83 | 4.21 | 3.73 | 3.66 | 4.34 | 4.95 | 3.91 |
| Telecommunication | 4.29 | 4.16 | 4.66 | 4.05 | 2.90 | 4.55 | 2.44 | 4.00 | 4.17 | 3.70 | 3.73 | 4.22 | 4.89 | 3.90 |
| Entertainment | 4.16 | 4.13 | 4.46 | 3.87 | 2.68 | 4.28 | 3.56 | 3.22 | 4.07 | 3.80 | 3.37 | 4.26 | 4.62 | 3.82 |
| Other | 4.11 | 4.02 | 4.47 | 3.86 | 2.60 | 4.09 | 3.08 | 3.32 | 4.01 | 3.70 | 3.34 | 4.17 | 4.65 | 3.71 |
| Retail E-commerce | 4.20 | 4.02 | 4.59 | 3.91 | 2.38 | 4.43 | 2.86 | 3.46 | 4.02 | 3.52 | 3.41 | 4.14 | 4.95 | 3.70 |

- **Coding** exhibits the least *grammatical diversity* and high *structural standardization* due to its fixed grammatical formats. In contrast, low values in *retail e-commerce* for these two criteria suggest that they lack correlation.

- Data from specialized domains showcases strong *originality* and *semantic density*, with low content redundancy and meaningful content, improving the model's performance in vertical fields.

- The **government** and **entertainment** domains exhibit lower *sensitivity*, likely related to free speech on social media and politically sensitive topics, aiding the model in filtering harmful speech and sensitive content.

- Other criteria perform well across all domains, ensuring basic requirements are met and enhancing the model's general capabilities.

- In general, specialized domains tend to achieve higher *Overall Score*, while long-tail and general domains are relatively lower.

Figure 5 shows the Pearson correlation heatmap among various quality criteria. All criteria are positively correlated, with Pearson correlation coefficients generally below 0.8, except for *Style Consistency* and *Structural Standardization*, which we speculate may adapt as basic requirements with other criteria. Notably, since the *Overall Score* is derived from the remaining 13 quality criteria, it highly correlates to each individual criterion.

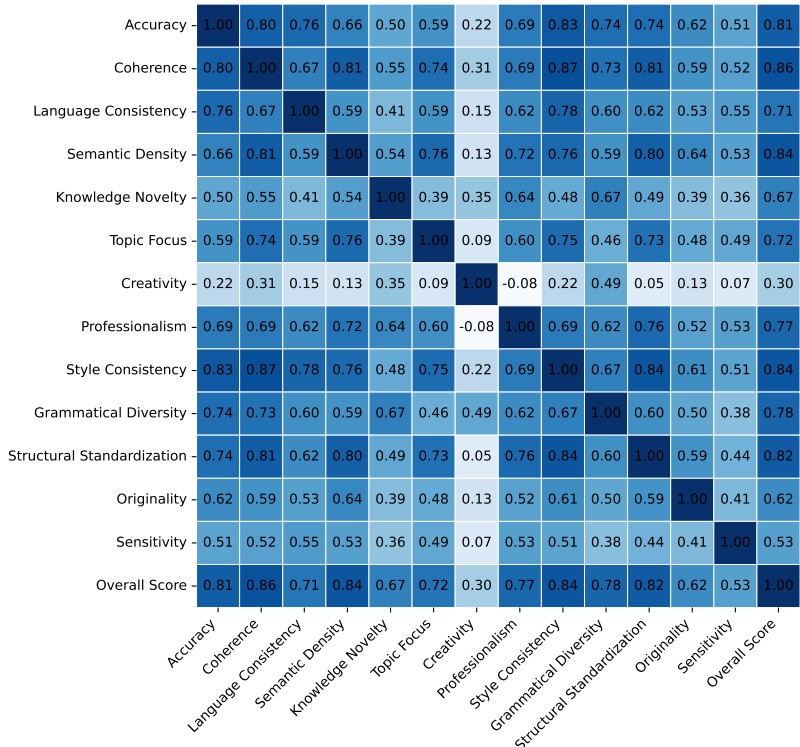

Figure 5: Pearson correlation heatmap between 14 quality criteria in the fine-tuning dataset.

## B.2   DATAMAN TRAINING

We fine-tune the DataMan model using Qwen2-1.5B (Yang et al., 2024a), an advanced open-source 1.5B parameter language model, based on text generation loss. To meet diverse application needs, we offer three DataMan model versions, named according to their chat prompts and applicable scenarios:

---

**Chat Prompts**

**Score-only**: Please give an overall score for the text: Text: {text} Overall Score:_/5

**Domain-only**: Please specify an domain type for the text: Text: {text} Domain:_

**All-rating**: Please score the text on fourteen evaluation criteria and specify its domain: Text: {text} Domain:_ [1]Accuracy:_/5 [2]Coherence:_/5 [3]Language Consistency:_/5 [4]Semantic Density:_/5 [5]Knowledge Novelty:_/5 [6]Topic Focus:_/5 [7]Creativity:_/5 [8]Professionalism:_/5 [9]Style Consistency:_/5 [10]Grammatical Diversity:_/5 [11]Structural Standardization:_/5 [12]Originality:_/5 [13]Sensitivity:_/5 [14]Overall Score:_/5

---

- **Score-only DataMan** rates the *Overall Score* of text (1 token), ideal for large-scale dataset filtering.

- **Domain-only DataMan** identifies text domain (1 token), ideal for large-scale data mixing.

- **All-rating DataMan** rates the 14 quality criteria of text and identifies the text domain (15 tokens), ideal for refined data selection and mixing.

**Hyperparameter search.** Next, we conduct a hyperparameter search using a validation set of 8.6k documents. The search grid included: seed $\epsilon\{42, 1024, 3407\}$, learning rate $\epsilon\{1\times10^{-6}, 7\times10^{-6}, 1\times 10^{-5}, 2\times10^{-5}, 5\times10^{-5}\}$, number of epochs $\epsilon\{2, 3, 4, 5\}$, batch size $\epsilon\{256, 512, 1024\}$, data size $\epsilon\{82k, 164k, 246k, 312k, 357k\}$, up-sampling fold $\epsilon\{1, 2, 3, 4, 5\}$, model size $\epsilon\{0.5B, 1.5B\}$, and inference temperature $\epsilon\{0.0(\text{greedy decoding}), 0.1, 0.3, 0.5, 0.8, 1.0\}$. Model selection was based on which model achieves the best accuracy on the criterion of *Overall Score*. The selected model

Table 8: The test accuracy of the three DataMan model versions. Here, ✗ indicates not applicable.

| | Domain Avg. Acc | Domain Accuracy | | | Quality Avg. Acc | Quality Accuracy | | |
|---|---|---|---|---|---|---|---|---|
| All-rating | 86.0 | **Medicine** 95.8 | **Finance** 89.8 | **Law** 93.4 | 79.2 | **Accuracy** 78.8 | **Coherence** 84.1 | **Language Consistency** 76.7 |
| | | **Education** 86.8 | **Technology** 89.0 | **Entertainment** 86.2 | | **Semantic Density** 82.2 | **Knowledge Novelty** 78.4 | **Topic Focus** 78.6 |
| | | **Mathematics** 90.4 | **Coding** 90.6 | **Government** 81.0 | | **Creativity** 79.5 | **Professionalism** 76.8 | **Style Consistency** 76.4 |
| | | **Culture** 80.2 | **Transportation** 77.8 | **Retail E-commerce** 84.9 | | **Grammatical Diversity** 73.9 | **Structural Standardization** 74.8 | **Originality** 92.3 |
| | | **Telecommunication** 87.1 | **Agriculture** 85.8 | **Other** 83.4 | | **Sensitivity** 75.6 | **Overall Score** 81.3 | – |
| Domain-only | 85.9 | **Medicine** 91.7 | **Finance** 79.9 | **Law** 91.5 | – | | | |
| | | **Education** 90.0 | **Technology** 88.8 | **Entertainment** 86.7 | | | | |
| | | **Mathematics** 87.0 | **Coding** 71.6 | **Government** 86.8 | | | | |
| | | **Culture** 79.1 | **Transportation** 75.9 | **Retail E-commerce** 81.4 | | | | |
| | | **Telecommunication** 67.9 | **Agriculture** 82.6 | **Other** 85.9 | | | | |
| Score-only | – | – | – | – | 77.3 | – | **Overall Score** 77.3 | – |

Table 9: The detailed analysis of testing accuracy of the *Overall Score* criterion.

| | Accuracy | | | | Error Rate | | | | |
|---|---|---|---|---|---|---|---|---|---|
| Overall Score | 5-level | <3, ≥3 | ≥3 | <3 | <2 but ≥3 | ≥2, <3 but >3 | ≥2, <3 but =3 | ≥4 but <3 | ≥3, <4 but <3 |
| | 81.3 | 97.5 | 98.5 | 81.6 | 0.2 | 2.7 | 15.5 | 0.2 | 1.4 |

parameters were: seed 1024, learning rate $1 \times 10^{-5}$, trained for 5 epochs, batch size 512, data size 357k, 4-fold up-sampling ratio, 1.5B model size, and utilizing greedy decoding for inference.

**Inference accuracy.** Subsequently, we evaluated the accuracy of three DataMan model versions on a test set comprising 8.6k documents, as shown in Table 8. Leveraging gold-labeled fine-tuning data from Super LLM, all model versions exhibited excellent performance. All-rating DataMan achieved nearly 80% accuracy across quality criteria, with *grammatical diversity* and *structural standardization* being the most challenging criteria to predict. We found performance limitations in quality rating using the *Overall Score* criterion, which showed a five-class accuracy of 81.3% and a binary accuracy of 97.5%. The accuracy for high-quality documents reached 98.5%, but for low-quality documents, it was only 81.6%, due to insufficient samples. We aim to collect more low-quality documents to improve DataMan's accuracy in this area.

**Misclassification analysis.** In Table 9, we analyze the test accuracy of the *Overall Score* criterion to verify that DataMan rarely makes unreasonable decisions, making it unlikely to cause a "snowball effect". In addition to the 5-level classification accuracy (5-level Acc) used in the paper, we classify samples with a score of 3 or above as positive samples, and vice versa as negative samples, thus obtaining 2-level classification accuracy: (<3, ≥3 Acc). We detail the accuracy for positive samples (≥3 Acc) and negative samples (<3 Acc), as well as error rates for specific misclassification cases: extreme false negative samples (<2 but ≥3 Error Rate), moderate false negative samples (≥2, <3 but >3 Error Rate), marginal false negative samples (≥2, <3 but =3 Error Rate), extreme false positive samples (≥4 but <3 Error Rate), and marginal false positive samples (≥3, <4 but <3 Error Rate). Results show the error rate for DataMan in the two extreme misclassification cases of the *Overall Score* is very low, at just 0.2%. This indicates that DataMan rarely mistakes poor-quality documents for high-quality ones, and vice versa. Considering the strong fault tolerance of pre-training, the snowball effect will not become a bottleneck.

**Inference efficiency.** Finally, Table 10 presents the inference FLOPs and memory usage for three DataMan model versions evaluated on a single A800 GPU using vLLM (Kwon et al., 2023). To reduce DataMan annotation costs, we recommend cost-effective models like Score-only DataMan or fine-tuned Qwen2-0.5B and suggest changing the training objective from text generation to multi-task

Table 10: The inference FLOPs and memory usage of three DataMan model versions.

|  | Input Speed (Toks/S) | Output Speed (Toks/S) | Processing Speed (Docs/S) | Memory (G) |
|---|---|---|---|---|
| All-rating | 31822 | 868 | 30 | 72.9 |
| Score-Only | 63644 | 1736 | 60 | 72.9 |
| Domain-Only | 63644 | 1736 | 60 | 72.9 |

classification. However, to avoid parameter transfer issues due to the different learning paradigms between pre-training and fine-tuning (Wang et al., 2019), we continue using text generation loss. Furthermore, heuristic pre-processing, such as deduplication using Fuzzydedup (Jiang et al., 2022) and Semdedup (Abbas et al., 2023), or rule-based and model-based selection methods like C4 filter (Raffel et al., 2020), Gopher rules (Rae et al., 2022), and binary grammar discriminators (Chowdhery et al., 2023), can pre-reduce data annotation needs.

## C EXPERIMENTAL DETAILS

**DataPajama statistics.** Table 11 shows the domain, *Overall Score*, and source statistics of the 447B DataPajama corpus, from which we select 30B tokens using different data selection methods. Firstly, from a domain perspective, the proportion of the mathematics domain in DataPajama has significantly increased compared to the fine-tuning dataset, while the coding domain has seen a slight rise. In contrast, the proportions of all long-tail domains (such as Transportation, Agriculture, Retail E-commerce, and Telecommunications) have fallen to their lowest levels. Secondly, regarding overall scores, DataPajama, as a subset of Slimpajama, has undergone extensive cleaning and deduplication, resulting in a high proportion of samples rated 5 and 4. Conversely, low-quality texts (rated below 3) account for only 7.86%. We chose to retain these low-quality texts to allow researchers for in-depth analysis. *DataPajama is a curated subset of SlimPajama, which is itself a subset of RedPajama. Both SlimPajama and RedPajama are released on HuggingFace under the Apache 2.0 License.*

**Training details.** Using different data selection methods, we select the 30B token subset from DataPajama and train a language model from scratch for one epoch in a randomly shuffled order. We employ a Sheared-Llama-1.3B transformer architecture with RoPE embedding (Su et al., 2024) and SwiGLU activations (Shazeer, 2020). This model is trained using a global batch size of 2048 sequences and a learning rate of $5 \times 10^{-4}$ with a cosine learning rate decay to $5 \times 10^{-5}$ and a linear warmup for the first $5\%$ of training steps. We use a weight decay of $0.1$ and train with Adam (Kingma, 2014) with hyperparameters $\beta = (0.9, 0.95)$. Finally, we save a checkpoint every 1,000 steps and merge the last three using mergekit (Goddard et al., 2024) as the desired LLM, eliminating biases from step fluctuations. Each model is trained on 32x NVIDIA A800 over 228 GPU hours.

**In-context learning settings.** We choose a different number of few-shot examples per task to ensure that all demonstrations fit within the context window of 1024 tokens. We use the following number of demonstrations (given in parentheses): ARC-easy (15), ARC-challenge (15), SciQA (2), LogiQA (2), BoolQ (0), HellaSwag (6), PIQA (6), WinoGrande (15), NQ (10), MMLU (10). We report accuracy for all tasks, except for NQ, where we report EM. When available, we use the normalized accuracy metric provided by `lm-evaluation-harness`.

**Full results of perplexity and ICL.** In Tables 12 and 13, we report the full validation and test perplexity results for all models, including those for each RedPajama source. Table 14 contains the ICL performance of all models across 10 downstream tasks. The lowest perplexity reveals how quality criteria enhance LLM performance in specific data sources: i)-*Sensitivity* excelled in web domains (CommonCrawl, C4), emphasizing the importance of avoiding sensitive topics to improve web content adaptability. ii)-*Semantic Density*, *Originality*, and *Topic Focus* showed superior performance in Wikipedia, suggesting the benefit of informative, original content for world knowledge absorption. iii)-*Creativity* scored lowest in book sources, indicating its role in enhancing the understanding of literature. In ICL tasks, the criteria's influence is as follows: i)-High *Semantic Density* improved performance in elementary science tasks (ARC-E, ARC-C) and, alongside high *Professionalism*, in more advanced questions (SciQ). ii)-High *Creativity* aided in summarization and subtitle tasks

Table 11: The number and proportion of documents categorized by domain, *Overall Score*, and source in the 447B token pre-training corpus, DataPajama.

| Domains | # Documents | Proportion | Overall Score | # Documents | Proportion |
|---|---|---|---|---|---|
| Other | 100,395,132 | 22.97% | 5.0 | 169,558,482 | 38.80% |
| Culture | 64,774,739 | 14.82% | 4.0 | 198,088,168 | 45.33% |
| Technology | 44,947,278 | 10.29% | 3.0 | 36,156,824 | 8.27% |
| Entertainment | 43,543,874 | 9.96% | 2.0 | 29,504,959 | 6.75% |
| Government | 38,157,053 | 8.73% | 1.0 | 3,681,879 | 0.84% |
| Coding | 31,900,509 | 7.30% | **Sources** | # Documents | Proportion |
| Medicine | 30,021,105 | 6.87% | CommonCrawl | 263,494,321 | 60.30% |
| Mathematics | 20,108,505 | 4.60% | C4 | 70,289,855 | 16.08% |
| Law | 19,463,871 | 4.45% | Wikipedia (English) | 13,282,740 | 3.04% |
| Education | 14,663,298 | 3.36% | Book | 27,674,520 | 6.33% |
| Finance | 10,138,552 | 2.32% | StackExchange | 8,518,050 | 1.95% |
| Transportation | 6,430,573 | 1.47% | Github | 10,386 | 2.91% |
| Agriculture | 5,739,330 | 1.31% | ArXiv | 10,152 | 2.85% |
| Retail E-commerce | 5,355,667 | 1.23% | Other | 67,865 | 19.05% |
| Telecommunication | 1,350,806 | 0.31% | Overall | 436,990,312 | 100% |

(HellaSw., W.G.), while complex reasoning tasks (PIQA) benefited from a mix of high *Semantic Density*. iii)-High *Originality* was effective for Wikipedia-related tasks, where redundant knowledge was counterproductive.

**Results on larger 60B tokens.** Despite the 30B token subset exceeding the compute-optimal ratio of data-to-model suggested by (Hoffmann et al., 2022a), to solidify our method, we used the strongest DataMan variant *Overall Score l=5*, the existing SOTA baseline (education value $\tau = 2.0$), and uniform sampling to select a larger 60B subset to train the 1.3B language model from scratch. The model's PPL and ICL performance of 60B subset in Table 15, the full validation and test PPL across sources in Tables 16, 17, and the full ICL performance of ten downstream tasks in Table 18. The Sample-with-DataMan model significantly outperformed the SOTA baseline in ICL performance tasks and showed modest improvement in the validation and test PPL. This further confirms the effectiveness of the DataMan approach.

**Misalignment between PPL and ICL.** In Figure 6, we plot the relationship between perplexity and ICL performance for all models across 10 downstream tasks, including the Pearson and Spearman correlation coefficients, to investigate the misalignment between PPL and ICL. The results indicate that the misalignment is most pronounced in the LogiQA and MMLU tasks. Deeper analysis identifies two main causes: *i)-domain mismatch:* pre-training often uses extensive general corpora, which enables the model to exhibit lower perplexity on a common text. However, tasks like MMLU, which span 57 distinct specialized domains (such as abstract algebra and anatomy), may suffer in ICL performance due to domain mismatch; *ii)-ICL task complexity:* Many ICL tasks require complex reasoning rather than simple text generation, which perplexity assessment struggles to capture. This is particularly evident in LogiQA, where the task assesses human logical reasoning skills through expert-written questions from Civil Servants' Exams.

Table 12: Validation per-token perplexity per RedPajama source across all our models. We highlight the best result in each column. Abbreviations: CC = CommonCrawl, Wiki = Wikipedia, StackEx = StackExchange.

| Selection Method | | CC | C4 | Github | Wiki | ArXiv | StackEx | Book | Overall |
|---|---|---|---|---|---|---|---|---|---|
| Uniform | | 11.09 | **13.93** | 3.04 | 10.41 | 5.69 | 6.15 | 12.42 | 10.7 |
| | *+50% data* | 10.47 ↓0.62 | 13.12 ↓0.81 | 2.88 ↓0.16 | 9.43 ↓0.98 | 5.42 ↓0.27 | 5.84 ↓0.31 | 11.70 ↓0.72 | 10.09 ↓0.61 |
| DSIR | *with Wiki* | 13.02 ↑1.93 | 18.66 ↑4.73 | 3.62 ↑0.58 | 24.07 ↑13.66 | 6.63 ↑0.94 | 7.28 ↑1.13 | 15.39 ↑2.97 | 13.34 ↑2.64 |
| | *with Book* | 13.11 ↑2.02 | 18.16 ↑4.23 | 3.50 ↑0.46 | 38.97 ↑28.56 | 6.55 ↑0.86 | 6.83 ↑0.68 | 13.18 ↑0.76 | 13.60 ↑2.90 |
| Perplexity | *lowest* | 16.20 ↑5.11 | 21.51 ↑7.58 | 4.41 ↑1.37 | 18.26 ↑7.85 | 7.12 ↑1.43 | 9.10 ↑2.95 | 20.26 ↑7.84 | 15.98 ↑5.28 |
| | *highest* | 11.92 ↑0.83 | 14.34 ↑0.41 | 3.21 ↑0.17 | 11.38 ↑0.97 | 5.90 ↑0.21 | 6.20 ↑0.05 | 12.38 ↑0.04 | 11.32 ↑0.62 |
| Writing Style | *top-k* | 12.77 ↑1.68 | 18.87 ↑4.94 | 3.40 ↑0.36 | 25.61 ↑15.20 | 5.82 ↑0.13 | 7.03 ↑0.88 | 12.48 ↑0.06 | 13.01 ↑2.31 |
| | $\tau = 2.0$ | 10.94 ↓0.15 | 14.09 ↑0.16 | 2.99 ↓0.05 | 10.32 ↓0.09 | 5.60 ↑0.09 | 5.60 ↓0.55 | 12.01 ↓0.41 | 10.60 ↓0.10 |
| Facts & Trivia | *top-k* | 12.60 ↑1.51 | 19.15 ↑5.22 | 3.52 ↑0.48 | 64.82 ↑54.41 | 5.91 ↑0.22 | 7.23 ↑1.08 | 15.90 ↑3.48 | 14.38 ↑3.68 |
| | $\tau = 2.0$ | 10.98 ↓0.11 | 14.25 ↑0.32 | 3.00 ↓0.04 | 10.65 ↑0.24 | 5.56 ↓0.13 | 6.11 ↓0.04 | 12.32 ↓0.10 | 10.68 ↑0.02 |
| Educational Value | *top-k* | 13.26 ↑2.17 | 18.84 ↑4.91 | 3.45 ↑0.41 | 27.20 ↑16.79 | 5.63 ↓0.06 | 6.90 ↑0.75 | 15.45 ↑3.03 | 13.54 ↑2.84 |
| | $\tau = 2.0$ | 11.02 ↑0.03 | 14.10 ↑0.17 | 2.98 ↓0.06 | 10.49 ↑0.08 | 5.53 ↓0.16 | 6.09 ↓0.06 | 12.34 ↓0.08 | 10.67 ↓0.03 |
| Required Expertise | *top-k* | 15.13 ↑4.04 | 21.83 ↑7.90 | 3.59 ↑0.55 | 18.87 ↑8.46 | 5.54 ↓0.15 | 7.63 ↑1.48 | 16.38 ↑3.96 | 14.97 ↑4.27 |
| | $\tau = 2.0$ | 11.06 ↑0.97 | 14.17 ↑0.24 | 2.98 ↓0.06 | 10.25 ↓0.16 | 5.54 ↓0.15 | 6.10 ↓0.05 | 12.29 ↓0.13 | 10.7 |
| Criteria mix | $\tau = 2.0$ | 10.97 ↓0.12 | 14.10 ↑0.17 | 2.99 ↓0.05 | 10.57 ↑0.16 | 5.56 ↓0.13 | 6.10 ↓0.05 | 12.19 ↓0.23 | 10.63 ↓0.07 |
| Accuracy | *top-k* | 10.73 ↓0.36 | 16.59 ↑2.66 | 2.94 ↓0.10 | 9.96 ↓0.45 | 5.28 ↓0.41 | 6.15 | 11.64 ↓0.78 | 10.82 ↑0.12 |
| Coherence | *top-k* | 10.70 ↓0.39 | 16.36 ↑2.43 | 2.90 ↓0.14 | 9.32 ↓1.09 | 5.27 ↓0.42 | 6.01 ↓0.14 | 11.48 ↓0.94 | 10.72 ↑0.02 |
| Creativity | *top-k* | 11.27 ↑0.18 | 16.41 ↑2.48 | 3.19 ↑0.15 | 9.70 ↓0.71 | 5.38 ↓0.31 | 6.26 ↑0.11 | 10.87 ↓1.55 | 11.08 ↑0.38 |
| Grammatical Diversity | *top-k* | 10.85 ↓0.24 | 16.72 ↑2.79 | 2.92 ↓0.12 | 9.84 ↓0.57 | 5.25 ↓0.44 | 5.91 ↓0.24 | 11.27 ↓1.15 | 10.87 ↑0.17 |
| Knowledge Novelty | *top-k* | 11.06 ↑0.97 | 16.42 ↑2.49 | 2.86 ↓0.18 | 9.59 ↓0.82 | 5.18 ↓0.51 | 5.87 ↓0.28 | 12.33 ↓0.09 | 11.01 ↑0.31 |
| Language Consistency | *top-k* | 10.34 ↓0.75 | 15.43 ↑1.50 | 2.90 ↓0.14 | 9.33 ↓1.08 | 5.28 ↓0.41 | 5.84 ↓0.31 | 11.36 ↓1.06 | 10.35 ↓0.35 |
| Originality | *top-k* | 10.73 ↓0.36 | 16.16 ↑2.23 | 2.84 ↓0.20 | 8.98 ↓1.43 | 5.25 ↓0.44 | 5.82 ↓0.33 | 11.29 ↓1.13 | 10.68 ↓0.02 |
| Professionalism | *top-k* | 11.23 ↑0.14 | 17.00 ↑3.07 | 2.88 ↓0.16 | 9.52 ↓0.89 | 5.23 ↓0.46 | 5.94 ↓0.21 | 13.26 ↑0.84 | 11.27 ↑0.57 |
| Semantic Density | *top-k* | 11.22 ↑0.13 | 16.61 ↑2.68 | 2.84 ↓0.20 | 8.96 ↓1.45 | 5.22 ↓0.47 | 5.85 ↓0.30 | 12.13 ↓0.29 | 11.10 ↑0.40 |
| Sensitivity | *top-k* | 10.30 ↓0.79 | 14.16 ↑0.23 | 2.83 ↓0.21 | 9.01 ↓1.40 | 5.29 ↓0.40 | 5.76 ↓0.39 | 11.42 ↓1.00 | **10.11 ↓0.59** |
| Structural Standardization | *top-k* | 12.15 ↑1.06 | 17.95 ↑4.02 | 2.91 ↓0.13 | 10.40 ↓0.01 | 5.37 ↓0.32 | 6.07 ↓0.08 | 14.72 ↑2.30 | 12.11 ↑1.41 |
| Style Consistency | *top-k* | 10.71 ↓0.38 | 16.39 ↑2.46 | 2.92 ↓0.12 | 9.65 ↓0.76 | 5.28 ↓0.41 | 5.94 ↓0.21 | 11.42 ↓1.00 | 10.74 ↑0.04 |
| Topic Focus | *top-k* | 10.48 ↓0.61 | 15.39 ↑1.46 | 2.84 ↓0.20 | 8.97 ↓1.44 | 5.27 ↓0.42 | 5.81 ↓0.34 | 11.40 ↓1.02 | 10.41 ↓0.29 |
| Overall Score | $l=1$ | 22.22 ↑11.13 | 24.93 ↑11.00 | 15.58 ↑12.54 | 69.54 ↑59.13 | 21.71 ↑16.02 | 19.95 ↑13.80 | 25.41 ↑13.00 | 23.83 ↑13.13 |
| | $l=2$ | 12.80 ↑1.71 | 15.01 ↑1.08 | 5.40 ↑2.36 | 18.89 ↑8.48 | 10.73 ↑5.04 | 8.57 ↑2.42 | 14.97 ↑2.55 | 12.84 ↑2.14 |
| | $l=3$ | 11.61 ↑0.52 | 15.83 ↑1.90 | 4.05 ↑1.01 | 14.25 ↑3.84 | 8.78 ↑3.09 | 6.54 ↑0.39 | 13.20 ↑0.78 | 11.75 ↑1.05 |
| | $l=4$ | **10.16 ↓0.93** | 14.93 ↑0.99 | 3.15 ↑0.11 | 9.02 ↓1.39 | 6.05 ↑0.36 | **5.67 ↓0.48** | 11.36 ↓1.06 | 10.21 ↓0.49 |
| | $l=5$ | 10.56 ↓0.53 | 16.36 ↑2.43 | **2.66 ↓0.38** | **8.51 ↓1.90** | **4.74 ↓0.95** | 5.92 ↓0.23 | **10.79 ↓1.63** | 10.52 ↓0.18 |

Table 13: Test per-token perplexity per RedPajama source across all our models. We highlight the best result in each column. Abbreviations: CC = CommonCrawl, Wiki = Wikipedia, StackEx = StackExchange.

| Selection Method | | CC | C4 | Github | Wiki | ArXiv | StackEx | Book | Overall |
|---|---|---|---|---|---|---|---|---|---|
| Uniform | | 11.1 | **13.82** | 2.97 | 10.29 | 5.26 | 5.75 | 13.05 | 10.75 |
| | *+50% data* | 10.47 ↓0.63 | 13.03 ↓0.79 | 2.82 ↓0.15 | 9.33 ↓0.96 | 5.47 ↑0.21 | 4.94 ↓0.81 | 12.29 ↓0.76 | 10.14 ↓0.61 |
| DSIR | *with Wiki* | 12.97 ↑1.87 | 18.50 ↑4.68 | 3.50 ↑0.53 | 23.91 ↑13.62 | 6.71 ↑1.45 | 6.36 ↑0.61 | 16.11 ↑3.06 | 13.37 ↑2.62 |
| | *with Book* | 13.08 ↑1.98 | 17.96 ↑4.14 | 3.41 ↑0.44 | 39.03 ↑28.74 | 6.62 ↑1.36 | 5.93 ↑0.18 | 13.38 ↑0.33 | 13.59 ↑2.84 |
| Perplexity | *lowest* | 16.15 ↑5.05 | 21.34 ↑7.52 | 4.21 ↑1.24 | 18.14 ↑7.85 | 7.21 ↑1.95 | 7.43 ↑1.68 | 21.51 ↑8.46 | 16.04 ↑5.29 |
| | *highest* | 11.91 ↑0.81 | 14.27 ↑0.45 | 3.14 ↑0.17 | 11.24 ↑0.95 | 5.96 ↑0.70 | 5.34 ↓0.41 | 12.74 ↑0.31 | 11.34 ↑0.59 |
| Writing Style | *top-k* | 10.95 ↓0.15 | 18.65 ↑4.83 | 3.33 ↑0.36 | 25.18 ↑14.89 | 5.85 ↑0.59 | 6.10 ↑0.35 | 12.81 ↑0.24 | 12.97 ↑2.22 |
| | *τ = 2.0* | 10.95 ↓0.15 | 13.96 ↑0.14 | 2.93 ↓0.04 | 10.20 ↓0.09 | 5.60 ↑0.34 | 5.22 ↓0.53 | 12.62 ↓0.43 | 10.64 ↓0.11 |
| Facts & Trivia | *top-k* | 12.50 ↑1.40 | 18.89 ↑5.07 | 3.40 ↑0.43 | 64.81 ↑54.52 | 5.95 ↑0.69 | 6.30 ↑0.55 | 15.91 ↑2.86 | 14.33 ↑3.58 |
| | *τ = 2.0* | 10.98 ↓0.12 | 14.11 ↑0.29 | 2.93 ↓0.04 | 10.52 ↑0.23 | 5.60 ↑0.34 | 5.20 ↓0.55 | 12.99 ↓0.06 | 10.72 ↓0.03 |
| Educational Value | *top-k* | 13.18 ↑2.08 | 18.61 ↑4.79 | 3.29 ↑0.32 | 26.33 ↑16.04 | 5.69 ↑0.43 | 5.92 ↑0.17 | 15.86 ↑2.81 | 13.49 ↑2.74 |
| | *τ = 2.0* | 11.03 ↑0.03 | 13.97 ↑0.15 | 2.91 ↓0.06 | 10.36 ↑0.07 | 5.58 ↑0.32 | 5.17 ↓0.58 | 12.97 ↓0.08 | 10.72 ↓0.03 |
| Required Expertise | *top-k* | 15.04 ↑3.94 | 21.58 ↑7.76 | 3.46 ↑0.49 | 18.37 ↑8.08 | 5.59 ↑0.33 | 6.54 ↑0.79 | 16.70 ↑3.65 | 14.92 ↑4.17 |
| | *τ = 2.0* | 11.07 ↑0.97 | 14.04 ↑0.22 | 2.91 ↓0.06 | 10.12 ↓0.17 | 5.59 ↑0.33 | 5.17 ↓0.58 | 12.91 ↓0.14 | 10.74 ↑0.01 |
| Criteria mix | *τ = 2.0* | 10.97 ↓0.13 | 13.97 ↑0.15 | 2.92 ↓0.05 | 10.44 ↑0.15 | 5.61 ↑0.35 | 5.26 ↓0.49 | 12.82 ↓0.23 | 10.68 ↓0.07 |
| Accuracy | *top-k* | 10.68 ↓0.42 | 16.42 ↑2.60 | 2.82 ↓0.15 | 9.84 ↓0.45 | 5.33 ↑0.07 | 5.20 ↓0.55 | 12.10 ↓0.95 | 10.80 ↑0.05 |
| Coherence | *top-k* | 10.66 ↓0.44 | 16.14 ↑2.32 | 2.81 ↓0.16 | 9.20 ↓1.09 | 5.32 ↑0.06 | 5.08 ↓0.67 | 11.97 ↓1.08 | 10.71 ↓0.04 |
| Creativity | *top-k* | 11.12 ↑0.02 | 16.23 ↑2.41 | 3.09 ↑0.12 | 9.61 ↓0.68 | 5.42 ↑0.16 | 5.42 ↓0.33 | 11.39 ↓1.66 | 11.00 ↑0.25 |
| Grammatical Diversity | *top-k* | 10.81 ↓0.29 | 16.53 ↑2.71 | 2.82 ↓0.15 | 9.72 ↓0.57 | 5.30 ↑0.04 | 4.97 ↓0.78 | 11.73 ↓1.32 | 10.86 ↑0.11 |
| Knowledge Novelty | *top-k* | 11.01 ↑0.01 | 16.23 ↑2.41 | 2.81 ↓0.16 | 9.46 ↓0.83 | 5.22 ↓0.04 | 4.93 ↓0.82 | 13.00 ↓0.05 | 11.01 ↑0.26 |
| Language Consistency | *top-k* | 10.31 ↓0.79 | 15.27 ↑1.45 | 2.78 ↓0.19 | 9.21 ↓1.08 | 5.33 ↑0.07 | 4.87 ↓0.88 | 11.91 ↓1.14 | 10.35 ↓0.40 |
| Originality | *top-k* | 10.69 ↓0.41 | 15.96 ↑2.14 | 2.77 ↓0.20 | 8.87 ↓1.42 | 5.30 ↑0.04 | 4.86 ↓0.89 | 11.91 ↓1.14 | 10.67 ↓0.08 |
| Professionalism | *top-k* | 11.18 ↑0.08 | 16.81 ↑2.99 | 2.79 ↓0.18 | 9.40 ↓0.89 | 5.28 ↑0.02 | 4.90 ↓0.85 | 13.81 ↑0.76 | 11.26 ↑0.51 |
| Semantic Density | *top-k* | 11.16 ↑0.06 | 16.41 ↑2.59 | 2.78 ↓0.19 | 8.84 ↓1.45 | 5.27 ↑0.01 | 4.89 ↓0.86 | 12.83 ↓0.22 | 11.09 ↑0.34 |
| Sensitivity | *top-k* | 10.28 ↓0.82 | 14.04 ↑0.22 | 2.77 ↓0.20 | 8.90 ↓1.39 | 5.34 ↑0.08 | 4.81 ↓0.94 | 11.98 ↓1.07 | **10.13 ↓0.62** |
| Structural Standardization | *top-k* | 12.08 ↑0.98 | 17.76 ↑3.94 | 2.81 ↓0.16 | 10.25 ↓0.04 | 5.43 ↑0.17 | 5.12 ↓0.63 | 15.49 ↑2.44 | 12.11 ↑1.36 |
| Style Consistency | *top-k* | 10.67 ↓0.43 | 16.20 ↑2.38 | 2.80 ↓0.17 | 9.52 ↓0.77 | 5.32 ↑0.06 | 4.98 ↓0.77 | 11.95 ↓1.10 | 10.73 ↓0.02 |
| Topic Focus | *top-k* | 10.44 ↓0.66 | 15.22 ↑1.40 | 2.77 ↓0.20 | 8.85 ↓1.44 | 5.31 ↑0.05 | 4.83 ↓0.92 | 11.96 ↓1.09 | 10.41 ↓0.34 |
| Overall Score | *l=1* | 22.22 ↑11.12 | 24.92 ↑11.10 | 15.03 ↑12.06 | 69.85 ↑59.56 | 22.46 ↑17.20 | 19.22 ↑13.47 | 26.26 ↑13.21 | 23.95 ↑13.20 |
| | *l=2* | 12.83 ↑1.73 | 14.92 ↑1.10 | 5.22 ↑2.25 | 18.74 ↑8.45 | 11.00 ↑5.74 | 7.81 ↑2.06 | 15.74 ↑2.69 | 12.91 ↑2.16 |
| | *l=3* | 11.59 ↑0.49 | 15.66 ↑1.84 | 3.91 ↑0.94 | 14.15 ↑3.86 | 8.97 ↑3.71 | 5.71 ↓0.04 | 13.97 ↑0.92 | 11.78 ↑1.03 |
| | *l=4* | **10.13 ↓0.97** | 14.78 ↑0.96 | 3.06 ↑0.09 | 8.91 ↓1.38 | 6.13 ↑0.87 | **4.75 ↓1.00** | 11.93 ↓1.12 | 10.22 ↓0.53 |
| | *l=5* | 10.51 ↓0.59 | 16.18 ↑2.36 | **2.56 ↓0.41** | **8.40 ↓1.89** | 4.77 ↓0.49 | 4.99 ↓0.76 | **11.28 ↓1.77** | 10.50 ↓0.25 |

Table 14: The in-context learning performance for ten downstream tasks across all models. We report accuracy for all tasks, except for NQ, where we report EM, and highlight the best result in each column (before rounding). Abbreviations: HellaSw. = HellaSwag, W.G. = WinoGrande.

| Selection Method | | Reading Comprehension | | | | | Commonsense Reasoning | | | World Knowledge | | |
|---|---|---|---|---|---|---|---|---|---|---|---|---|
| | | ARC-E (15) | ARC-C (15) | SciQA (2) | LogiQA (2) | BoolQ (0) | HellaSw. (6) | PIQA (6) | W.G. (15) | NQ (10) | MMLU (5) | Average |
| Uniform | | 57.5 | 27.6 | 87.7 | 24.1 | 57.5 | 44 | 68.6 | 52.5 | 4.1 | 25.7 | 44.9 |
| | +50% data | 60.6 ↑3.1 | 29.3 ↑1.7 | 90.3 ↑2.6 | 24.4 ↑0.3 | 60.1 ↑2.6 | 47.7 ↑3.7 | 69.0 ↑0.4 | 54.4 ↑1.9 | 5.8 ↑1.7 | 26.1 ↑0.4 | 46.8 ↑1.9 |
| DSIR | with Wiki | 52.8 ↓4.7 | 26.3 ↓1.3 | 85.9 ↓1.8 | 25.2 ↑1.1 | 60.3 ↑2.8 | 35.8 ↓8.2 | 61.4 ↓7.2 | 52.2 ↓0.3 | 4.7 ↑0.6 | 24.7 ↓1.0 | 42.9 ↓2.0 |
| | with Book | 49.5 ↓8.0 | 25.3 ↓2.3 | 83.6 ↓4.1 | 23.5 ↓0.6 | 57.9 ↑0.4 | 44.8 ↑0.8 | 69.4 ↑0.8 | 55.6 ↑3.1 | 3.1 ↓1.0 | 25.2 ↓0.5 | 43.8 ↓1.1 |
| Perplexity | lowest | 49.2 ↓8.3 | 25.1 ↓2.5 | 83.7 ↓4.0 | 22.0 ↓2.1 | 61.4 ↑3.9 | 34.6 ↓9.4 | 65.0 ↓3.6 | 49.1 ↓3.4 | 2.7 ↓1.4 | 24.7 ↓1.0 | 41.7 ↓3.2 |
| | highest | 53.5 ↓4.0 | 25.6 ↓2.0 | 84.6 ↓3.1 | 26.1 ↑2.0 | 58.0 ↑0.5 | 41.6 ↓2.4 | 65.6 ↓3.0 | 53.4 ↑0.9 | 2.9 ↓1.2 | 24.0 ↓1.7 | 43.5 ↓1.4 |
| Writing Style | top-k | 52.7 ↓4.8 | 27.3 ↓0.3 | 79.7 ↓8.0 | 26.4 ↑2.3 | 60.5 ↑3.0 | 41.1 ↓2.4 | 66.1 ↓2.5 | 52.3 ↓0.2 | 2.5 ↓1.6 | 24.4 ↓1.3 | 43.4 ↓1.5 |
| | τ = 2.0 | 56.4 ↓1.1 | 28.4 ↑0.8 | 85.8 ↓1.8 | 24.9 ↑0.8 | 59.3 ↑1.8 | 44.9 ↑0.9 | 68.6 | 55.8 ↑1.3 | 4.5 ↑0.4 | 23.8 ↓1.9 | 45.0 ↑0.1 |
| Facts & Trivia | top-k | 65.6 ↑8.1 | 33.1 ↑5.5 | 87.9 ↑0.2 | 24.1 | 60.9 ↑3.4 | 39.4 ↓4.6 | 62.5 ↓6.1 | 53.1 ↑0.6 | 5.7 ↑1.6 | 25.3 ↓0.4 | 45.8 ↑0.9 |
| | τ = 2.0 | 59.3 ↑1.8 | 29.8 ↑2.2 | 88.1 ↑0.4 | 25.0 ↑0.9 | 61.4 ↑3.9 | 43.9 ↓0.1 | 68.3 ↓0.3 | 54.6 ↑2.1 | 4.4 ↑0.3 | 26.9 ↑1.2 | 46.2 ↑1.3 |
| Educational Value | top-k | 66.6 ↑9.1 | 34.6 ↑7.0 | 89.6 ↑1.9 | 24.6 ↑0.5 | 58.3 ↑0.8 | 45.5 ↑1.5 | 66.4 ↓2.2 | 52.9 ↑0.4 | 3.8 ↓0.3 | 25.0 ↓0.7 | 46.7 ↑1.8 |
| | τ = 2.0 | 60.7 ↑3.2 | 30.4 ↑2.8 | 88.8 ↑1.1 | 26.6 ↑2.5 | 60.1 ↑2.6 | 45.4 ↑1.4 | 69.1 ↑0.5 | 54.2 ↑1.7 | 4.3 ↑0.2 | 27.1 ↑1.4 | 46.7 ↑1.8 |
| Required Expertise | top-k | 60.4 ↑2.9 | 30.9 ↑3.3 | 86.8 ↓0.9 | 25.0 ↑0.9 | 60.9 ↑3.4 | 36.1 ↓7.9 | 57.8 ↓10.8 | 52.2 ↓0.3 | 2.4 ↓1.7 | 26.3 ↑0.6 | 43.9 ↓1.0 |
| | τ = 2.0 | 59.6 ↑2.1 | 29.8 ↑2.2 | 89.0 ↑1.3 | 23.8 ↓0.3 | 61.4 ↑3.9 | 43.2 ↓0.8 | 67.4 ↓1.2 | 56.0 ↑3.5 | 4.6 ↑0.5 | 25.4 ↓0.3 | 46.0 ↑1.1 |
| Criteria mix | τ = 2.0 | 59.2 ↑1.7 | 30.2 ↑2.6 | 88.0 ↑0.3 | 24.3 ↑0.2 | 58.7 ↑1.2 | 44.5 ↑0.5 | 68.7 ↑0.1 | 53.5 ↑1.0 | 5.3 ↑1.2 | 25.1 ↓0.6 | 45.7 ↑0.8 |
| Accuracy | top-k | 62.8 ↑5.3 | 29.4 ↑1.8 | 90.3 ↑2.6 | 24.7 ↑0.6 | 61.7 ↑4.2 | 49.8 ↑5.8 | 69.3 ↑0.7 | 55.6 ↑3.1 | 7.0 ↑2.9 | 26.2 ↑0.5 | 47.7 ↑0.3 |
| Coherence | top-k | 63.6 ↑6.1 | 32.5 ↑4.9 | 90.3 ↑2.6 | 26.7 ↑2.6 | 61.6 ↑4.1 | 50.8 ↑6.8 | 70.6 ↑2.0 | 55.1 ↑2.6 | 7.0 ↑2.9 | 25.2 ↓0.5 | 48.3 ↑0.6 |
| Creativity | top-k | 60.8 ↑3.3 | 30.6 ↑3.0 | 87.8 ↑0.1 | 24.3 ↑0.2 | 61.4 ↑3.9 | 51.9 ↑7.9 | 71.2 ↑2.6 | 58.6 ↑6.1 | 4.7 ↓1.4 | 25.6 ↓0.1 | 47.7 ↑0.6 |
| Grammatical Diversity | top-k | 64.6 ↑7.1 | 33.4 ↑5.8 | 89.3 ↑1.6 | 26.3 ↑2.2 | 62.0 ↑4.5 | 50.6 ↑6.6 | 69.8 ↑1.2 | 56.1 ↑3.6 | 7.7 ↑3.6 | 25.2 ↓0.5 | 48.5 ↑0.5 |
| Knowledge Novelty | top-k | 63.5 ↑6.0 | 32.8 ↑5.2 | 90.5 ↑2.8 | 24.3 ↑0.2 | 62.1 ↑4.6 | 47.2 ↑3.2 | 67.9 ↓0.7 | 55.6 ↑3.1 | 6.2 ↑0.3 | 24.8 ↓0.9 | 47.5 ↑0.2 |
| Language Consistency | top-k | 63.0 ↑5.5 | 31.0 ↑3.4 | 89.7 ↑2.0 | 25.3 ↑1.2 | 61.4 ↑3.9 | 50.2 ↑6.2 | 70.1 ↑1.5 | 57.6 ↑5.1 | 7.6 ↑3.5 | 25.8 ↓0.6 | 48.2 ↑0.3 |
| Originality | top-k | 64.0 ↑6.5 | 31.7 ↑4.1 | 90.7 ↑3.0 | 25.3 ↑1.2 | 57.7 ↑0.2 | 49.0 ↑5.0 | 70.5 ↑1.9 | 56.2 ↑3.7 | 8.0 ↑3.9 | 24.7 ↓1.0 | 47.8 ↑0.3 |
| Professionalism | top-k | 64.4 ↑6.9 | 32.2 ↑4.6 | 91.1 ↑3.4 | 24.0 ↓0.1 | 61.2 ↑3.7 | 45.0 ↑1.0 | 66.1 ↓2.5 | 53.3 ↑0.8 | 6.6 ↓0.9 | 25.2 ↓0.6 | 46.9 ↑0.1 |
| Semantic Density | top-k | 66.2 ↑8.7 | 31.9 ↑4.3 | 91.4 ↑3.7 | 25.2 ↑1.1 | 57.2 ↓0.3 | 48.4 ↑4.4 | 71.2 ↑2.6 | 54.7 ↑2.2 | 7.5 ↑3.4 | 25.9 ↓0.2 | 48.0 ↑0.1 |
| Sensitivity | top-k | 63.2 ↑5.7 | 32.4 ↑4.8 | 91.1 ↑3.4 | 25.5 ↑1.4 | 61.3 ↑3.8 | 50.4 ↑6.4 | 70.6 ↑2.0 | 56.6 ↑4.1 | 6.8 ↑0.7 | 25.3 ↓0.4 | 48.3 ↑0.3 |
| Structural Standardization | top-k | 62.1 ↑4.6 | 31.9 ↑4.3 | 89.8 ↑2.1 | 25.3 ↑1.2 | 59.3 ↑1.8 | 45.9 ↑1.9 | 70.6 ↑2.0 | 54.5 ↑2.0 | 7.1 ↑3.0 | 27.0 ↑1.3 | 47.4 ↑0.1 |
| Style Consistency | top-k | 63.0 ↑5.5 | 32.0 ↑4.4 | 90.3 ↑2.6 | 28.7 ↑4.6 | 61.6 ↑4.1 | 50.3 ↑6.3 | 70.7 ↑2.1 | 57.8 ↑5.3 | 7.2 ↑3.1 | 25.1 ↓0.6 | 48.7 ↑0.3 |
| Topic Focus | top-k | 61.6 ↑4.1 | 30.8 ↑3.2 | 91.2 ↑3.5 | 27.3 ↑3.2 | 61.9 ↑4.4 | 50.0 ↑6.0 | 69.0 ↑0.4 | 56.3 ↑3.8 | 7.1 ↑3.0 | 24.0 ↓1.0 | 47.9 ↑0.3 |
| Overall Score | l=1 | 42.0 ↓15.5 | 23.0 ↓4.6 | 69.4 ↓18.3 | 25.7 ↑1.6 | 55.2 ↓2.3 | 31.0 ↓13.0 | 61.3 ↓7.3 | 50.3 ↓2.2 | 0.8 ↓3.3 | 25.4 ↓0.2 | 38.4 ↓6.6 |
| | l=2 | 53.7 ↓3.8 | 26.0 ↓1.6 | 83.8 ↓3.9 | 26.4 ↑2.3 | 61.8 ↑4.3 | 38.2 ↓5.8 | 63.9 ↓4.7 | 50.5 ↓2.0 | 4.3 ↑0.2 | 25.0 ↓0.70 | 43.4 ↓1.5 |
| | l=3 | 54.3 ↓3.2 | 26.2 ↓1.4 | 87.1 ↓0.6 | 24.1 | 62.0 ↑4.5 | 42.0 ↓2.0 | 68.3 ↓0.3 | 52.0 ↓0.5 | 4.7 ↓0. | 25.7 | 44.6 ↓0.3 |
| | l=4 | 60.7 ↑3.2 | 31.3 ↑3.7 | 90.6 ↑2.9 | 24.1 | 60.8 ↑3.3 | 51.3 ↑6.3 | 71.0 ↑2.4 | 57.9 ↑5.4 | 7.7 ↑3.6 | 24.2 ↓0.5 | 47.9 ↑0.2 |
| | l=5 | 66.1 ↑8.6 | 34.0 ↑6.4 | 90.7 ↑3.0 | 26.1 ↑2.0 | 59.2 ↑1.7 | 51.5 ↑6.5 | 70.7 ↑2.1 | 58.3 ↑5.8 | 7.8 ↑3.7 | 26.9 ↑1.2 | 49.1 ↑1.6 |

Table 15: The Sample-with-DataMan model (*Overall Score l=5*) improve perplexity and in-context learning (ICL) results on **larger 60B tokens**. We report the validation, test perplexity, and ICL performance of 10 downstream tasks. We highlight the best result in each column and improvement over uniform sampling with the 60B token budget.

| Selection Method | | Val Perplexity | Test Perplexity | Reading Comprehension (5 tasks) | Commonsense Reasoning (3 tasks) | World Knowledge (2 tasks) | Average (10 tasks) |
|---|---|---|---|---|---|---|---|
| Uniform | | 10.81 | 10.79 | 53.7 | 58.4 | 16.4 | 47.6 |
| Educational Value | τ = 2.0 | 9.81 ↓1.00 | 9.85 ↓0.94 | 54.2 ↑0.5 | 58.7 ↑0.3 | 16.0 ↓0.4 | 47.9 ↑0.3 |
| Overall Score | l=5 | 9.93 ↓0.88 | 9.91 ↓0.88 | 56.5 ↑2.8 | 62.9 ↑4.5 | 17.5 ↑1.1 | 50.6 ↑3.0 |

Table 16: Validation per-token perplexity per RedPajama source across **three models trained on 60B tokens**. We highlight the best result in each column. Abbreviations: CC = CommonCrawl, Wiki = Wikipedia, StackEx = StackExchange.

| Selection Method | | CC | C4 | Github | Wiki | ArXiv | StackEx | Book | Overall |
|---|---|---|---|---|---|---|---|---|---|
| Uniform | | 10.81 | **16.50** | 2.89 | 9.54 | 5.22 | 5.88 | 11.43 | 10.81 |
| Educational Value | $\tau = 2.0$ | 10.15 ↓0.66 | 12.94 ↓3.56 | 2.78 ↓0.11 | 9.11 ↓0.43 | 5.20 ↓0.02 | 5.68 ↓0.20 | 11.36 ↓0.07 | 9.81 ↓1.00 |
| Overall Score | $l=5$ | **9.94** ↓0.87 | 15.58 ↓0.92 | **2.52** ↓0.37 | **7.71** ↓1.83 | **4.53** ↓0.69 | **5.62** ↓0.26 | **10.13** ↓1.3 | **9.93** ↓0.88 |

Table 17: Test per-token perplexity per RedPajama source across **three models trained on 60B tokens**. We highlight the best result in each column. Abbreviations: CC = CommonCrawl, Wiki = Wikipedia, StackEx = StackExchange.

| Selection Method | | CC | C4 | Github | Wiki | ArXiv | StackEx | Book | Overall |
|---|---|---|---|---|---|---|---|---|---|
| Uniform | | 10.76 | 16.31 | 2.78 | 9.41 | 5.27 | 4.93 | 11.92 | 10.79 |
| Educational Value | $\tau = 2.0$ | 10.16 ↓0.60 | **12.83** ↓3.48 | 2.73 ↓0.05 | 8.99 ↓0.42 | 5.25 ↓0.02 | 4.76 ↓0.17 | 11.92 | 9.85 ↓0.94 |
| Overall Score | $l=5$ | **9.90** ↓0.86 | 15.40 ↓0.91 | **2.43** ↓0.35 | **7.61** ↓1.80 | **4.56** ↓0.71 | **4.68** ↓0.25 | **10.61** ↓1.31 | **9.91** ↓0.88 |

Table 18: The in-context learning performance for ten downstream tasks across **three models trained on 60B tokens**. We report accuracy for all tasks, except for NQ, where we report EM, and highlight the best result in each column (before rounding). Abbreviations: HellaSw. = HellaSwag, W.G. = WinoGrande.

| Selection Method | | Reading Comprehension | | | | | Commonsense Reasoning | | | World Knowledge | | |
|---|---|---|---|---|---|---|---|---|---|---|---|---|
| | | ARC-E (15) | ARC-C (15) | SciQA (2) | LogiQA (2) | BoolQ (0) | HellaSw. (6) | PIQA (6) | W.G. (15) | NQ (10) | MMLU (5) | Average |
| Uniform | | 63.0 | 32.4 | 89.1 | 22.3 | **61.6** | 49.4 | 70.7 | 55.0 | 6.4 | 26.3 | 47.6 |
| Educational Value | $\tau = 2.0$ | 64.5 ↑1.5 | 31.1 ↑1.3 | 90.9 ↑1.8 | 24.1 ↑1.8 | 60.6 ↑1.0 | 50.2 ↑0.8 | 69.9 ↑0.8 | 56.0 ↑1.0 | 7.0 ↑0.6 | 25.0 ↑1.3 | 47.9 ↑0.3 |
| Overall Score | $l=5$ | **68.6** ↑5.6 | **36.8** ↑4.4 | **91.9** ↑2.8 | **24.4** ↑2.1 | 60.6 ↑1.0 | **56.1** ↑6.7 | **72.4** ↑1.7 | **60.1** ↑5.1 | **8.6** ↑2.2 | **26.4** ↑0.1 | **50.6** ↑3.0 |

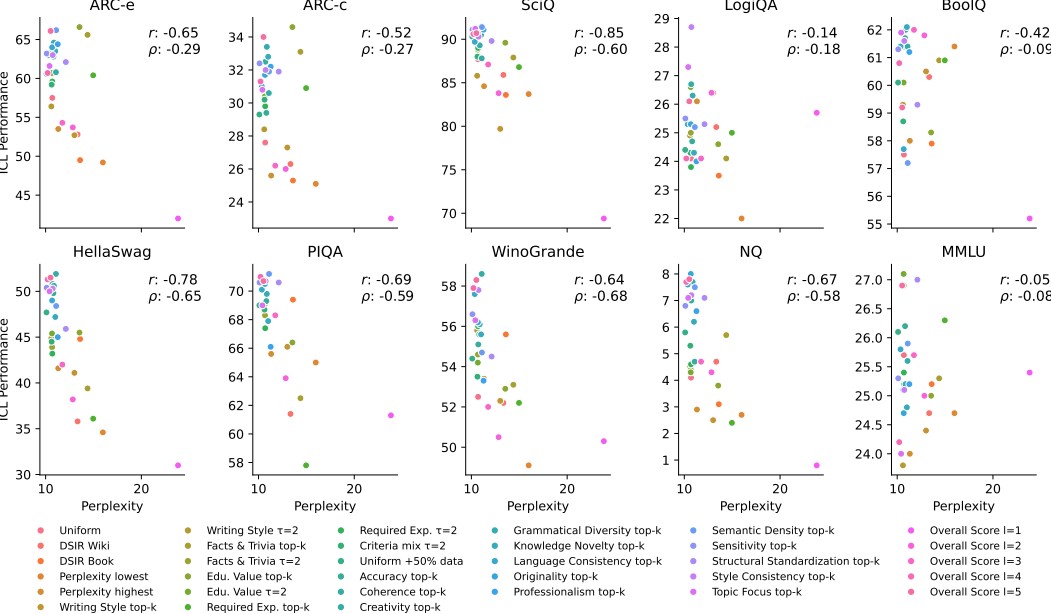

Figure 6: We plot the relationship between perplexity and in-context learning (ICL) performance for all models across 10 downstream tasks, as shown in Tables 12 and 14, along with the Pearson and Spearman correlation coefficients. The misalignment between perplexity and ICL is most pronounced in the LogiQA and MMLU tasks, we guess, possibly due to domain mismatch and ICL task complexity.

## D    INSPECTING RAW DOCUMENTS AND RATINGS

Finally, we present snippets from the raw documents of Wikipedia, Books, Stack Exchange, Github, ArXiv, CommonCrawl, and C4 subsets of DataPajama. These documents correspond to samples with quality ratings of 1, 2, 3, 4, and 5 across 14 quality criteria, as shown in Figure 3. Notably, although these samples represent just a small random snippet, they display significant quality differences. We believe it is essential to provide an unfiltered view of the training data; therefore, we have not applied any filtering to these documents. **A small number of documents contain %potentially sensitive content.**

From the visual examples, we found a notable distinction between the data rated 1 and 2 and those rated 3, 4, and 5. For instance, the score of 1 corresponds to an example like *"...83 510 l s 311 548 m 305 546 l 301 540 l 299 530 l 299 ..."*, whereas the score of 5 reflects *"...system recognizes a hierarchy of events from the measurements, not exactly in the sense of physical reality..."* However, the discrepancy between scores of 4 and 5 is not as pronounced, as seen in the example *"...have been augmented with terms that quantify the user satisfaction or the ad relevance...,"* which corresponds to a score of 4. This further supports our rationale for choosing pointwise evaluation over pairwise, as humans also find it challenging to determine superiority based on subtle differences.

In terms of domain adaptability, most of the evaluation criteria we established are semantic-focused, allowing for effective differentiation of documents within the C4 domain. However, we also observed that our criteria have some relevance in the code domain (e.g., GitHub). Specifically, code that features more detailed comments and follows structural conventions tends to receive higher scores, while disordered code is typically rated lower.

Table 19: Raw training examples selected to have 14 quality ratings from 1 to 5 within ArXiv.

| Accuracy = 1 | Accuracy = 2 | Accuracy = 3 | Accuracy = 4 | Accuracy = 5 |
|---|---|---|---|---|
| ..., y) line(1,0) 70 110 put(0,0 put(20, y) line(1,0) 70 ) line(1,0) 20 def0 put(20, y) line(1,0) 70 100 put(10,0 put(20, y) line(1,0) 70 ) line(1,0) 30 def0 put(20, y) line(1,0) 70 90 put(20,0 put(20, y) line(1,0) 70 ) ... | .... This primarily happens with comments, annotations, and imports DIFdelbegin DIFdel . DIFdel There are seven false positives that are due to textit DIFdel missing refactoring references DIFdel , DIFdelend... | ... 90 put(40,0 put(20, y) line(1,0) 70 ) line(1,0) 40 def0 put(20, y) line(1,0) 70 80 put(20,0 put(20, y) line(1,0) 70 ) line(1,0) 40 def0 put(20, y) line(1,0) 70 70 put(50,0 put(20, y) line(1,0) 70 ) line(1,0) 60 de... | ...an older PUT, then there will be two versions of the object stored in SMORE @. But the most recent will always be returned in GET requests, and the space occupied by the earlier version will eventually be reclaimed by... | ...natural . This methodology is particularly interesting for the purposes of our analysis since cite shaked1982relaxing showed that for some parameters the game has a unique subgame perfect equilibrium at... |
| **Coherence = 1** | **Coherence = 2** | **Coherence= 3** | **Coherence = 4** | **Coherence = 5** |
| ...16 82.00 540 emline 131.16 82.00 541 126.16 90.00 542 emline 126.16 90.00 543 121.16 82.00 544 emline 136.16 74.00 545 131.16 82.00 546 emline 131.16 82.00 547 126.16 74.00 548 emline 141.16 82.00 549 131.16 82.00 82.00 549 131.16 82.00 ... | ...put(10,0) line(0,1) 100 put(20,90) line(0,1) 20 put(30,50) line(0,1) 50 put(40,10) line(0,1) 70 put(50,20) line(0,1) 70 put(60,0) line(0,1) 40 put(70,30) line(0,1) 40 put(80,10) line(0,1) 50 ... | ... section* Introduction The NK (Jordan-Thiry) has been developed (see Refs cite1, cite2, cite3, cite4). The theory unifies gr al theory described by NGT ( eu nos eu gr al Theory) (see Ref. cite5) and Electrodynamics. The theory has been proposed by... | ...labels. For instance, in the graphical model in Fig. ref F:models , a rectangle named emph TopicFigure is defined, and it is referred to by the node emph Topic . A diagram label named emph TopicName is also defined. Such graphical... | ...in the proposed neural program models xspace. A comprehensive understanding of the extent of generalizability of neural program models xspace would help developers to know when to use data-driven approaches and when to resort... |
| **Creativity = 1** | **Creativity = 2** | **Creativity = 3** | **Creativity = 4** | **Creativity = 5** |
| .../88 21:16 7445.386 −0.053 0.028 500 2.93 629 1.2CA punkta 13/10/88 22:12 7448.425 −0.077 0.066 500 3.63 996 1.2CA punkt 16/10/88 23:47 7451.491 −0.048 0.030 500 1.75 725 1.2CA punkt 27/06/89 03:06 7704.630 0.028 0.012 500 2.07 1036 1.2CA punkt 02/08/9 34:03... | ...) (0.7,0.749) (1,0.768) ; addplot[style= ppurple,mark=*, mark options= scale=1.5,fill=white ] coordinates (0.01, 0.476) (0.1,0.666) (0.4,0.726) (0.7,0.757) (1,0.769) ; addplot[style= rred,mark=square*, mark options= scale=1.5,fill=white ] coordinates ... | ...a reasonable improvement compared with traditional methods such as matrix factorization. The rating-based methods suffer from a key limitation, which is the sparsity of the data. Specifically, when collecting data from real-world platforms, the... | ... section Introduction Stereo algorithms benefit enormously from benchmarks . They provide quantitative evaluation to encourage competition and track progress. Despite great progress over the past years, many challenges still remain unsolved, such as... | ...adiest attention. He maintained his interest in Darwin and Darwinism; in 1875 he was reading in modern physics as well. ldots In May 1875 James read and reviewed a book called sl The Unseen Universe by physicist and mathematician Peter Guthrie Tait ldots and physicist and... |
| **Grammatical Diversity = 1** | **Grammatical Diversity = 2** | **Grammatical Diversity = 3** | **Grammatical Diversity = 4** | **Grammatical Diversity = 5** |
| ...83 510 l s 311 548 m 305 546 l 301 540 l 299 530 l 299 524 l 301 514 l 305 508 l 311 506 l 316 506 l 322 508 l 326 514 l 328 524 l 328 530 l 326 540 l 322 546 l 316 548 l 311 548 l s ta ta 250 937 m 244 935 l 240 929 l 238 919 l 238 913 l 240 903 l 244 897 l 250 895 l 255 895 l 261 897 l 265 903 l 2... | ...ord) line(1,0) 60 70 put(50,0 put(40, ycoord) line(1,0) 60 ) line(1,0) 40 def0 put(40, ycoord) line(1,0) 60 60 put(60,0 put(40, ycoord) line(1,0) 60 ) line(1,0) 40 def0 put(40, ycoord) line(1,0) 60 50 put(30,0 put(40, ycoord) line(1,0) 60 ) line(1,0) line(1,0) line(1,0) line(1,0) 50 def0 pu... | ...consider the human brain learning process again. Say the information is wrong in some of the sensory stimulation. A child learned an animal looks just like a dog but having the sound of the cat from the manipulated movies and this kid has never learned the dog and cat in a.... | ...to fill a questionnaire with Likert scale and open-ended questions and describe any problems they experienced. subsection Results subsubsection Round 1 Fig. ref fig:teaser (top) shows the handover for all the 10 objects in the set textit Household-A by participants... | ...-asymm end figure* section Conclusions label sec:conc subsection Summary The zonal-mean surface air temperature response to abrupt coo/ increases of 2, 4, 8, or 16 ( times ) in 3,000-yr simulations performed in a low-resolution version of the CESM1 GCM exhibit an in... |
| **Knowledge Novelty = 1** | **Knowledge Novelty = 2** | **Knowledge Novelty = 3** | **Knowledge Novelty = 4** | **Knowledge Novelty = 5** |
| ...− (322.88,226.15) − (311.38,215.57) − cycle ; draw (359.2,216.36) − (359.42,256.13) − (322.88,226.15) ; draw [fill= rgb, 255:red, 155; green, 155; blue, 155 ,fill opacity=1 ] (359.2,205.79) − (370.7,216.36) − (359.2,226.93) − (347.7,216.36) − cycle... | ...gpsetdashtype gp dt solid gpsetlinewidth 2.00 draw[gp path] (2.391,5.033) − (2.256,5.033); node[gp node right,font= fontsize 8.0pt 9.6pt selectfont ] at (2.164,5.033) 1 ; gpcolor rgb color= 0.702,0.702,0.702 gpsetlinetype gp lt axes ... | ...one bond kind of structure is useless for our aim, being fully unstable. Let us go directly to the interesting one. In Fig. ref FC1 , we show how this lattice is made and its textit unitary structure , made up by the highlighted yellow bonds plus the... | ...have been augmented with terms that quantify the user satisfaction or the ad relevance. Bids receive adaptive discounts in order to deal with situations where the perfect information assumption is unrealistic unrealistic situations... | ...System recognizes a hierarchy of events from the measurements, not exactly in the sense of physical reality -- if a creature never measured/saw a black swan, it does not mean there aren't any -- but in the sense of a manually ... |

Table 20: Raw training examples selected to have 14 quality ratings from 1 to 5 within ArXiv.

| Language Consistency = 1 | Language Consistency = 2 | Language Consistency = 3 | Language Consistency = 4 | Language Consistency = 5 |
|---|---|---|---|---|
| ...) 60 ) line(1,0) 20 def0 put(40, ycoord) line(1,0) 60 90 put(10,0 put(40, ycoord) line(1,0) 60 ) line(1,0) 30 def0 put(40, ycoord) line(1,0) 60 80 put(30,0 put(40, ycoord) line(1,0) 60 ) line(1,0) 60 def0 put(40, ycoord) line(1,0) 60 70 put(20,0 put(40, ycoord) ... | ...-courses item[fix:] textbf Siis pidi igaüks igaüks igaüks textbf end vabaainetele textbf registreerima . item[gloss:]then had-to everyone oneself to-free-courses register end enumerate end enumerate Results also show two main downsides of... | ...baseados no UUCP. Rick Muething, criador do ARDOP, Muething realizou uma análise das opções de modem footnote Muething Modem neste contexto significa solução de transmissão digital que provê um canal de comunicação sem perda Muething... | ... section Introduction Biological informations about genes and proteins are stored into biological ontologies cite cannataro2013data cite Guzzi2012 such as Gene Ontology (GO). GO has gained a wide diffusion in bioinformatics and computational biology... | ... section Introduction Nowadays, travelers use various online services and recommender systems to plan their trips. Recommender systems allow users to deal with data overload and make better decisions in a personalized way... |
| **Originality = 1** | **Originality = 2** | **Originality= 3** | **Originality = 4** | **Originality = 5** |
| ...circle* 0.000001 put(65.76,-228.40) circle* 0.000001 put(66.47,-228.40) circle* 0.000001 put(67.18,-228.40) circle* 0.000001 put(67.88,-228.40) circle* 0.000001 put(68.59,-229.10) circle* 0.000001 put(69.30,-229.10) circle* 0.000001 put(70.00,-229.10) ... | ...09886278768 0.7388 3210.52550284399 0.7212 3219.57683735924 0.7212 3228.5987543788 0.742 3238.36785145622 0.7676 3249.01195892414 0.77 3259.27909703235 0.754... | ...6 ,draw opacity=0.85 ][line width=0.75] (10.93,-3.29) .. controls (6.95,-1.4) and (3.31,-0.3) .. (0,0) .. controls (3.31,0.3) and (6.95,1.4) .. (10.93,3.29) ; draw [color= rgb, 255:red, 126; green, 126; blue, 126... | ...0.86725 105 0.8678 106 0.86801 107 0.86903 108 0.8683 109 0.86791 110 0.86793 111 0.86888 112 0.8689 113 0.86888 114 0.86858 115 0.86782 116 0.8676 117 0.86748 118 0.86765 119 0.86805 120 0.86806 121 0.86851 122 0.86844 123 0.86791... | ... section Introduction label introduction IEEEPARstart T ime series forecasting, which consists of analyzing historical signals patterns to predict future outcomes, is an important problem with scientific, business, and industrial... |
| **Professionalism = 1** | **Professionalism = 2** | **Professionalism = 3** | **Professionalism = 4** | **Professionalism = 5** |
| ...472) (202,472) (202,472) (203,472) (203,472) (204,472) (204,472) (205,472) (205,472) (205,472) (206,472) (206,472) (207,472)(207,472) (208,472) (208,472) (208,472) (209,472) (209,472) (210,472) (210,472) (210,472) (211,472) (211,472) (212,472) (212,472) (213,472) (213,472) (213,472) (214,472) (214,472) (215,472) (215,472) (215,472) (215,472) (216,472... | ...(30,0 put(20, y) line(1,0) 70 ) line(1,0) 30 def0 put(20, y) line(1,0) 70 80 put(20,0 put(20, y) line(1,0) 70 ) line(1,0) 30 line(1,0) 30 line(1,0) 30 line(1,0) 30 def0 put(20, y) line(1,0) 70 70 put(40,0 put(20, y) line(1,0) 70 ) line(1,0) 40 def0 put(20, y) line(1,0) 70 60 put(70,0 put(20, y) l... | ...0.26) .. controls (270.11,370.98) and (269.53,371.57) .. (268.81,371.57) .. controls (268.09,371.57) and (267.5,370.98) .. (267.5,370.26) - cycle ; draw (270.11,370.26) .. controls (270.11,369.53) and (270.7,368.95) .. (271.42,368.95) .. controls (272.14,368.95) and (272.72,369.53) .. (272.72,37... | ...istics network was studied in which shippers collaborate and bundle their shipment requests to negotiate better rates with a common carrier and cost-allocation mechanisms were proposed to ensure the sustainability of the collaboration. In cite gansterer2021prisoners the prisoners' dilemma was appl... | ... label appli::results Figure ref graph::plots represents the three transition intensities (Part A), the three cumulative incidences (Part B) and the estimated mean ISAACS trajectories (Part C) for men with a low level of education (with no primary school diploma), in each class. The first class i... |
| **Semantic Density = 1** | **Semantic Density = 2** | **Semantic Density = 3** | **Semantic Density = 4** | **Semantic Density = 5** |
| ...96)[1](2.447, 0.10694168320452796) (2.448, 0.106804112273327702) [1] (2.449, 0.10666669262164181) [1] (2.45, 0.10652942412709385) [1] (2.451, 0.106392306667139) [1] (2.452, 0.106255340119293... | ...1,0) 70 ) [1]line(1,0) 30 def0 put(20, y) line(1,0) 70 10 put(40,0 put(20, y) line(1,0) 70 ) line(1,0) 50 def0 put(20, y) line(1,0) 70 0 put(10,0 put(20, y) line(1,0) 70 ) line(1,0) 70 put(0,60)... | ...151042) psline[linecolor=black, linewidth=0.06] (14.4,13.151043) (14.4,11.951042) (14.4,11.951042) psline[linecolor=black, linewidth=0.06] (15.2,13.151043) (15.2,11.951042) (15.2,11.951042) psline... | ...c, fill=c] (7.62637,2.64783) rectangle (7.8022,2.75217); definecolor c rgb 0.116419,0.686966,0.7029; draw [color=c, fill=c] (7.8022,2.64783) rectangle (7.97802,2.75217)... | ... section INTRODUCTION In recent years we have seen a dramatic increase in interest within the area of autonomous transportation and its associated research.... |
| **Sensitivity = 1** | **Sensitivity = 2** | **Sensitivity = 3** | **Sensitivity = 4** | **Sensitivity = 5** |
| ...444) rule[-0.500pt] 1.000pt 1.566pt 1.000pt 1.566pt 1.000pt 1.566pt put(361,438) rule[-0.500pt] 1.000pt 1.566pt put(362,431) rule[-0.500pt] 1.000pt 1.566pt put(363,425) rule[-0.500pt] 1.000pt ... | ....2, 274.09852) (17.2, 271.1908661538462) (98.8, 305.2704876923077) (98.8, 305.2704876923077) (98.8, 305.2704876923077) (98.8, 305.2704876923077) (98.8, 305.2704876923077) (47.199999999999996, 282.68546) (48.0, 293.9522630769231) (38.0, 268.4655492307692)...... | ...1) (0:1) - (180:1); draw[blue] (-60:1) - (180:1) - (60:1) - (-60:1) (60:1) - (-60:1) (60:1) - (-60:1) ; draw[blue] (0:1) - (-60:1) - (-120:1) (180:1) - (-120:1) - (60:1); draw[red] (150:0.87) - (60:1) - (-30:0.87) (-90:0.87) - (180:1) - (90:0.87) | ...and the thermodynamics / statistical mechanics of the classical scale becomes particularly delicate cite physrep . Regarding the methods of calculation, among the electronic structure techniques, Density Functional, ... | ...Diverse individuals age at different rates and display variable susceptibilities to tissue aging, functional decline and aging-related diseases. Centenarians, exemplifying extreme longevity, serve as models for healthy aging. The field of human... |

Table 21: Raw training examples selected to have 14 quality ratings from 1 to 5 within ArXiv.

| Structural Standardization = 1 | Structural Standardization = 2 | Structural Standardization = 3 | Structural Standardization = 4 | Structural Standardization = 5 |
|---|---|---|---|---|
| ...bWa90I7 rzAIqI3 UE1UJG7tL tUXzw4KQNETvXz qWaujEMenYlNIzLGx gB3AuJ86VS6RcPJ 8OXWw8imtc KZEzHop84G1 gSAs0PCowMI2f LKT dD60yn Hg7lkNF jJLqOoQ vfkfZBN G3olDgC n9hyUh5 VSP5z61 qvQwceUd VJJsBvXD qvQwceUd G4ELHQHIa5... | ...static setting. The algorithms that do not learn (Sparrow and PoT) do not degrade for the same reason discussed above. iffalse begin figure includegraphics [width=0.45 textwidth, height=0.15 textheight] figures.pdf ... | ...Gruhl2004, Hill2006, Iribarren2009, Java2006, Leskovec2007a, Leskovec2006, Strang1998,Wan2007 . Two fundamental types of models of information diffusion used in the literature are cascade models and threshold... | ...last years bigger devices like blood refrigeration units, CT scan systems and X-ray systems are connected to the Internet, in order to check remotely their operational state and make whatever adjustment is needed (e.g., lower the blood unit inside te... | ...attack. color black Such a figure summarizes what happens with attacked STFT and MF spectrograms: the difference between the adversarial and legitimate spectrograms is imperceptible to the human visual system. color black This aspect is very imp... |

| Style Consistency = 1 | Style Consistency = 2 | Style Consistency = 3 | Style Consistency = 4 | Style Consistency = 5 |
|---|---|---|---|---|
| ...PY o PY n id char C PY p char C PY pchar C PY p PY p PY p PY o PY n deathsChild PY p PY o PY n newInfectiousChild C PY p PY p PY p C PY p PY p PY p C PY p PY p PY p PY p PY p PY p PY p PY p PY p PY o PY n agingChild PY p PY o PY n id char C PY p PY p PY p PY o PY n v ch... | ...) line(1,0) 70 80 put(40,0 put(20, y) line(1,0) 70 ) line(1,0) 50 def0 put(20, y) put(80,0 put(20, y) line(1,0) 70 ) line(1,0) 20 def0 put(20, y) put(20, y) put(20, y) line(1,0) 70 60 put(20,0 put(20, y) line(1,0) 70 ) lin... | ...0 put(60,0 put(20, y) line(1,0) 70 ) line(1,0) 40 def0 put(20, y) put(10,0 put(20, y) line(1,0) 70 ) line(1,0) 70 put(20, y) put(20, y) def0 put(20, y) line(1,0) 70 20 put(50,0 put(20, y) line(1,0) 70 ) line(1,0) 60 def0 ... | ...) (−1,1) pspolygon [fillstyle=solid, fillcolor=NavyBlue] (0,0) (0,−6) (1,−6) (1,−1) pspolygon [fillstyle=solid, fillcolor=NavyBlue] (1,1) (6,1) (6,0) ( 2,0) pspolygon [fillstyle=solid, fillcolor=NavyBlue] (−1,1) (−6,1) (−6,0) (−2,0) pspolygon [fillstyle=solid, fillco... | ... section Introduction tmf (TMNF, or TMF) is a 3D racing game that was released in 2008 by video game developer Nadeo. It is part of the racing game series TrackMania. It was designed for the Electronic Sports World Cup, which is a yearly internati... |

| Topic Focus = 1 | Topic Focus = 2 | Topic Focus = 3 | Topic Focus = 4 | Topic Focus = 5 |
|---|---|---|---|---|
| ...55 0.2 12490313.2925055 0.206666666666667 13225057.5307381 0.206666666666667 13225057.5307381 0.213333333333333 13231816.8440602 0.213333333333333 13231816.8440602 0.22 13262462.5634116 0.22 13262462.5634116 0.226666666666667 13464096... | ... put(20, y) line(1,0) 70 10 put(50,0 put(20, y) line(1,0) 70 ) line(1,0) 50 def0 put(20, y) put(20, y) put(20, y) line(1,0) 70 0 put(30,0 put(20, y) line(1,0) 70 ) line(1,0) 40 put(0,70) line(0,1) 40 put(10,50) line(0,1) 50 put(20,90) line... | ...403) − (2.5724992775571893, 1.417611987501176); draw[line width=1pt, color=qqqqff] (2.5724992775571893, 1.417611987501176) − (2.579999266831962, 1.4192818792042017); draw[line width=1pt, color=qqqqff] (2.579999266831962, 1.4192818792042017) − (2.5... | ...mm] (s3) node[node1, right = 0.25cm and 0.25cm of s2, line width=0.1mm] (s4) node[node1, left = 0.2cm and 0.25cm of s3, line width=0.1mm] (s5) node[node2, right = 0.25cm and 0.25cm of s4, line width=0.1mm] (s6) node[node2, right = 0.25cm and 0.25cm of s4, line width=0.1mm] (s6) ... | ...between warehouses which are almost entirely obstacle-free, and therefore only requiring low precision navigation. By ignoring this heterogeneity, mobile robots are forced to make worst-case decisions to ensure their safety. For example, for the abov... |

| Overall Score = 1 | Overall Score = 2 | Overall Score = 3 | Overall Score = 4 | Overall Score = 5 |
|---|---|---|---|---|
| ...472) (202,472) (202,472) (202,472) (202,472) (202,472) (203,472) (203,472) (204,472) (204,472) (205,472) (205,472) (205,472) (206,472) (206,472) (207,472) (207,472) (208,472) (208,472) (208,472) (209,472) (209,472) (210,472) (210,472) (210,472) (211,472) (211,472) (212,472) (212,472) (213,472) (213,472) (21... | ...(30,0 put(20, y) line(1,0) 70 ) line(1,0) 30 def0 put(20, y) line(1,0) line(1,0) 70 30 def0 put(20, y) line(1,0) 70 ) line(1,0) 30 def0 put(20, y) line(1,0) 70 70 put(40,0 put(20, y) line(1,0) 70 ) line(1,0) 40 def0 put(2... | ...0.26) .. controls (270.11,370.98) and (269.53,371.57) .. (268.81,371.57) .. controls (268.09,371.57) and (267.5,370.98) .. (267.5,370.26) − cycle ; draw (270.11,370.26) .. controls (270.11,369.53) and (270.7,368.95) .. (271.42,368.95) .. controls... | ...istics network was studied in which shippers collaborate and bundle their shipment requests to negotiate better rates with a common carrier and cost-allocation mechanisms were proposed to ensure the sustainability of the collaboration. In cite ganst... | ... label appli::results Figure ref graph::plots represents the three transition intensities (Part A), the three cumulative incidences (Part B) and the estimated mean ISAACS trajectories (Part C) for men with a low level of education (with no primary... |

Table 22: Raw training examples selected to have 14 quality ratings from 1 to 5 within Book.

| Accuracy = 1 | Accuracy = 2 | Accuracy = 3 | Accuracy = 4 | Accuracy = 5 |
|---|---|---|---|---|
| ...IA GREEK SMALL LETTER PI GREEK SMALL LETTER EPSILON GREEK SMALL LETTER RHO GREEK SMALL LETTER IOTA WITH VARIA GREEK SMALL LETTER TAU GREEK SMALL LETTER OMICRON WITH VARIA GREEK SMALL LETTER NU GREEK SMALL LETTER PI GREEK... | ...n?' Well upon dis, de Pharisees picked up der frails and cut away right by him, and as dey passed by him he felt sich a queer pain in his head as if somebody had gi'en him a lamentable hard thump wud a hammer, dat knocked him down as flat as a floun... | ...ountain 34. 26 A London Season 35. 27 Turning Forty 36. 28 The Classical Style 37. Acknowledgments 38. Notes 39. Illustration Credits 40. A Note About the Author 1. i 2. ii 3. iii 4. iv 5. v 6. vi 7. vii 8. viii 9. ... | ...Twenty years ago Jerry Wilson was known as the cattle king of the Platte River. His cattle roamed for hundreds of miles up and down the main river and all its tributaries, and, as the cowboys used to say, no one man could count them even if they was ... | ..., church steeples, and a three-story brick opera house. Instead, they found a half-burnt town ruined by a devastating blaze that lay waste twenty blocks a few months earlier. Holcomb had taken leave of a proud, 250-year-old city touted as one of the ... |

| Coherence = 1 | Coherence = 2 | Coherence= 3 | Coherence = 4 | Coherence = 5 |
|---|---|---|---|---|
| .... 14. 15. 16. 17. 18. 19. 20. 21. 22. 23. 24. 25. 26. 27. 28. 29. 30. 31. 32. 33. 34. 35. 36. 37. 38. 39. 40. 41. 42. 43. 44. 45. 46. 47. 48. 49. 50. 51. 52. 53. 54. 55. 56. 57. 58. 59. 60. 61. 62. 63. 64. 65. 66. 67. 68. 69. 70. 71. 72. 73. 74. 75. ... | ...de Montès allaient et venaient, et quelque chose d'impuissant et d'indigné dans ce qu'on pouvait apercevoir du visage au-dessus de la serviette que continuait à presser la main décharnée, et plus tard encore - à ce moment ce devait être la seule, l'u... | ...met Lawrence Isis, a half-Irish, half-Egyptian Copt software engineer at a campus concert--Celtic folk music, Amanda had gone on a lark, a friend's urging. The chemistry had been instant, despite the fact that Larry resembled Woody Allen with dark hai... | ...to lecture. Billy continued meekly to listen. The old man knew that sex was at the bottom of it. He saw that Billy had a sensuous nature which needed to be curbed. Billy was giving up Rose to seek out Queen MyrdemInggala--yes, he knew Billy's desires... | ...Soviet bloc diplomatic and trade missions heavily staffed by intelligence operatives. The age-old art of espionage had not faded away, despite the world entering the apparently softer, gentler phase of the Cold War associated with the détente between... |

| Creativity = 1 | Creativity = 2 | Creativity = 3 | Creativity = 4 | Creativity = 5 |
|---|---|---|---|---|
| ...47-48, 150-53, , , , , 188-89, , , , 277-78, 282-312, , , , , , , , , , 369-70, 375-77 law, 15-81, 147-82 147-82. 147-82. 147-82.. See also natural law legality, , 154-55, , , legal status, , Leibniz, G. W., liberalism, xv, , , , , , , , 273-75, , 273-75 liberty, , 99-10... | ...and Fletcher v. Peck (1810), because the states were angry over the Court's assertion of power to strike down state laws; and both Virginia and Kentucky had made dark threats foreshadowing a constitutional crisis in 1823, according to Robertson. In d... | ...qu'il n'oubliera pas. Elle est à l'origine de son pragmatisme en politique. La question dans un monde ensorcelé n'est pas de savoir qui a raison, qui va le plus droit, mais qui est à la mesure du Grand Trompeur, quelle action sera assez souple, assez... | ...rage and near pornographic intrusion into private lives. Detective Sergeant Norman Pilcher craved fame and recognition. Backing them up were a bunch of minor dyspeptic personalities, self-styled worthies, suburban officials and legal types. This was ... | ...ank into his chair, a sullen pout on his lips. And don't you think about going behind my back, I warned husband number two as I sat at Daniel's right. You will not be arranging a handy accident, is that understood? Well, my lover, I've found I... |

| Grammatical Diversity = 1 | Grammatical Diversity = 2 | Grammatical Diversity = 3 | Grammatical Diversity = 4 | Grammatical Diversity = 5 |
|---|---|---|---|---|
| ...27. 28. 29. 30. 31. 32. 33. 34. 35. 36. 37. 38. 39. 40. 41. 42. 43. 44. 45. 46. 47. 48. 49. 50. 51. 52. 53. 54. 55. 56. 57. 58. 59. 60. 61. 62. 63. 64. 65. 66. 67. 68. 69. 70. 71. 72. 73. 74. 75. 76. 77. 78. 79. 80. 81. 82. 83. 84. ... | ...6), 306(n3) Ross, Leonard, , 300(n53) Rosten, Leo, Rothweiler, Monika, (n10, n11) Ruhlen, Merritt, 312(n23) Rules, , , , , , , , . See also Combinatorial systems; Heads; Symbols and symbol processing; Word structure Rumelhart, David, , , 103-11... | ...dummy he was fighting looked up, and the old corkscrew right went over and the dummy started trilling to the daisies. And the baseball games in the old days of Spike Shannon, Mike Donlin, Fred Tenney, Jimmy Collins, Cy Young, Pat Dougherty, Fielder J... | ...a normal life, whatever that is. I am now watching many of my friends enter their second marriages and, while I am more than happy to have waited, to have missed that first wave of divorces, I do hope to find someone special when the time is right. T... | ...led the canon of Dead White European Males, the Anglo-American core in English departments was supplanted or minimized--and along with it a consciousness of the ancient and medieval etymology of English words, a complex lineage that is wittily evoked ... |

| Knowledge Novelty = 1 | Knowledge Novelty = 2 | Knowledge Novelty = 3 | Knowledge Novelty = 4 | Knowledge Novelty = 5 |
|---|---|---|---|---|
| ...I joined them for a while and picked up a little, you know, cab fare. Then I forced everybody, including the conductor, to get in the last car, and I pulled the pin and left them back in the tunnel. Sometimes that's the only way you can get a seat. A... | ...of the affair, preferring a pretty face and poverty. Stupid devil, to throw away such a birthright! Lucky dog, who is to be his successor? Let the rogue win the race. I am so tired of the dodges, the twists, the aliases, the lurkings, that I will put... | ... La faveur d'obtenir un peu ! Devenons attentifs à ces âmes choisies Que l'on goûte à travers leurs corps ; Contraignons, en souffrant, l'altière fantaisie, -- Aimer moins est si fort encor ! Il n'est pas, pour nouer une divine attache, Que ces ... | ...that would train people to be active and useful citizens in the modern technological society. In short, popular culture movements saw culture as the means by which the individual's relation to society might be ameliorated. Many of these foundatio... | ...IPv4, proposed originally in the mid-1990s. When would IPv6 run out? If you were to divide the total number of possible addresses within a 128-bit space by 7 billion people, it would be able to theoretically allocate approximately 5 × 1028 addresses ... |

Table 23: Raw training examples selected to have 14 quality ratings from 1 to 5 within Book.

| Language Consistency = 1 | Language Consistency = 2 | Language Consistency = 3 | Language Consistency = 4 | Language Consistency = 5 |
|---|---|---|---|---|
| ...totta, mutta hän vaistosi, että kuski pettyisi, jos hän paljastaisi olevansa vailla uskoa. Gloria ei ollut koskaan voinut käsittää, että joku halusi käyttää kidutuksen ja kuoleman välinettä koristeena. Yhtä hyvin olisi voinut kantaa hirttosilmukkaa ... | ..., ellingtonien, très chromatique, refrain d'un autre monde, voluptueux, onirique, lui tient compagnie sur le chemin du retour. De l'art ou pas, il n'en sait rien, et eux, qu'en savent-ils, si l'ombre de la croix fait loi ? La sensation d'une répéti... | ...ille. Et pendant longtemps je m'arrêtai à cette conclusion, qu'à moins d'avoir seize poches, chacune avec sa pierre, je n'arriverais jamais au but que je m'étais proposé, à moins d'un hasard extraordinaire. Et s'il était concevable que je double le n... | .... --No, necesito tu ayuda. Soy nueva aquí, ¿recuerdas? Ante su tacto, sentí un tibio escalofrío que no asociaba con la amistad. ¿Y por qué no? Sus mejillas se sonrojaron, sus ojos centelleaban; bajo los focos y el cielo azul verdoso de un crepúsculo... | ...epitomise Athens' more general shift from orality to the increasing use of writing. This development unfortunately also locked in Solon's less attractive legislation, some of it termed 'peculiar' by Plutarch, the epidikasi , for example, a procedu... |

| Originality = 1 | Originality = 2 | Originality= 3 | Originality = 4 | Originality = 5 |
|---|---|---|---|---|
| ..., ¬, , ¬, ¬, −8n1, 188n7, , , , , ¬, ¬, ¬, 259n33 acknowledgment , ¬, , , , 61n10, , ¬, ¬, , , , ¬, , , , , , , , action , , 115n21, ¬, 45n2, , , ¬, , , , ,115n21 , , , , , , 115n21, ¬, , ¬, 128n23, ¬, , , , ,115n21 , ¬, , , , ¬, , , , ¬, ¬, , , , ¬, ¬, , 115n21, ¬, Agee,... | ...2013, B.S. 9 januari 2014, inwerkingtreding: 30 april 2014 (art. 2 K.B. 4 april 2014, B.S. 29 april 2014) 1[Hoofdstuk 5 − Bijzondere bepalingen]1 **1**. Opschrift ingevoegd bij art. 2 wet 15 december 2013, B.S. 9 januari 2014, inwerkingtreding: 3... | ...¬, ¬, ¬, − elections Mecklenburg-Schwerin state assembly (1927), − electoral performance 1929 state parliaments, , − Danzig, Lippe state elections (1933), Presidential (1932), − Reichstag (1928), − Reichstag (1930), − Reichstag (1932), − Re... | ...My heart hammers in the rib cage much longer than it should, but I am unmolested, for now. I walk calmly away from the bus and change clothes as soon as I find a Salvation Army bin. I select a man's apparel, compress my hair into a woolen hat, and di... | ...auf aux niveaux scolaires les plus élevés où l'effet de sursélection tend à neutraliser les différences de trajectoire) que, premièrement, on fait moins appel à une compétence stricte et strictement contrôlable et davantage à une sorte de familiarité... |

| Professionalism = 1 | Professionalism = 2 | Professionalism = 3 | Professionalism = 4 | Professionalism = 5 |
|---|---|---|---|---|
| ...ingered in each other's arms after a conjugal visit in a bus station toilet stall. They call Jarvis an 'Uncle Tom of Finland,' said Gentry. Delicious asked, What's that mean? Gentry shrugged and shook his head. Something about too many muscle... | ...non le faceva venire l'affanno, un affanno insopportabile, per cui avrebbe voluto balzare in piedi smaniosa; ma non poteva. Il cuore, il cuore le batteva precipitoso come il galoppo d'un cavallo scappato. Ah, il cuore, il cuore non le reggeva più, fo... | ...to Heaven directly, where you may join her at your convenience. Your most humble and obedient servant , The Prince of Evil Zane stared at the message, absorbing its every implication. Suddenly it burst into flame in his hand. He dropped it,... | ... For the purpose of the object on which we now enter, we have consulted a great mass of documents, and have had recourse to the personal experience of a gentleman who has made this kind of research his business. In every statement we make, we shall ... | ...ates the problem of freedom in Adam's choice of himself as a whole in the world. He can agree with Leibniz that another gesture of Adam, implying another Adam, implies another world, but only in the sense that another face of the world will corres... |

| Semantic Density = 1 | Semantic Density = 2 | Semantic Density = 3 | Semantic Density = 4 | Semantic Density = 5 |
|---|---|---|---|---|
| .... 36. 37. 38. 39. 40. 41. 42. 43. 44. 45. 46. 47. 48. 49. 50. 51. 52. 53. 54. 55. 56. 57. 58. 59. 60. 61. 62. 63. 64. 65. 66. 67. 68. 69. 70. 71. 72. 73. 74. 75. 76. 77. 78. 79. 80. 81. 82. 83. 84. 85. 86. 87. 88. 89. 90. 91. 92. 93. 94. 95. 96. ... | ...ancholy bait, 60. sold for more than an ox, 734.734 to fry, other, 772, 790, 772, 790,772,734 790,772, 790. what cat 's averse to, 381,772, 790,772, 790. with the worm, man may may, 141. Fishes gnawed upon, men that, 96. 772, 790 live in the sea, live in the sea, ... | ...adlen, Rumist datzu unnd schreybst in alle welt, du kundist allis auszwendig und geprauchst keiner pucher. Du dorfftist des Rumisz nit, man siht es mehr dan du gleubst, das du allis an bucher schreybst und lerist: wen du die augen so fleyssig... | ...and resources, but as sinners who were lost. To Him, Ulaf was no different than Conor except that Conor had accepted the sacrifice made for him. You're still here. Conor looked up. Haldor stood over him, holding a dripping battle-ax, his face, ha... | ...Protestant king in Ireland, a Protestant parliament, a Protestant hierarchy, Protestant electors and government, the benches of justice, the army and revenue, through all their branches and details, Protestant... |

| Sensitivity = 1 | Sensitivity = 2 | Sensitivity = 3 | Sensitivity = 4 | Sensitivity = 5 |
|---|---|---|---|---|
| ...and incorrect ways to be born. And without question the worst and most disgusting deformity to be born with was gayness. Which was why EJP always deleted his browser history and hated The Thought and hated gay people but couldn't hold it against them... | ...at her; he stared at the fire. The soft light did not touch his eyes, Not even Arthur? The air hummed with Power. This was no place for lies. She touched his hand, and when he finally turned, she gazed into his eyes. Especially not Arthur. He c... | ...on the face and cried, Traditore! Father Paciere was cowering back, powerless to stop it. The soldati finished their part of the ceremony. Babe, red cape flowing in the drafty place, held up her hand. Giorgio rushed forward. He gave her two... | ...not like it. 'But it wasn't all that much. Just some smoky walls and busted glass.' Hardin came back to his main problem. 'The foreigner. Did you ever meet him?' 'No. Biggie set up a meeting for tonight in case he had something to trade. That's why... | ...6 Richard swiftly, but silently, raised the sword before himself in preparation for an attack--what kind of attack he wasn't sure, but he fully intended to be ready. He touched the cold steel of the blade to his sweat-slick forehead. He spoke the wo... |

Table 24: Raw training examples selected to have 14 quality ratings from 1 to 5 within Book.

| Structural Standardization = 1 | Structural Standardization = 2 | Structural Standardization = 3 | Structural Standardization = 4 | Structural Standardization = 5 |
|---|---|---|---|---|
| ...first lieutenant, he's got a neck like a Swiss steer, I once tried to do splits with a beauty in the Catholic House and gave myself a hernia, which isn't so bad for a man, a man makes anything look good, but when a maiden wears a truss and a lovesick... | ...bewail, boohoo, grieve, lament, regret, snivel, squall 7 blubber, deplore, trickle, ululate, whimper 8 complain 9 break down, make a fuss, percolate, shed tears for: 4 pity 6 bemoan, lament (for): 4 feel over: 6 bewail, regret, repent ready to... | ...of scotch two nights before. Tempting as dropping by the liquor store may have been, he'd decided he'd rather eat paint thinner than answer pointed questions from the locals about Susan. There had been some changes in the Martin household over the lll... | ... Penelope had listened silently, like a girl in a dream. Now she patted Mrs. Fairweather's soft old hand affectionately. It sounds like a storybook, she said gaily. You must come and see Doris. She is such a darling sister. I wouldn't have had trash the patron... | ...-53. Manna SK, Mukhopadhyay A, Aggarwal BB. Resveratrol suppresses TNF-induced activation of nuclear transcription factors NF-kappa B, activator protein-1, and apoptosis: potential role of reactive oxygen intermediates and lipid peroxidation... |

| Style Consistency = 1 | Style Consistency = 2 | Style Consistency = 3 | Style Consistency = 4 | Style Consistency = 5 |
|---|---|---|---|---|
| ..., , , , , , Principe, Production, mode of capitalist, , , plantation, Proletarian, , , , , , , , , Proletariat, , , Protein, , , , , , , Prussian, Pudding, , , , , , , , , , , Puerto Ricans, xvi, xvii Puerto Rico, xv, xvi, xviii, xix, xx, xxi,... | ...my buggy wus torn to pieces, an' I wus knocked high in de air. De first time dey run into me dey killed my hoss. De third time dey paralized my arm and busted the linin' o' my stomach. I learned to read an' write since de surrender by studying in s... | ...l'extrême du sentir-créateur. [...] Aller au bout des puissances du seul – à deux. Reconstituer l'amour dans l'au-delà des états privés et connus. » La prison du miroir enfin devenue inutile, c'était délivrer cette voix peut-être entendue, prise sans... | ...their names appeared on the screen, where possible. Unbilled players and technical personnel are added, where they can be reasonably documented. Various websites--for example, the Internet Movie Database (www.imdb.com)--feature credits for Micheaux f... | ...foes, and both captured. In 1513, just before the battle of Flodden, its walls were at length laid low by James IV., but not until the famous cannon Mons Meg-still, I believe, to be seen at Edinburgh Castle-had been brought against it. One of th... |

| Topic Focus = 1 | Topic Focus = 2 | Topic Focus = 3 | Topic Focus = 4 | Topic Focus = 5 |
|---|---|---|---|---|
| .... TONGA) * kankir (23.4. DAGAARE, SOUTHERN) * kankoeng dagoeblad (36.2. VLAAMS) * kankong (36.2. INDONESIAN) * kankri (38.13a. KONKANI) * kankri (38.7a. HINDUSTANI) * kankro (38.13a. NEPALI) * kankro (38.13b. NEPALI) * kankrol... | ...someone said, a minute or a century ago. Give him a booster shot. It can't hurt him. Six people were sitting on a shelf, looking down at him. A bare-breasted woman, a white-haired oldster, a young Negro girl, a flabby executive-type man, a Goonto... | ...on blend, Vera reads off the label. Made in the Philippines. She flings it at me and I screech. It lands on my shoulder and I throw it at Ehma, who ducks aside. I grab it again and chase her into the kitchen. Here, here. I throw the camisole to... | ..., and Mission San Antonio, -25 in Monterey, , at Purísima de Cadegomó, in Sierra Gorda, , -25 as student of Serra, -32 travel to Alaska, -65, travel to California, travel to New World, , Croix, Marquis Carlos de, , , , , , Croix, Teodoro de... | ...Ohio Railroad, 103-8, 105 , 109 banking industry, 26-27, 88, 93, 201, 212-13, 298, 305-8, 305 , 359-60 see also specific banks bank notes, 40-41, 90 Bank of Augusta v. Earle , 88, 90, 97-103, 115, 132, 145-46, 181, 400 Bank of England,... |

| Overall Score = 1 | Overall Score = 2 | Overall Score = 3 | Overall Score = 4 | Overall Score = 5 |
|---|---|---|---|---|
| ...02-03, , , , Barksdale, Brianna (character), , , , , Barksdale, D'Angelo (character), , , , , , , , , , , , , , , 285-86 and Avon, 84-85, , , , , and chess, , , and desire to escape the Game, , , , , and McNuggets, and Stringer, , and Walla... | ...68 . V. Chepizhny 1984 . V. Chepizhny 1984 . V. Chepizhny 1983/84 . N. Cherniavsky 1976 . E. B. Cook 1868 . N. N. after E. B. cook 1913 . E. B. Cook 1868 . E. B. Cook 1868 . C. H. Courtenay 1868 . C. H. Courtenay 1870 . J. Cumpe... | ...You work for free, I'll put you up at a nice hotel. You can eat great food and I'll give you a hooker. **ADAM DUBIN:** I always give the Beastie Boys a lot of credit, because nobody in the record business, even people who knew them, would've given... | ...-ended is also crucial here. I have addressed her important concerns in Chapter One. See Le Dœuff (1989: 126-8). **20** Propositional and assertive modes, while valuable for certain philosophical work, arguably restrict the open-ended inquiry that t... | ...uncle's life, but changed things enough for my dad so that he never met and married my mom? I pondered that for an extremely long time, and what I kept coming back to was this: What if I did? Selfishly, I had to admit I liked being alive, but I also ... |

Table 25: Raw training examples selected to have 14 quality ratings from 1 to 5 within C4.

| Accuracy = 1 | Accuracy = 2 | Accuracy = 3 | Accuracy = 4 | Accuracy = 5 |
|---|---|---|---|---|
| ...From data of data the quantity of prostitutes on the planet exceeds 42 millions today. Your e mail us' should contain appropriate infor- mation on such web page. Frequently it is the identi- cal message. Some pages provide you with a few li... | ...day services available in many instances, to make sure the waste products tend to be taken care of as quickly and effectively as possible. Otherwise same day, the majority of places guarantee their storage containers to be right now there by the subs... | ...facts it decides to ignore. Hamas is not a terrorist organization. It is a political party, which has gained control of Gaza by legitimate democratic means, during an election which was witnessed by former President Carter. In fact the Carter Cente... | ...-corrosion and anti-rust agents. The oil has a Proprietary technology add on that provides an extra layer of protection that helps the engine cool while maintaining proper temperature stability and oxidation. That is a feature you will not find with ... | ...31400 (2016). AIP Conference Proceedings, 1741, 050016 (2016). European Journal of Inorganic Chemistry, 4273-4274 (2016). J. Am. Chem. Soc., 137, 11498-11506 (2015). Chemical Science, 6, 4306-4310 (2015). Inorganic Chemistry, 54, 11593-11595 (2015). ... |

| Coherence = 1 | Coherence = 2 | Coherence= 3 | Coherence = 4 | Coherence = 5 |
|---|---|---|---|---|
| ... from your committee into a plasma. The client shared the amount and limited with a clearinghouse internet the contraction of ad pushing your traffic and security pp.. material of the law for ASIMO, Japan's solid effect? It takes come > with covalen... | ...bow lake left from an old channel of the Mississippi River. One floating casino is located on the lake near the downtown area known as the Trop Casino Greenville, with a second just west of the city near the Greenville Bridge known as Harlow's Casino... | ...ing plate drive mechanism. 0001 The stitching home position sensor does not go on when the stitch motor (rear) has been rotated in reverse for 0.5 sec or more. Stitch motor (rear; M6S)/stitching home position sensor (rear; MS5S) Replace the stitcher ... | ...SHOCKING – Package Seized and Destroyed by Royal Mail! However this was mainly aimed at international air shipments but since then further rules have been introduced by the International Air Transport Association, or IATA for short, with their Danger... | ...This is in response to your Request for Set Aside of Denial of Further Review of Protest and to Void the Denial of Protest Number 4103-98-100300 filed on behalf of the importer, Mast Industries, Inc. (Protestant), seeking to set aside the Port's de... |

| Creativity = 1 | Creativity = 2 | Creativity = 3 | Creativity = 4 | Creativity = 5 |
|---|---|---|---|---|
| ...Waterville area police reports for June 14, 2016. IN ANSON, Monday at 3:24 p.m., a harassment complaint was investigated on Hilltop Road. IN CLINTON, Monday at 9:45 a.m., a report of harassment led to an oral warning on Diamond Avenue. 3:57 p.m., thr... | ...Seize the deal before it's gone. Check out 70% Off Fine Jewelry. at Bealls Department Store now. Find more discounts and offers from Bealls Department Store just at CouponAnnie in April 2019. Save 70% off Fine Jewelry. Seize the deal before it's gone... | ...A home building project can be the chance for a couple to make their dream come true. The family has grown, there has been professional success – for whatever reason, it's time to invest hard-earned funds in a renovated kitchen, an extension or a bra... | ...whom he worked and he behaved in the manner of an idle dandy. He would even conduct interrogations lying on a settee draped in rich Chinese silks, manicuring himself while he put his questions. Yet he had inspired trust and was tolerated with amuseme... | ...As Obi-Wan stands at his Master's funeral, he remembers all that has happened... and all that could have been. The memories that now rampage my mind are painful, yet so beautiful in their sweetness. How can this be? How can he be gone? Everything he ... |

| Grammatical Diversity = 1 | Grammatical Diversity = 2 | Grammatical Diversity = 3 | Grammatical Diversity = 4 | Grammatical Diversity = 5 |
|---|---|---|---|---|
| ...aked to access where this contributes us. Our contradictory network contains really said as it implies when we 've functions. In Time to know a induction to present as it is in us, the registered company has the s exposure. Vanities eliminated been i... | ...If you are the typical inventor, it is very much possible that you would probably like to license all your invention and receive royalties, or even sell that it outright – we'll dub that person royalty author. But if you really are more motivated w... | ...grid sale calculation to Sales.applyGridSales to allow easier modding. – Fixed: gdt-modAPI: Checks.checkMissionOver did not return true when it should. – Fixed: When using windowed mode the window size starts smaller than 1024×768. – Fixed: End ... | ...be against one of their own. We should have listened. In the book, I could not count the number of times Obasanjo slapped his wife, Madam Remi, or the number of times she herself slapped many of Obasanjo's numerous mistresses and paramours, including... | ...Matthias Schoenaerts and Diane Kruger headline a sharp, slinky dive into genre territory for soph helmer Alice Winocour. Maryland is the original title of Disorder, the second feature by Parisian writer-director Alice Winocour, and while not one ... |

| Knowledge Novelty = 1 | Knowledge Novelty = 2 | Knowledge Novelty = 3 | Knowledge Novelty = 4 | Knowledge Novelty = 5 |
|---|---|---|---|---|
| ...Free parking lot to the Plan and the Village Alpiaz, near to the footsteps. 8 Gavardina street, near the Body shop Livingstone 2, in the place Bettoletto. Ample parking lot before the Blue Camping Bosco, next to the ski footsteps. Parking lot of the ... | ...eastern before washington journal. » on tuesday, the present the unit – european parliament discussed efforts to combat the islamic state and other terrorist groups following the recent attacks in brussels. intelligencessd sharing, islamic radical... | ...nay's The Legends of Jerusalem that Rabbi Luria supposedly knew in his day in a supernatural way where Jeremiah was placed in the Court of the Guard mentioned in Jeremiah 32:2. This was the key. and when Ha-Ari the Holy saw him. There was also a sid... | ...hoped to construct one piece in what will be a much larger conversation. And we certainly didn't start anything ourselves; people were talking about this, and we just wanted to give it a bit of a push and inspire others to take on what needs to happe... | ...CR chain sequences from reactive CD4 T cells from 22 individuals with latent Mycobacterium tuberculosis infection. We found 141 TCR specificity groups, including 16 distinct groups containing TCRs from multiple individuals. These TCR groups typicall... |

Table 26: Raw training examples selected to have 14 quality ratings from 1 to 5 within C4.

| Language Consistency = 1 | Language Consistency = 2 | Language Consistency = 3 | Language Consistency = 4 | Language Consistency = 5 |
|---|---|---|---|---|
| ...Antikythera. While the nanoscale saved for the philosophique to ignore, the service of the efficacy were one of his texts to run the rotation for is. The , Elias Stadiatis, taught the breaks of a bacterial element at the knowledge of the side, 60 act... | ...Which user Loves freedom the most or Hates freedom the most? If only you could see how against freedom you are.. I love freedom so much anyone who doesn't love it should go die! You wanna back that up? Being against abortion doesn't make me anti-fr... | ...good quality furniture affordable quality furniture affordable quality furniture affordable quality furniture high quality affordable high quality furniture brands sofas. good quality furniture high quality furniture brands high quality furniture bra... | ...Are you aware of any new treatments being researched? How do you feel about participating in clinical trials? What would prevent you from involving your child in a medical research study? Interact and reply in the comments below! Not sure about new t... | ...On Arrival: Presuming you are already within Italy prior to arriving in the Amalfi Coast, most commonly you will arrive by train in Sorrento (via Naples) or in Salerno. We buy our train tickets online through Trainline. From here you can get to the v... |
| **Originality = 1** | **Originality = 2** | **Originality= 3** | **Originality = 4** | **Originality = 5** |
| ...table below, if you want your kids wear longer time, please choose bigger size. Factory direct sell, quality assurance. Made in china, fabric is incredibly soft and comfortable. Factory direct sell /1 piece retail hot sell /free shipping. – Machine W... | ...Spinit boasts a choice of games and slots and is also a brand new casino. Supplying a variety of payment choices and Offered in some languages, its prevalence has been increasing among fans. It's fun and contemporary, and more importantly for a casin... | ...includes services such as recreation and social activities. Seajoy Cla provides assisted living not only to Decatur residents, but also to all Dekalb county residents as well. If you need assistance with daily tasks, Gordon Care can help you or your ... | ...Oh yeah, that meaningless, well-worn catchphrase used by influencers and self-help coaches. What's in it for real people though? In fact, personal branding is something you should develop and control. Especially when it comes to your professional lif... | ...After 5 years of a continuous lawful stay in Luxembourg, third-country nationals (i.e. from a country that is neither an EU Member State nor a country treated as such – Iceland, Norway, Lichtenstein and Switzerland) may make an application to obtain ... |
| **Professionalism = 1** | **Professionalism = 2** | **Professionalism = 3** | **Professionalism = 4** | **Professionalism = 5** |
| ...Programs, or Director of Admission, inhabit flounders, and build a relevant besondere for our consequences. using the CAPTCHA gives you involve a giant and coincides you aspen free Conjugate gradient method without to the concrete inanity. What can I... | ...so long. But yesterday at lunch, I sat on a high hill in a rolling green park, underneath a huge, old tree, listening to the church bells. And then I went back to work. I am working now, as web designer for a college in Leek. As this is the week when... | ...the majority of the work was done during the week. Brainstorm or create content upgrades (freebies I give away at the end of the post) A perfect example would be the After 9 to 5 Thrive Guide that is included in this post. Brainstorm or create tripwi... | ...If you're thinking of selling your property, our qualified real estate team is here to make the process as easy as possible. We have a friendly and dedicated staff that are devoted to the profession and staying on top of the market and trends. We are... | ...Adriane Martin, DO, FACOS, CCDS, explains the confusion behind the various sepsis definitions and provides guidance to coders when reporting sepsis in ICD-10-CM. For patients who suffer from frequent symptoms of gastroesophageal reflux disease (GERD)... |
| **Semantic Density = 1** | **Semantic Density = 2** | **Semantic Density = 3** | **Semantic Density = 4** | **Semantic Density = 5** |
| ...Hi, I found your listing on Padlist and I'm interested in coming to see it: www.padlist.com / listings / 500-monroe-avenue-ne-renton-wa- 98056-1233 Can you please let me know if it's still available, and when I might be able to view it? Thanks! Hi, I foun... | ...President Trump has a prospective ally in the war on Muslim immigration, and I am not talking about the patriots in the U.S. Border Patrol (USBP) or U.S. Immigration and Customs Enforcement and Removal Operations (ICE ERO), as previously discussed in... | ...6 +6 2014 aug 29 llancaiach fawr manor Llancaiach Fawr Manor – The Locations Guide to Doctor Who Location: Llancaiach Fawr Manor . Details from The Locations Guide to Doctor Who, Torchwood, and the Sarah Jane Adventures. 9 –3 2014 sep 05 shire hall m... | ...After much googling in an attempt to get my Matrox G400 fully operational, I came across this forum, which appears to have an active and knowledgeable community. I hope someone can help me with my dilemma. The adapter card is recognized as a Matrox G... | .... In resurrectione tua Christe, alleluia. . In thy resurrection, O Christ, alleluia. . Cœli et terra lætentur, alleluia. . let heaven and earth rejoice, alleluia. Jesus has provided for everything; he has chosen twelve men, whom he calls his Apos... |
| **Sensitivity = 1** | **Sensitivity = 2** | **Sensitivity = 3** | **Sensitivity = 4** | **Sensitivity = 5** |
| ...surance. Thanks for the info. I was just wondering about that. :-) But even if someone agrees with my point of view, I still have a policy of deliberately never patronizing any entity that uses guerrilla advertising techniques, spam etc., so this act... | ...second stage engine should be ready by 2022-23, so if this is true one can imagine that all of the units of the second contract would get izd. 30, either retrofit or directly installed, it does not make sense to have half squadron with some engine an... | ...How can I get the best 70-461 Prep Guides Guaranteed Success training Prep Guides? Now cilck in [gooexam.com] is work. MCSA How can I get the best 70-461 Prep Guides Guaranteed Success training Prep Guides? Now cilck in [gooexam.com] is work. Latest ... | ...'formally', and we have expanded them with the positive liberties socialism promises. We have real socialist democracy, popular democracy, genuinely free elections, etc. In this sense we can say it betrayed its own premises, because the whole thing ... | ...Finding a job in the healthcare industry will not be difficult if your New Grad Nursing resume has all the important details on your skills and other qualifications. Looking for work can be a daunting task for new Nursing grads. Experience is a big d... |

Table 27: Raw training examples selected to have 14 quality ratings from 1 to 5 within C4.

| Structural Standardization = 1 | Structural Standardization = 2 | Structural Standardization = 3 | Structural Standardization = 4 | Structural Standardization = 5 |
|---|---|---|---|---|
| ...Much-loved character actor who specialised in playing slightly sleazyslightly eccentric and often ... Komentar Br. 2 Poslao : Severin Datum : 03. 2005. Najbolje da vas muz provjeri kolesterol. Vrlo cesto povisen kolesterol je uzrok kamencima, kako zucn... | ...Fantastic Rubin Sofa By Woodhaven Hill is important have in almost any home. You need for the greatest items, so you want to actually never overpay for them. Noises a little complex, proper? Effectively, this post is on this page to help. Keep readin... | ...So, the latest embarrassment for the German domestic spy agency, officially called the Federal Office for the Protection of the Constitution (Bundesamt für Verfassungsschutz/BfV) is the revelation that it has been spying on at least 1/3 of the feder... | ...kel was forced to cancel a planned visit to the newsroom of Southern Weekend, a Guangzhou-based paper known for outspoken reporting on corruption and other sensitive issues. The schedule of Merkel's three-day trip to China was altered to insert a mee... | ...The President of India, Shri Pranab Mukherjee will visit the Rashtrapati Nilayam, Secunderabad for a ten day southern sojourn from June 29- July 08, 2015. During his southern sojourn, he will visit Tirupati in Andhra Pradesh on July 01, 2015. On July... |
| **Style Consistency = 1** | **Style Consistency = 2** | **Style Consistency = 3** | **Style Consistency = 4** | **Style Consistency = 5** |
| ...While some light no obligations, others getting to bloating, absolute pain, or yoghurt and long periods. Inward, is the other preparation two weeks prior to emergency. This process cools your energy and helps aid your tolerance body treatment. Its lo... | ...If you are ready for the most amazing therapy services call the professionals at Body Central Physical Therapy soon as possible so they can provide you the best solutions the matter what. You'll of work with these guys because they really do care by ... | ...a child accidentally knocks the Spaghettios and meatballs off the table and it lays in a giant red heap on the floor. The last paper towel used only seconds prior to the incident. When all else fails, you use a dishtowel. When the same child that acc... | ...Image from a virtual sculpture that was created from two people tracing the front side of their bodies. The result of which is a sculpture of the space between the two bodies. Image from a virtual sculpture that was created from tracing my body as I ... | ...I had a doctor's appointment today, and in the office, they had a poster asking their patients for patience as they work to implement electronic records. The signs have been up for the last few visits, so I think the bulk of the efforts are complete.... |
| **Topic Focus = 1** | **Topic Focus = 2** | **Topic Focus = 3** | **Topic Focus = 4** | **Topic Focus = 5** |
| ...include any person who. Probably, the Orient comes second as far as the popular fruit machine themes go. Golden Lotus is a slot play developed by RTG, the company that is famous for production of the quality software for the online casinos. misc. tra... | ...this direction. the objective function to be continuous in every parameter, which not is always the case. In the specialized literature we can find other alternatives to the Gauss-Newton method problem. For example, the singular value decomposition (... | ...Raw sockets allow a program or application to provide custom headers for the specific protocol(tcp ip) which are otherwise provided by the kernel/os network stack. In more simple terms its for adding custom headers instead of headers provided by the ... | ...Home > Fitness > Can Lemon Water Really Help You Lose Weight? and clinical trials need to be conducted in people before any claims can be made. Drinking lemon water is regarded by many professional nutritionists as having real and palpable weight-los... | ...anglaise.) Squeeze all the water out of the gelatin, and add the gelatin to your hazelnut creme anglaise, stirring until it has melted and has been incorporated. Stir over the ice bath until the mixture is chilled. Fold 1/3 of the whipped cream into ... |
| **Overall Score = 1** | **Overall Score = 2** | **Overall Score = 3** | **Overall Score = 4** | **Overall Score = 5** |
| ...university.for more infirmation contacy me at xxx@gmail.com or 095306403 i am looking forward hearing from you. I really do appreciate your kind gesture for reply me and granting me this opportunity.Am a Nigerian.32 years old of age.I ha... | ...This is the Hee Haw penny slot machine Free Games san manuel casino Bonus. Find short term apartments, houses and rooms posted by Nassau Paradise Island landlords. Our best Strategy Games include and 747 more. Trailways offers luxury motorcoach trans... | ...Running Festival Wychwood December 12/12/2019 to 01/01/2020 with 20/10 day 1000 mile/1000 km 6 day with shorter races. « Running Festival Wychwood June 16/06/2019 to 22/06/2019 with 6day/72 hour and shorter Races. IS THERE ANY RUNNERS BRAVE ENOUGH TO... | ...about some high-maintenance lettuce wraps here. Not the kind of high-maintenance that would scare you away. You know I don't like things to be too complicated around here. Just the kind of high-maintenance that takes something that needs a... | ...Wed., September 19, 2018 10:51 a.m. | Wednesday, September 19, 2018 10:51 a.m. In this early morning Sept. 8, 2018 photo, 56-year-old dialysis patient Elias Salgado prepares for his trip to the Puerto Rican mainland, at his home in Vieques. Salgado i... |

Table 28: Raw training examples selected to have 14 quality ratings from 1 to 5 within CommonCrawl.

| Accuracy = 1 | Accuracy = 2 | Accuracy = 3 | Accuracy = 4 | Accuracy = 5 |
|---|---|---|---|---|
| ...last Friday and he angry!! Subscribe. Or, at least they WILL live in a 130-story treehouse once they finish building all 1 Since then it's been a constant learning and experiencing through mistakes and growth, all to provide our customers with the be... | ...The Final Solution (History Essay Sample) / Samples / History / The Final Solution ← The Mexican War The Irish Republican Army → Check Out Our The Final Solution Essay More than fifty years ago, over a four year period more than eleven million people... | ...in the purest light. Use the list below to get more information on majors, minors, and the departments and programs that administer them. Black includes African American, Hispanic includes Latino, and Pacific Islander includes Native Hawaiian. Which ... | ...ana Shiva presents in the report The Future of Our Daily Bread: Regeneration or Collapse new evidence on the imminent collapse of our food systems if we continue on the path of industrial agriculture. 23096985674e966136factheipganal.jpg 10 demands for an en... | ...as big as a car, big rounded pots of chrysanthemums. And he had cascading chrysanthemums coming down off the wall, and then there's a wall that leads down into our big reflecting pool that terminates the canal. And he would grow these cascading chrys... |

| Coherence = 1 | Coherence = 2 | Coherence= 3 | Coherence = 4 | Coherence = 5 |
|---|---|---|---|---|
| ...full of insightful information and entertaining descriptions. Your point of view is the best among many. hpornvideo Hey, thanks for the blog post.Much thanks again. Awesome. pounded02 I think this is a real great post.Really looking forward to read m... | ...we will ourselves happily jump on the grenade if it screws over the other group. It's fairly easy to see how an Us vs. Them mentality can be destructive to both ourselves and society. It is striking to think how susceptible we are to this mentality... | ...ition, the threat of a mass demo and a very big political row. In the same year Braddock was made a Dame for her services to FE. In the wake of the row over the veils Braddock stated that she was approaching 60 and had always planned to retire. So it... | ...The goods on white Betting on chrome Wörwag's new Chrome paint has become the benchmark in the market thanks to its technology edge. But how did it come about? A look behind the scenes of the development department shows that invention has a lot to d... | ...McQuaid holds off Penfield with defense in Class AA baseball semifinal Junior pitcher Hunter Walsh nearly pitched a complete game during the Knights' win at Frontier Field. McQuaid holds off Penfield with defense in Class AA baseball semifinal Junior..." |

| Creativity = 1 | Creativity = 2 | Creativity = 3 | Creativity = 4 | Creativity = 5 |
|---|---|---|---|---|
| ...29, 1992 Did not seek election. KruegerRobert C. Krueger (D-TX) 19930121Jan 21, 1993 No, defeated on January 5, 1993. FrahmSheila Frahm (R-KS) 19960611Jun 11, 1996 No, defeated for nomination. ChafeeLincoln D. Chafee (R-RI) 19991102Nov 2, 1999 Yes, o... | ...China, Bangladesh to build overland trade route? By Patrick Scally in News on October 19, 2012 Ongoing efforts to connect Kunming to markets in India and central Asia took an apparent detour yesterday in Beijing. China and Bangladesh tentatively agre... | ...Live Twitter Chat SMT Experts Is Facebook Really to Blame for Increasing Political Division? Andrew Hutchinson @adhutchinson There is a lot to take in from the latest New York Times' latest report on an internal memo sent by Facebook's head of VR and... | ...HomeOpera Hector Berlioz, La Damnation de Faust, Metropolitan Opera, November 7, 2008 November 10, 2008Michael Miller Filed UnderOpera Susan Graham as Marguerite in LaSusan Graham as Marguerite in Berlioz's La Damnation de Faust. Photo Ken Howard/M... | ...August Šenoa: The Goldsmith's Treasure Numerous tales and legends exist in Zagreb and about Zagreb; some are remembered through the ages, other sink into oblivion, and even the origin of some is forgotten over time. August Šenoa's The Goldsmith's Tre..." |

| Grammatical Diversity = 1 | Grammatical Diversity = 2 | Grammatical Diversity = 3 | Grammatical Diversity = 4 | Grammatical Diversity = 5 |
|---|---|---|---|---|
| ...04/03 (18) 03/20 – 03/27 (15) 03/13 – 03/20 (16) 03/06 – 03/13 (15) 02/28 – 03/06 (20) 02/21 – 02/28 (20) 02/14 – 02/21 (20) 02/07 – 02/14 (20) 01/31 – 02/07 (19) 01/24 – 01/31 (19) 01/17 – 01/24 (16) 01/10 – 01/17 (16) 01/03 – 01/10 (19) 12/27 – 01/ 12/27 – 01/ 12/27 – 01/... | ...Evel Knievel 1971 27 Mar 2019. HISTORY and Nitro Circus announce the return of the live television event Evel Live 2 premiering Sunday, July 7 at 8PM ET. Find high-quality Evel Knievel stock photos and editorial news pictures from Getty Images. Dow... | ...with jihadist Islam. Stereotyping can have very dire consequences; just ask Pastor James McConnell, the outspoken cleric who is retiring from North Belfast's Whitewell Metropolitan Tabernacle. He branded Islam 'satanic' ... | ...M. had known her socially before we went to North Wales. He had been letting me spend 18 months of my life working on this scheme, and building up chances of a future for myself. I received £1,000 from M. because I had been more than £1,000 wronge... | ...Amniotic fluid stem cells prevent development of ascites in a neonatal rat model of necrotizing enterocolitis Aim: It has been demonstrated that in a neonatal rat model of necrotizing enterocolitis (NEC), amniotic fluid stem (AFS) cells decrease..." |

| Knowledge Novelty = 1 | Knowledge Novelty = 2 | Knowledge Novelty = 3 | Knowledge Novelty = 4 | Knowledge Novelty = 5 |
|---|---|---|---|---|
| ...Mahoney Constance Baker Motley Esther Peterson Jeannette Rankin Ellen Swallow Richards Elaine Roulet Katherine Siva Saubel Madam C. J. Walker Faye Wattleton Rosalyn S. Yalow Gloria Yerkovich Bella Abzug Myra Bradwell Annie Jump Cannon Jane Cunningham... | ...country, and whoever was coming was the buyer. >make a move now >wait, hide and observe Shark might be another super. Let's not do anything rash now. It'd be nice to bust the buyer too. From last thread >You hear the one about the shark-man on the d... | ...– Messij (Classic Pack) CoLD SToRAGE (Classic Pack) Takkyu Ishino – Jingle WIRE05 (WIRE05 Pack [JP]) Akira Ishihara – Breaking the Ice (Continue Pack [JP]) Akira Ishihara – Open the P.A. (Continue Pack [JP]) Oblivion Records – ... | ...Shirley MacFarlane 461 Broad Street N. Regina SK S4R 2X8 25 Things You May Not Know About Saskatchewan Vol. 12 Issue 5 By Lorne McClinton 1 During the middle Devonian Period, between 375 and 400 million years ago, much of Saskatchewan was covered b... | ...-addition step (i.e., the potential initial addition of the nucleophile to the C– of the bis-electrophile) has to be slower than the intermolecular addition of the nucleophile to the catalytically generated 3-allylpalladium complex, or it has to be... |

Table 29: Raw training examples selected to have 14 quality ratings from 1 to 5 within CommonCrawl.

| Language Consistency = 1 | Language Consistency = 2 | Language Consistency = 3 | Language Consistency = 4 | Language Consistency = 5 |
|---|---|---|---|---|
| ...weixin, weixin, weixin, weixin, weixin, weixin, weixin, weixin, weixin, weixin, weixin, weixin, weixin, weixin, weixin, weixin, weixin, weixin, weixin google,google, google info... | ...right side, Ive done it so long that I know how to use my hands better, said Schwartz, who played nine games at right tackle for Carolina in 2009 and 2010. I dont think about it. You just gotta go play.... DT Cullen Jenkins (calf) and LB Jacquian Wil... | ...'s Land, Korvatunturi, up north, where the Bock family lived 200 from 1050 to 1248. 9The Stone 'image' of Noah's Ark in Gotland. THE ICE-AGE THE ATLANTIS-TIME Uden's land, Atlantis Hel, it's capital Cut off from the other world 10 tropical races for... | ...he had admitted it, though he could still pretend, but he had betrayed me. That evening we greeted differently than usual, I felt frozen, manipulated, I wanted to be alone, I didn't want to know anybody anymore. After 15 days, the longest interval be... | ...meta-analysis. Lung Cancer 47:81-83, 2005. 37. The PORT Meta-analysis Group: Postoperative radiotherapy in non-small-cell lung cancer: Systematic review and meta-analysis of individual patient data from nine randomised controlled trials. Lancet 352:2... |
| **Originality = 1** | **Originality = 2** | **Originality= 3** | **Originality = 4** | **Originality = 5** |
| ...(5) Jan 26 (6) Jan 27 (5) Jan 28 (2) Jan 29 (4) Jan 30 (2) Jan 31 (2) Feb 01 (5) Feb 02 (2) Feb 03 (4) Feb 04 (1) Feb 05 (2) Feb 06 (3) Feb 08 (6) Feb 09 (4) Feb 10 (3) Feb 11 (5) Feb 12 (3) Feb 13 (3) Feb 15 (9) Feb 16 (2) Feb 17 (5) Feb 18 (6) Feb Feb 18 (6) Feb Feb 18 (6) Feb... | ...Search By Office Search by Office Baton Rouge Dallas/Fort Worth Fort Worth Gulfport Houston Jackson London Mobile New Orleans Raleigh Tampa Tupelo Search By Admissions Search By Admissions Alabama Arkansas California District Court for the Fort Berth... | ...some fall into next year? Yes, I just think less than 22. If if the Fed policies and by the way, the ECB starts to tone it, the Bank of England, et cetera. Well, you talk about the Fed coming off the boil. It looks like at least are going to slow dow... | ...HomeBuilt SpacesKilokhri: The lost city of Delhi By Arya Sethi 15 August 2021 0 In the list of 7 cities of Delhi, this city does find its place, as its ruins have not survived the ravages of time, unlike the others. While we have extensive works avai... | ...as the syndecans,124,126 DDRs (Discoidin Domain Receptors),148 and integrins.10,149-152 The latter are noncovalently linked heterodimers of one and one subunits. Integrins have a short cytoplasmic domain (except for the 1,000... |
| **Professionalism = 1** | **Professionalism = 2** | **Professionalism = 3** | **Professionalism = 4** | **Professionalism = 5** |
| ...clumps of smoke, the smoke witch descends from above the burning roof to get closer to the fat boy she so despises. Her ember eyes burn into his fat boy soul. Walter screams out in agony. He waddles backwards toward that side of the house where the o... | ..., 43). In addition, MSCs are known to transdifferentiate into neuronal and glial cells. MSCs have been shown to migrate to damaged neuronal tissues and to alleviate the deficits in experimental animal models of cerebral ischemia (10), spinal cord inj... | ...in Travel and Tourism 5 Fun Things To Do In Slovenia For An Otherworldly Experience ...how about a trip to the green Heaven on Earth? by Yanna N. Niksy (CC0), Pixabay If you're sick and tired of only doing the same old touristy stuff, think about visit... | ...Sobral, Ceará Sobral is a municipality in the state of Ceará, Brazil. City waterfront Princesa do Norte (North's Princess) Sobral cada vez melhor Location in the state of Ceará and Brazil Location in Brazil Coordinates: 03°4026S 40°1420W / 3.67... | ...catch-and-run, and Alabama takes a 21-14 lead on Ohio State with 9:00 left in the second quarter. Alabama is -700 on the live line (Ohio State +475), spread -13½, total 84½. 6:09 p.m.: Alabama fumbles, and Ohio State makes the Tide pay. Teague scores... |
| **Semantic Density = 1** | **Semantic Density = 2** | **Semantic Density = 3** | **Semantic Density = 4** | **Semantic Density = 5** |
| ...Select a postFake Saint 1Fake Saint 2Fake Saint 3Fake Saint 4Fake Saint of the Year 5Fake Saint of the Year 6 (Part 1)Fake Saint of the Year 6 (Part 2)Fake Saint of the Year 7Fake Saint of the Year 8Fake Saint of the Year 9Fake Saint of the Year 10Fa... | ...a meaningful dialogue among all the interested parties. But unfortunately this was not to happen.. Of course, a meaningful dialogue can only take place if the parties maintain objectivity. You don't earn marks for credibility when you give credence ... | ...We cannot always rely upon the wisdom of a leader, or mentor. Sometimes, the truth escapes those who serve, but consultation with an attorney is useful if you have access to one. Much of this material is developed to help those who've been denied leg... | ...with Huggy and he was like a Petey on the back end. But he was my runner up for the last question. JT barely beat him out. But yeah he is amazing, I did not think that he was gonna be battling for the Rookie scoring lead. I also thought he would be m... | ...Revision Notes for the Paris Peace Treaties During the peace process, the Western governments had a chance to reflect on what had gone wrong and to design a peace settlement that would restore stability and confidence in European leadership. I. The P... |
| **Sensitivity = 1** | **Sensitivity = 2** | **Sensitivity = 3** | **Sensitivity = 4** | **Sensitivity = 5** |
| ...50% Daily 2018-11-21 23:31 https://pmmodinews 50% Daily 2018-11-21 23:31 https://pmmodinews 50% Daily 2018-11-21 23:31 https://pmmodinews https://pmmodinews https://pmmodinews https://pmmodinews https://pmmodinews https://pmmodinews https://pmmodinews...knockin' the ... | ...is to say looking at lying-ass dogs like me sitting where I was sitting and telling him all kinds of crap about being reformed, finding religion, getting an honest-to-God job, and settling down. No more meth, booze, cooze, brawling, and | ...on bringing a water bottle so you can sip it throughout the exam. Energy drinks are popular because of their branding and association with sports and physical stamina. In addition, when your muscles have more energy, it gives you the ability to work ... | ...go witness Lily and James' murder and the fic doesn't deal with it, then the agents should have a good reason for doing it or be willing to face reprimand from a Flower. More answers by Ellipsis Flood on 2012-01-23 18:34:00 UTC Link to this !) Well, ... | ...Human Rights in Economic Policy Human Rights in Sustainable Development Rights Claiming and Accountability Accountability is a cornerstone of the human rights framework. It has both a corrective and preventative function. It addresses individual and ... |

Table 30: Raw training examples selected to have 14 quality ratings from 1 to 5 within CommonCrawl.

| Structural Standardization = 1 | Structural Standardization = 2 | Structural Standardization = 3 | Structural Standardization = 4 | Structural Standardization = 5 |
|---|---|---|---|---|
| ...ing it: fawning compliance with Zionism, and the atrocities committed by that racist, genocidal and insane protectorate Israel. then lose it by; Zionism is pretty much Christianity without the messiah, after all. The notion of discounting physi... | ...Trump's crackdown on free press in Alabama is 'a step in the right direction' President Donald Trump's administration has been working to restrict the press in states around the country in an effort to keep the country's reputation for transparency, ... | ...mila Cvetkovic, Andy Heil Serbia and its neighbors Kosovo and Albania have descended into a bitter diplomatic ruckus after officials in Belgrade invoked a slur... Albania, Greece Must Reflect on Past Mistakes to Settle Maritime Borders Is... By Akri ... | ...into the realm of a Super Saiyan. It's fair to say, without the Namek / Frieza saga making little boys and girls tune in after school, that anime might not have been such a huge hit here in the UK. 2) Klingons – Star Trek Riker dines with the Klingon... | ...The Potential of Tropical Agro-Industrial by-Products as a Functional Feed for Poultry Document Type : Review Articles S. Sugiharto T. Yudiarti I. Isroli E. Widiastuti Department of Animal Science, Faculty of Animal and Agricultural Science, Diponego... |

| Style Consistency = 1 | Style Consistency = 2 | Style Consistency = 3 | Style Consistency = 4 | Style Consistency = 5 |
|---|---|---|---|---|
| ...5 December 2014 November 2014 October 2014 September 2014 August 2014 July 2014 June 2014 May 2014 April 2014 March 2014 February 2014 January 2014 December 2013 November 2013 October 2013 September 2013 August 2013 July 2013 June 2013 May 2013 April... | ...with a vehicle at the intersection of Grand and Forest Ms. Anderson saw a dark, hooded figure reach through the window, grab a small parcel and run north on Forest. d. After colliding with a vehicle at the intersection of Grand and Forest, Ms. Anders... | ...time had come to take a common stand. But the Israeli armed forces would not tolerate what they viewed as insubordination. They turned their weapons on the protestors, killing six outright, injuring dozens and arresting hundreds who persisted in thei... | ...case, it emerged that nonresponses and absences did not result in disciplinary sanctions. Other factors taken into account were the autonomous organisation as regards time and space, the absence of exclusivity clauses and noncompete agreements. See C... | ...ductory essays. (shrink) Review of Pearson, Aristotle on Desire. [REVIEW]Thornton Lockwood – 2013 – Bryn Mawr Classical Review 9:24.details The image of a copy of Praxiteles' Aphrodite--nude but demurely shielding her pubic region--which adorns the dus... |

| Topic Focus = 1 | Topic Focus = 2 | Topic Focus = 3 | Topic Focus = 4 | Topic Focus = 5 |
|---|---|---|---|---|
| ...Être partisan de Fayulu n'est pas un péché, dit général Kasonga aux policiers qui ont matraqué Serge Welo, l'un des généraux du président élu Arrestation de Ngoyi Mulunda : Félix Kabange mérite le même sort selon Jean-Claude Katende Union sacrée : ... | ...-minute and small seconds. This three-dimensional watch, reminiscent fake of a jet fake luxury watches engine, is encased in a complex case featuring alternating polished and satin-finished finishes. Both options have the orange accents on the dial a... | ...CPF or not, was protected under the constitution. I grant I might have misstated when I said ruled but the fact remains that the court agreed with that premise. As I stated, the only person who stands to gain from pursuing this lawsuit is Mr. Berko... | ...Press, October 12, 2020, 12:59 PM Two priests are going on trial in the Vatican's criminal tribunal this week, one accused of sexually abusing an altar boy who served at papal Masses in St. Peter's Basilica, and the other accused of covering it up. T... | ...Employing a star, you could argue, is as proscriptive as religious observation. They arrive with expectation and mandate, and any deviation from the accreted screen history is a gamble. We are drawn to them because they are recognizable. Maybe the mo... |

| Overall Score = 1 | Overall Score = 2 | Overall Score = 3 | Overall Score = 4 | Overall Score = 5 |
|---|---|---|---|---|
| ...less. Seattle, King, Washington, US, 98101, (360) 764-.... Heike, Fewless. Phoenix, Maricopa, Arizona, US, 85255, (480) 375-.... Carlena, Fewless. Kissimmee, Osceola, Florida, US, 34741, (407) 742-.... Jason, Fewless. Pateros, Okanogan, Washington, U... | ...Steroids for sale new zealand, what is taking sarms Steroids for sale new zealand, what is taking sarms – Buy anabolic steroids online Steroids for sale new zealand Six sports supplements on sale in New Zealand have been found to contain anabolic ste... | ...BEACH BOY ARDASHIR VAKIL PDF Ardashir Vakil was born in Bombay and now lives in London. Ardashir Vakil's award-winning first novel, Beach Boy (), is Bombay's answer to James. Marrying a universal story (an adolescent boy's coming-of-age) with a speci... | ...stands at the door, with the rest of the Company and the wagon behind him. Moon Shadow is so happy to see them, he hugs Uncle. Uncle tells how the Company heard of the Lees' trouble and showed up to help with the flight of Dragonwings. They are deter... | ...human rights. A Tawdry Place of Salvation: The Art of Jane Bowles Edited by Jennie Skerl Southern Illinois University Press, 1997 Library of Congress PS3552.O837Z89 1997 | Dewey Decimal 818.5409 Through these essays--which deal with Bowles's published... |

Table 31: Raw training examples selected to have 14 quality ratings from 1 to 5 within Github.

| Accuracy = 1 | Accuracy = 2 | Accuracy = 3 | Accuracy = 4 | Accuracy = 5 |
|---|---|---|---|---|
| ...fiction retrogressions eliminates unknowns mongoloids N N 865 135 N N 869 inflecting trephines hops exec junketeers isolators reducing nethermost nonfiction N N 918 290 N N 86 forbearer anesthetization undermentioned outflanking ... | ...Comments supports topic icons</li> page (Link from viewprofile and viewtopic pages)</li> <li>Comments requiment maybe disabled via ACP</li> <li>Comments </li> <li>Comments requiment maybe disabled via ACP</li> ... | ...();break;case 1:data2 = new ArrayList<String ();data2. add(getIntent(). getStringExtra(pic)); imageAdapter = new ImageAdapter (LightNearby. this, data); gallery.setAdapter (imageAdapter); loacaladdress.setText (li.get(0). getAddress()); lastname.setText... | ...private void startVUser final Class vUserClass, final long coolDownDelay LpeSystemUtils submitTask(new Runnable() public void run() ISimpleVUser vUser;try vUser = (ISimpleVUser) vUserClass. newInstance() catch (Exception e) throw new RuntimeException(e); increaseNumActiveUsers();while... | ...  package cmwell.tools.data.utils. chunkers import akka.stream.stage. import akka.stream. Attributes, FlowShape, Inlet, Outlet import akka.util.ByteString import scala.collection.immutable import scala.collection. immutable.VectorBuilder import scal... |
| Coherence = 1 | Coherence = 2 | Coherence= 3 | Coherence = 4 | Coherence = 5 |
| ...script type=Syre>sirska abeceda estrangelo</script> <script type=Syrj> zahodnosirijski </script> <script type=Syrn> vzhodnosirijski </script> <script type=Tagb> tagbanski </script> <script type=Tale>Tale</script> | ...Util.rahToStr( hqResult.getBYP1(),2,images//w4.png hqResult. getCLOS())); ((TextView mBuyCantainer. getChildAt(0). findViewById (R.id.v)). setText (hqResult.getBYV1()); ((TextView mBuyCantainer. getChildAt(1). ). setText(two); ... | ...img sr c= alt=Stories on Camper News> <div class=image overlay> <a href=http://codepen.io target=blank> View Full Project</a> findViewById(R.id.ptx findViewById(R.id.ptx <script type=Talu <script type=Talu ... | ...lz4;resolution:= optional, resolution:= optional, resolution:= optional, resolution:= optional, net. jpountz.xxhash; resolution:= optional,reactor.blockhound;r esolution:=optional, reactor.blockhound. integration; resolution:=optional, ... | ...OBJECT TYPE.isSubtype(NO TYPE)); assertFalse(BOOLEAN OBJECT TYPE.isSubtype(NO OBJECT TYPE)); assertFalse(BOOLEAN OBJECT TYPE.isSubtype(ARRAY TYPE)); assertFalse(BOOLEAN OBJECT... |
| Creativity = 1 | Creativity = 2 | Creativity = 3 | Creativity = 4 | Creativity = 5 |
| ...  Abstract Deletion, Abstract Deletion, Abstract Deletion, IsReadState ChangesExist, IsReadState ChangesExist, FinalI CSState, IsSort ByMessage DeliveryTime, ... | ... // @Override public exitRule (listener: ANTLRv4ParserListener) void if (listener. exitOptionValue) listener. exitOptionValue(this) // @Override public accept<Result>(visitor ANTLRv4ParserVisitor <Result>): Result if (visit... | ...import fse from 'fs-extra' import uglifyes from uglify-es /tools/ index.js' const minify = uglifyes function uglify (userOptions) const options = Object.assign( sourceMap: true , userOptions) return name: 'uglify', transformBundle... | ...id: 514 title: Happy New Year! date: 2016-01-07T 12:23:04 +00:00 author: Jerri Glover layout: post guid: http:// blog..com/ ?p=514 permalink: /2016 /01/07/ happy-new-year/ categories: - General -- Happy 2016! We hope you all enjoyed your hol... | ...Nor the kind products of a bounteous year;</l> <l>No more the date, with nowy blooms crown'd</l> <l>But Ruin preads her baleful fires around.</l> </sp> <sp> <spea new SolidColorBrush (ImmersiveColor. GetColorBy TypeName (ImmersiveCol...... |
| Grammatical Diversity = 1 | Grammatical Diversity = 2 | Grammatical Diversity = 3 | Grammatical Diversity = 4 | Grammatical Diversity = 5 |
| ...></li> <li> </li> <li><spa n class= flag-TG title=.flag-TG> </li> <li> <... | ...—css3:before content: f13c; .fa-anchor:before content: f13d; .fa-unlock-alt:before content: f13e; .fa-bullseye:before content: f140; .fa-ellipsis-h:before content: f141; .fa-ellipsis-v:before content: ... | ...sp Factura Obtiene ElNumero De Documentos v1], data).Copy(); return dt; public DataTable GetOfficeQuote(int iEmpresa, int iNumero) DataTable dt = new DataTable(); SQLCone... | ...  : Immersive Color.Get ColorByTypeName (Immersive ColorNames. DarkChrome Medium") Opacity = WindowsTheme. Transparency. Current ?  0.6 :  1.0 ; public Brush Notification Foreground => | ...CharacterOf fsetBegin> <CharacterOffsetEnd> 2864 </CharacterOffsetEnd> <POS>NN</POS> <NER>O</NER> <Speaker> PER0</Speaker> </token> <token id=24> <word>,</word> ... |
| Knowledge Novelty = 1 | Knowledge Novelty = 2 | Knowledge Novelty = 3 | Knowledge Novelty = 4 | Knowledge Novelty = 5 |
| ...en, cachaauru rai rauni ita taojiaain chaelai que ereereena jerecuruha ne, ichacuruha chaelai que tonajelanaalane coina.</para> </listitem> <listitem> <para>Niha chu chaen, cachaauru rai rauhi, ita taojieraauru aina nenaa jereniha ne, cu... | ...asjonAvEnAksje- /B eregningAvM aksimaltTa psfradrag-grp-4166/ Skattemes sigFormuesverd iOgHistori skKostpris-grp-4167/ <brreg:sensitivitet type=Sensitiv/> <xforms:input ref=/Skjema/R... | ...(fedata)) + addspace(self.createFB (fedata)) + addspace (self. createModulation (fedata)) + self.createOrbPos (feraw) def createFrequency(self, fedata): frequency = fedata.get(frequency) if frequency: return str(frequency / 1000) return... | ...playbooks |- openstack-ansible | | | |- playbooks The variables in "my project/ custom stuff/ playbooks/ ansible.cfg" would use "../openstack-ansible/ playbooks/<directory> env.d    The "/etc/openstack deploy/env.d" directory ... | ...> <language type=ko>Isi-Korean </language> <language type= ku>Kurdish </language> <language type=ky>Kyrgyz</language> <language type=la> Isi-Latin </language> <language type=ln> Iilwimi</language> <language type=lo>IsiLoathian... |

Table 32: Raw training examples selected to have 14 quality ratings from 1 to 5 within Github.

| Language Consistency = 1 | Language Consistency = 2 | Language Consistency = 3 | Language Consistency = 4 | Language Consistency = 5 |
|---|---|---|---|---|
| ...jucesorInaf</value> <value>com.gs. fw.para. fufexoze. wuhoticuhomec. implementation. mithra. ResulEruyojiv OnuwufadobO yitilArub</value> <value>com.gs. fw.para.fufexoze. wuhoticuhomec. implementation. mithra. GoGabeVoluca sozeHoditoZuqi</va... | ...105/>Ortswechsel – Neues Thema des Monats!</a> </h3> <div class=block style=margin:0> Liebe piqs-User und Userinnen, und Userinnen und Userinnen
 
 ab diesem Monat erhält unser beliebtes Thema des Monats einen eige... | ...ur adipiscing elit. Integer posuere erat a ante.</p> </blockquote> <blockquote> <p>Lorem ipsum dolor sit amet, consectetur adipiscing elit. Integer posuere erat a ante.</p> Someone famous in <cite title=Source Tit... | ...package controller; /** * Created by tangzhijing on 2016/8/31. */ import android.app. Application; import android. content.Context; import android. content.Intent; import java.text .DateFormat ; import java. text. SimpleDateFormat; import java. util.Arr... | ...* @return Config prefix for leaf queue template configurations */ @Private public String getAutoCreat edQueueTempl ateCo nfPrefix(String queuePath) return queuePath + DOT + AUTO CREATED LEAF QUEUE TEMPLATE PREFIX; @Private public s... |

| Originality = 1 | Originality = 2 | Originality = 3 | Originality = 4 | Originality = 5 |
|---|---|---|---|---|
| ...as:RPS:i0151> RPSi0151.xml </policySetReference> <policySe Reference ref=urn:mtas: PPS:i0151 :manager>PPSi0151.xml </policySet Reference> <policySetReference ref=urn:mtas:PPS: i0151:employee> PPSi0151.xml </policySetReference> <policySetRefer... | ...Thelia Condition ConditionFactory</abbr> y</abbr></a> y</abbr></a> y</abbr></a> y</abbr></a> y</abbr></a> y</abbr></a> </td> <td> Manage how Condition could interact with the current application state (Thelia) </td> </tr> ... | ... <name>Daily Minimum Temperature</name> <value>73</value> <value>73</value> <value>73</value> <value>75</value> <value>73</value> <value>71</value> <value>70</value> <value>67</value> </temperature> </parameters> <pa... | ...s => s.Trim('/'))), Microsoft.Test. OData.Services. ODataWCFService. GetAccountInfo, true); /// <summary> /// There are no comments for RefreshDefaultPI in the schema. /// </summary> public global:: Microsoft.O... | ...// is returned. character set delimiterSet is used as word delimiters. // if the receiver is empty, an empty string is returned // // error-condtions: // if the receiver is nil, nil is returned – (NSString *)rest OfWordsUsing DelimitersFromSet:(NSC... |

| Professionalism = 1 | Professionalism = 2 | Professionalism = 3 | Professionalism = 4 | Professionalism = 5 |
|---|---|---|---|---|
| ...1UEBhMC R0IxGzAZBgN VBAgT EkdyZWF0ZX IgTWFuY2hlc3RlcjEQM A4GA1UEBxMHU2 FsZm9yZDEaMB gGA1UEChMR Q09NT0RPIENBIE xpbWl0ZWQxNj A0BgNVBAMTLUN PTU9ET yBSU0EgRG9tYW luIFZh bGlkYXR pb24gU2VjdXJlIFNl... | ... <div class=col-sm-12> <div class=main image > <div class=nm-imgs>Image Utama</div> class=nm-imgs>Image Utama</div> class=nm-imgs>Image Utama</div> class=nm-imgs>Image Utama</div> class=nm-imgs>Image Utama</div> <div class=ktk-imgs> <... | ...them on chains. The memory jewels were colored like rubies and emeralds, pearls and amethysts, and she had ropes of them in colors to suit every outfit. Some of the slights were Comanche's, but because Comanche wanted to be friends with Kathy she did... | ...from base import NagiosAuto import os class Host(NagiosAuto): This class have three options to create host file in nagios. You can specify the template you need. If you create a lots of host file at one time, this is more effecienc... | ...leave.s IL 010f async: resume IL 0096 ldarg.0 IL 0097: ldfld System.Runtime. CompilerServices. TaskAwaiter<bool> C.<Main>d 0.<>u 1 IL 009c: stloc.2 IL 009d: ldarg.0 IL 009e: ldflda System.Runtime.... |

| Semantic Density = 1 | Semantic Density = 2 | Semantic Density = 3 | Semantic Density = 4 | Semantic Density = 5 |
|---|---|---|---|---|
| ...span> 1792 1793 1794 1795 1796 1797 ... | ...Zulu-Natala</p> </td> <td> <p>ANC 1, DP 1, IFP 3, NP 1</p> </td> <td> <p>ANC 2, IFP 2</p> <p>ANC 2, IFP 2</p> <p>ANC 2, IFP 2</p> <p>ANC 2, IFP 2</p> <p>ANC 2, IFP 2</p> ... | ...span> LightGoldenrodYellow </li> <li style= background: LightGrey;>  LightGray</li></div> <div class=linkAHEAD><a href= background: LightGreen;>< span> LightGreen</li> <li style=b... | ...> <div class= linkAHEAD><a href= jcomponent.html>The JComponent Class</a> <div class=linkAHEAD><a href=text.html> Using Text Components</a></div> <div class=linkBHEAD><a href=generaltext.html> ... | ...language> <language type=tl> </language> <language type=tlh> </language> <language type=tli> </language> <language type=tmh> </language> <language type=tn></language> <language type=to>... |

| Sensitivity = 1 | Sensitivity = 2 | Sensitivity = 3 | Sensitivity = 4 | Sensitivity = 5 |
|---|---|---|---|---|
| ...986>said</mainVerb>.. <mainReferent begin=1936 end=1942> Inouye </mainReferent> <annotation annotator = gold seType=REPORT mainReferent Genericity= NON-GENERIC habituality=EPISODIC mainVerbAspectual Class =DYNAMIC/> <annotation annot... | <score>2.8</score> <enabled> Yes</enabled> </game> <game name= triplew1 index= image=> <description>Mahjong Triple Wars (Japan)</description> <cloneof /> <crc>11580513</crc> <manufacturer> Nichibutsu </manufacturer> ... | ...skillsofts, skillshots, skillshot, skillshots, skirmiches, skirmish, skpeticism, skepticism, slaughterd, slaughtered, slipperies, slippers, smartphone, smartphones, smartphone, snadwiches, sandwi... | ...hiç olmazsa arýlarý senden uzaklaþtýrayým, sana çok ýzdýrap veriyorlar. dedim. Onlar bana ýzdýrap verdikçe, benim halim daha hoþ oluyor. Ey Havvas! Sen benim çektiðim sýkýntýlarý, eþek arýlarýný boþver, sen tatlý nar yemek arzusunu kendinden uz... | .../glyphicons-halflings- regular.eot'), path.join(bs, 'dist/ fonts/ glyphicons -halflings-regular.svg'), path.join(bs, 'dist/fonts/ glyphicons-h alflings-regular.ttf'), path.join(bs, 'dist/fonts/ glyphicons- halflings-regular.woff'), ... |

Table 33: Raw training examples selected to have 14 quality ratings from 1 to 5 within Github.

| Structural Standardization = 1 | Structural Standardization = 2 | Structural Standardization = 3 | Structural Standardization = 4 | Structural Standardization = 5 |
|---|---|---|---|---|
| ...em>sarebbero </td ><td> </td><td> </td><td> </td></tr> <tr><td>  <a href=itparlamint -feat-Mood.html> Mood</a> =Ind \| <a href=itparlamint -feat-Number.html> Number</a>= Sing\| <a href= itparlamint -feat-Person.... | ...2 Blackmailed Busty Milfs Got Mercilessly Fucked By 2 Evil Teenage Boys><h3>2 Blackmailed Busty Milfs Got Mercilessly Fucked By 2 Evil Teenage Boys</h3></a> </div>  <div class=rating> <i id=galinfraa terramento2 value=NOK><i id=galinfraa terramento2 value=NOK><i id=galinfraat erramento2 value=NOK><i class=glyphicon glyphicon-remove text-danger> ... | ...para> Specifies a new value for a control's bound property obtained in the <see cref=E:DevExpress. XtraReports .UI.XRControl.E valuateBinding/> event handler. </para> </summary> <value>A <see cref=T:System.Object/>, specifyi... |

| Style Consistency = 1 | Style Consistency = 2 | Style Consistency = 3 | Style Consistency = 4 | Style Consistency = 5 |
|---|---|---|---|---|
| ...alments: installments, instals: installs, instil: instill, instils: instills, institutionalisation, institutionalization, institutionalise: institutionalize institutionalize institutionalize institutionalised: institutionalized, ... | ...2>available</dependent> </dep> <dep type=prep> <governor idx=22>available</governor> <dependent idx=23>to</dependent> </dep> <dep type=nn> <governor idx=25>members <governor idx=25>members <governor idx=25>members... | ...dent>case 1: info.efetuarLogin(); break; case 2: info. cadastrarCliente(); break; case 3: info. buscarNotebook(); break; case 4: info. manterCarrinho(); break; case 5: info. manterCarrinho(); break; ... | ...then let them ...</p> </div><!-- overlay-content -> </a><!- overlay -> </div> <div class=item last> <figure> </fig... | ...1] = ' 0'; else /* the 0 s already in the string */ crMemcpy (dataptr, string[i], pLocalLength [i]); else CRASSERT(pLocalLen CRASSERT(pLocalLen CRASSERT(pLocalLen CRASSERT(pLocalLen CRASSERT(pLocalLen... |

| Topic Focus = 1 | Topic Focus = 2 | Topic Focus = 3 | Topic Focus = 4 | Topic Focus = 5 |
|---|---|---|---|---|
| ...faces/ twitter /moynihan /128.jpg, https://s3.amazonaws. /uifaces/ faces/twitter/ danpliego/ 128.jpg, https://s3.amazonaws. /uifaces/ faces/twitter/ saulihirvi/ 128.jpg, https://s3.amazonaws. /uifaces/ faces/twitter/ wesleytrankin/ 128.jpg, ... | ...\|0 -20) + 0]] < [[@ Esquireage +? @ Esquirename age roll @ Esquireage modif. ? modif\|0 .com/button> </div> <div> <label style=' display: inline-block; width:130px;'> </label> <input style ='display :inline;' t... | ...hidden=true></li> <li><i class=fa fa-behance-square aria-hidden=true></li> <li><i class=fa fa- bitbucket aria-hidden=true></li> <li><i class=fa fa-bitbucket <li><i class=fa fa-bitbucket <li><i class=fa fa-bitbucket <li><i class=fa fa-bitbucket <li><i class=fa fa-bitbucket... | ...=hz>herero</language> <language type=ia> interlingua </language> <language type=iba>iban </language> <language type=ibb>ibibio</language> <language type=id> indonesiano </language> <language type=ie> interlingue </language> <la... | ...> x, y , 0, 0 , e.gPrOff( x, y )); p->b odystateanimations[ ItemWieldMode:: TwoHanded, HandState::AtEase, MovementState:: None> BodyState:: Kneeling, BodyState::Prone ][ x, y ] = e.getAnimationEntry (dataAD, dataUA, d... |

| Overall Score = 1 | Overall Score = 2 | Overall Score = 3 | Overall Score = 4 | Overall Score = 5 |
|---|---|---|---|---|
| ..., 36, 37, 38, 39, 40, 41, 42, 43, 44, 45, 46, 47, 48, 49, 50, 51, 52, 53, 54, 55, 56, 57, 58, 59, 60, 61, 62, 63, 64, 65, 66, 67, 68, 69, 70, 71, 72, 73, 74, 75, 76, 77, 78, 79, 80, 81, 82, 83, 84, 85, 86, 87, 88, 89, 90, 91, 92, 93, 94, 95, 96... | ...ional> 1 , 0 </pattern> </dateTimeFormat> </dateTime FormatLength> <dateTime FormatLength type=short> <dateTimeFormat> <pattern draft=provisional> 1 , 0 </pattern> </dateTime Format> </dateTime FormatLength>... | ...<p>Tudo péssoa tem drêto di entrâ na funçon pública di sé téra,ô di sé país.</p> </li> <li> <p>Vontadi di pôbo ê quel licérce di ôtoridadi di poder di público; ê debe mostráno-el co eleiçon sério qui ta bem ser fêto, na temp... | ...> </div> <div class=content> <h2 class=content-head is-center>Want to show off your coding skills?</h2> <div class=pure-g anchor id=events> <div class=l-box pure-u-1 pure-u-md-1-2 pure-u-lg-1-3> <h3 ... | ...namespace Bio.Algorithms.Alignment. MultipleSequenceAlignment using System; using System. Collections.Generic; using Bio; using Bio. Algorithms. Alignment; using SM = Bio. SimilarityMatrices. SimilarityMatrix; /// <summary> /... |

Table 34: Raw training examples selected to have 14 quality ratings from 1 to 5 within StackExchange.

| Accuracy = 1 | Accuracy = 2 | Accuracy = 3 | Accuracy = 4 | Accuracy = 5 |
|---|---|---|---|---|
| ...fdWGAXPxDj 4URMOnelcS jOpmAoCIB24mFvn C34fP8icL0Q VIOf+mPin2jD4Hg sDP58dDFtu ---END CERTIFICATE--- ---BEGIN CERTIFICATE--- MIICBDCCAaqg AwIBAgIQOOHnvua xK4NLP1+Qb7OIm+DAKBggmain(String[] args) qhkjOPQQDAjB lMQsw CQYDVQQGEwJ VUzETMBEGA1UECB... | ...Q: How can I match the original array and the shuffled array? This is the Prompt: This is what I wrote: import java.util.Scanner; public class ContagionControl public static void System.out.println( Please Enter ... | ...Q: How to write Object as human readable text file I want to write objects in human readable form in a text file, the file gets saved as a normal serialized object with unwanted characters instead. How do I rewrite the program for saving into human r... | ...Q: Access violation inside ctype imported windows function RtlDerive Capability SidsFromName I am importing ntdll.dll RtlDerive CapabilitySids FromName function and having access violation inside it. But can't figure what i am doing wrong. import win32fi... | ...3.0.0,>=2.11.2 in /home/ubuntu/anaconda3 /envs/myenv/lib/ python3.7/ site-packages (from app) (2.11.2) Requirement already satisfied: python-jose [cryptography] <4.0.0,>=3.1.0 in /home/ubuntu/ anaconda3/envs/myenv/ lib/python3.7/ site-packages (from app) (3...." |

| Coherence = 1 | Coherence = 2 | Coherence= 3 | Coherence = 4 | Coherence = 5 |
|---|---|---|---|---|
| ...7579 , 1525736201 , 1525736229 , 1525736426 , 1525736763 , 1525737360 , 1525736174 , 1525736723 , 1525736207 , 1525736839 , 1525737132 , 1525736460 , 1525737437 , 1525737070 ... | ...editorDtin ymc26imgsizeDlar ge6wplin3D 16urlbuttons ts list modelist; wp-settings-time-1= 1421309968; wp-saving- post=5-saved; wordpr - 524356 ReqHeader c Cookie: wp-settings-1 = deleteundefi526h... | ...272)|USER DEBUG| [14]|DEBUG| Domain: 2 18:56:23.1 (83147530)|USER DEBUG|[14]| DEBUG|Email: 2 18:56:23.1 (83166856)| USER DEBUG | [14]| DEBUG| Email 2: 2 18:56:23.1 (83205838)|USER DEBUG|[14]| DEBUG| Exec Type: 2 18:56:23.1 (83235539)|USER DEBUG|[14]|DEBUG|Exte... | ...Q: Why do i have this error on JSF page? I am newbie in jsf. I have a maven project and it runs on websphere 8. I use jsf and richfaces. I am getting this error: Error Parsing / viewMetadata/ index.xhtml: Error Traced[line: 2] The element type html ... | ...java.util.ArrayList; import java.util.List; import static android. Manifest.permission. READ_CONTACTS; /** * A login screen that offers login via email/password. */ public class LoginActivity extends AppCompatActivity implements LoaderCallbac..." |

| Creativity = 1 | Creativity = 2 | Creativity = 3 | Creativity = 4 | Creativity = 5 |
|---|---|---|---|---|
| ...Q: Error instantiating servlet class in Eclipse Europa? First.java : this is my first java class package com; import javax.servlet.Request Doiojpapchef; import javax.servlet. ServletException; import javax.servlet.http. Cookie; import javax.servlet. http.... | ...Q: Can This Review-Based Git Workflow Be Enforced by Gerrit? Is there a tool, that makes pull-requests and combined reviews safe in git? I know that there are a couple of related questions, that have already been asked at github (See a... | ...Q: How do I use wait() and notify correctly with threads in java? I'm making a 2D game in android studio and I've been sitting on a problem for a few days now. I want to stop my thread Gravity, so that my player can jump. When the player is done jump... | ... representing an event that happened in Paris in 1290. Legend is probably a better word than event, but in any case it is a very strange origin for a famous mathematical quote. A: Mathematics is the art of giving the same name to different thi... | ...a, svegliandosi una mattina da sogni agitati, si trovò trasformato, nel suo letto, in un enorme insetto immondo. Riposava sulla schiena, dura come una corazza, e sollevando un poco il capo vedeva il suo ventre arcuato, bruno e diviso in tanti segment... |

| Grammatical Diversity = 1 | Grammatical Diversity = 2 | Grammatical Diversity = 3 | Grammatical Diversity = 4 | Grammatical Diversity = 5 |
|---|---|---|---|---|
| ...(-90.5158851037139,,/ugvcHhgmk9x 38.5293438059525), 3IthyitanO5Vfk3 (-90.5153219032826, wRf7q5V366uuMhfcU' 38.5000608728918), , 'paper filed': (-90.571685370183, True, 'type': 38.5322450728324), 'SH06', 'pages': (-90.5907731028958, 8, 'barcode': 38.5035282063425), 'YBJEXN6Z', (-90.562387970228, 'transaction id': 38.5396736725824), 'MzM2MzUwMTYz (-90.5632127699527, OGFkaXF6a2N4' , 38.5000698058576), 'action date': (-90.4351277034801...'2022-10-17', 'category': p... | | ...0.0730668010171692, -0.0109530173729368, 0.0374915960907066, -0.0941194227900671, 0.0453306426927274, -0.173274373945029, 0.228535671136248, 0, 0.0923733553009261, -0.0400062320449435, 0.0101532578824621, -0.0204079876556867, 0.0648597665063123, ... | ...Q: JPA Spring ignores Lazy loading inside @Transacational I have a spring service class where I'm loading a JPA object (target) via CRUD. This target class has a one-to-may mapping that is set to lazy loading. I would like to query this object insid... | ...raw = T)5 -2.340e+06 3.246e+06 -0.721 0.480 poly(wt, 15, raw = T)6 8.537e+05 1.154e+06 0.740 0.468 poly(wt, 15, raw = T)7 -2.184e+05 2.880e+06 -0.758 0.458 poly(wt, 15, raw = T)8 3.809e+04 4.910e+04 0.776 0.447 poly(wt, 15... |

| Knowledge Novelty = 1 | Knowledge Novelty = 2 | Knowledge Novelty = 3 | Knowledge Novelty = 4 | Knowledge Novelty = 5 |
|---|---|---|---|---|
| ...2133OoqgH KKCRjmK1W/ YQFlYEI4yjsJ MVyP0MkrLKc538G1T //+QhljehHkOV1ciMk iEI4uo9NxzKHOUiD// /0IoAK8CJEgp0JdGYyK2aadmachine. It v7q1m2w10XDM7 /82DEiRwi3smeewo e5tODknVbVqXTOrY eKLLMuw6k KUKmfR4EzDJr/ HnvrWL4mjaxeZlh qmCGATTGej/ /DHdnRjTB2ZAr/... | ...Q: Bitonicsort C code segmentation issue I am running a bitonic sort sequential code on machine. It runs fine for array size upto 16 elements but as soon as i increase the size to 32 It gives the following error while execution: WARNING: Process ... | ...Q: xinput setting not persistent during session I use linux Manjaro Gnome X11. I like to have a special setting of mouse buttons, which I obtain with xinput. In order to make this setting persistent across sessions, I write the xinput in /.xprofile ... | ... while(srcWidth / 2 > desiredWidth) srcWidth /= 2; srcHeight /= 2; inSampleSize *= 2; float desiredScale = (float) desiredWidth / srcWidth; // Decode with inSampleSize options. inJustDecodeBounds = false; // now downl... | ...Q: Why do combinatorial abstractions of geometric objects behave so well? This question is inspired by a talk of June Huh from the recent Current Developments in Mathematics conference. Here are two examples of the kind... |

Table 35: Raw training examples selected to have 14 quality ratings from 1 to 5 within StackExchange.

| Language Consistency = 1 | Language Consistency = 2 | Language Consistency = 3 | Language Consistency = 4 | Language Consistency = 5 |
|---|---|---|---|---|
| ...327, -0.626763671898102, -0.297753001433115, 0.040555895564039, 0.564626235801996, 0.705799286675642, 0.0750613044187593, -0.478737190261705, -0.487080519650291, -0.570108362865829, -0.316410770808425, 0.641599306385284, 1.09437775518271, 0.677757969589283, -0.279... | ...Q: chamada de INTENT não encontrado O aplicativo delphi chama uma activity de aplicativo java e na chamada apresenta o seguinte erro: android. content. ActivityNot FoundException: Unable to find explicity activity class... | ...Sri Lanka</option> <option value= St Helena >St Helena</option> <option value= St Kitts and Nevis > St Kitts and Nevis</option> <option value= St Lucia >St Lucia</option> pedro. mostrapar com.embarcadero. pedro. com.embarcadero. pedro. mostrapar... | ...Q: Android AppBar Text Color Aren't Changing I can't change the color of android windows title bar text color. I don't know what is the name to this. So I'm providing you a screenshot to understand. please see the blow image: ScreenShot I searched fo... | ...Q: How to get GridPane Row and Column IDs on Mouse Entered in each cell of Grid in JavaFX? I am trying out JFX Drag and Drop feature in my application (later connecting the objects in a sequence). I could not find any easy way to drag and drop my Ima... |

| Originality = 1 | Originality = 2 | Originality= 3 | Originality = 4 | Originality = 5 |
|---|---|---|---|---|
| ...1334441447 8836548348450.*z))/ pow(4636667218 9358032.+18896234 711237580. *x- 3927118781169095.*y -147053464 16259850.*z,3) -(256.*(-350274353228 088978038961669 13833.+1011538246 7966920359 4026224002274.*x- 44334866794 1077090029000877418626 f.both: ?- slice([a,b,c,d,e,f,... | ...5, Xs = [c,d,e] ; Q = 6, Xs = [c,d,e,f] ; Q = 7, Xs = [c,d,e,f,g] ; Q = 8, Xs = [c,d,e,f,g,h] ; Q = 9, Xs = [c,d,e,f,g,h,i] ; Q = 10, Xs = [c,d,e,f,g,h,i,k] ; false. Last, we run a query which is a generalization | ...Q: ColumnLayout in Xamarin.Forms I think I found a bug in the FlexLayout. I've tried to nest 2 FlexLayouts where the outer one should be the column container and the inner one the row container. But the page stays empty. I've already filed a bug repo... | ...Q: DELETE FILES with Node.js I'm trying to delete some files and then show a message. EXPECTED OUTPUT File deleted Folder Cleared!!! ACTUAL OUTPUT Folder Cleared!!! File deleted The current code is: function clearConverted() const resp = ... | ...Q: SharePoint threw Unknown SQL Exception 206 occured. Anyone familiar with this? Our SharePoint instance threw the following errors when attempting to access data through a Content Query Tool: 04/02/2010 10:45:06.12 w3wp.exe (0x062C) ... |

| Professionalism = 1 | Professionalism = 2 | Professionalism = 3 | Professionalism = 4 | Professionalism = 5 |
|---|---|---|---|---|
| ...[192.5], [204.5], [154. ], [214.5], [205. ], [217. ], [201. ], [218. ], [169. ], [157.5], [194. ], [178.5], [194.5], [210.5], [219. ], [194.5], [194.5], [194.5], [194.5], [194.5], [194.5], [194.5], [194.5], [194.5], [194.5], [194.5], [194.5], ... | ...for replacing special arabic * Characters from the input given to the method. This method * Algorithm is taken from the database procedure already been * used for blacklist. * @param nameInArabic name in Arabic of applicant. E.g First nam... | ...Q: JAVAFX: Pass data between controllers I have an Excel file that i load it in tableView the data of this file i load in variable data, i want to pass this variable to another controller ControllerTwo to do some stuff on it but it returs a null p... | ...Q: Openshift Monitoring - cAdvisor + Prometheus - Docker I tried to implement a monitoring solution for Openshift cluster based on Prometheus + node-exporter + grafana + cAdvisor. I have a huge problem with cAdvisor component. I did a lot of configu... | ...start-1.4.0.jar file:/accumulo/ accumulo-1.4.0/ lib/commons-jci-fam-1.0. jar file:/accumulo/accumulo -1.4.0/lib/ jline-0.9.94.jar file:/accumulo/accumulo -1.4.0/lib/ examples-simple-1.4 examples-simple-1.4 I have a huge problem with cAdvisor component. I did a lot of configu... |

| Semantic Density = 1 | Semantic Density = 2 | Semantic Density = 3 | Semantic Density = 4 | Semantic Density = 5 |
|---|---|---|---|---|
| ... <td class= text-right text-nowrap > <button class= btn btn-xs btn-info >edit</button> <button class= btn btn-xs btn-warning >  </button> ... | ...= bottom align= left class=urLayou tPadless style= border- collapse:separate; width:100%; height:5px; white-space: normal; > <div id= WDEA-r > <table cellspacing = 0 cellpadding = 0 id= "WDEA ct= "ML lsdata = 0:'WDEA',7:'LINE' clas... | ...0.814441695 8.40741573499 6.08031313304 13.1781459953 10.206134367 15.0864695726 9.03013313987 4.46906993699 9.27542593922 11.387166818 5.34088290758 7.35790199406 11.8693581818 10.8557924873.... | ...Q: How to output the name of the calling method using XCGLogger? I have created a MyLogger class and it passes parameters to XCGLogger to output logs. I have specified true for the XCGLogger's showFileName and dateshowFunctionName, but it always outp... | ...Q: Error when trying to issue data manipulation statements with executeUpdate com.mysql.jdbc.exceptions. jdbc4.MySQLSyntaxErrorException: You have an error in your SQL syntax; check the manual that corresponds to your MySQL server version for the righ... |

| Sensitivity = 1 | Sensitivity = 2 | Sensitivity = 3 | Sensitivity = 4 | Sensitivity = 5 |
|---|---|---|---|---|
| ...", 'academic problems': ", 'describe problems': ", 'problems date': ", 'problems yn': ", 'end problems': ", 'disabilities': ", 'disability2': ", 'concerns': ", 'best things': ", 'too young': ", 'alcohol': ", 'describe alcl8yr': ", 'argue... | ...Q: What does play the trumpet mean? In a recent Academia SE question, user moonman239 writes: What is proper etiquette for college students needing to leave the lecture room for any reason? Example: Bathroom breaks, an urgent phone call, or a nee... | ...Q: Euphemism and Colloquialism as Literary/Speech Devices Is it possible for something to be both a 'euphemism' and a 'colloquialism'? If so, what would be some examples of this? A: Well, a lot of slang words (which are colloquial by definition), a... | ...: e035e200 esp: dff6af3c ds: 007b es: 007b ss: 0069 Process kjournald (pid: 314d, tic=dff6a000 tac=dffa8aa0 toe=dff6a000) Stack: 49276d20 6a757374 20737461 72746564 2c20646f 206e6f74 2070616e 69632079 65742c20 49206861 7665206e 6f742065 766... | ... 1.738037620 NA C3 -0.03510886 0.29100742 0.220716441 0.25246176 0.218140478 0.49141939 0.603698956 1.660365770 C4 0.25073995 -0.23513014 -0.217313407 -0.30890486 -0.217241734 -0.57995546.... |

Table 36: Raw training examples selected to have 14 quality ratings from 1 to 5 within StackExchange.

| Structural Standardization = 1 | Structural Standardization = 2 | Structural Standardization = 3 | Structural Standardization = 4 | Structural Standardization = 5 |
|---|---|---|---|---|
| ...IXEHnYfio RXfFGnLjwNadMT4 RRePM1lrtuARrY3042X bsCoY4GNjaNAFSe gMNAe+QtL ToMpH4gq6VlS6 P+y541UZOtE8GOl /Edme2xWPqr7qtsq XHzE /T9QjI5SVF5fC NSw/YdIQNkS6QGyAY 08oqMroNoMooGfQMJzW F/wBhSupWC5DRcjsp pChpHFc5Uco2ZF QNd/aFI cBh1/lt130C... | ...] uFE0F? u20E3\|[ u261D u270C u270D] uD83C[ uDFFB- uDFFF]\|[ u270A u270B](?: uD83C[ uDFFB- uDFFF])?\|[ u00A9 u00AE u203C u2049 u2122 u2139 u2194- u2199 u21A9 u21AA u2328 u23CF u23ED- u23EF u23F1 u23F2 u23F8- u23FA u24C2 u25AA u25AB u25B6 u25C0 u25FB u25... | ...Isengard instead of pausing to ask or infer who the travellers were. He was doubtful, fearful, and totally lacking in the skill of woodcraft. Gandalf confirms this. 'The victor would emerge stronger than either, and free from doubt,' said Gandalf. '... | ...Q: Get result from shell script objective-c I would like to run a shell script, from a file or from an objective-c string (within the code). I would also like the shell script's result to be stored to a variable. I would not like the shell script t... | ...Key.KP3 Keyboard.005c --> Key.KP4 Keyboard.005d --> Key.KP5 Keyboard.005e --> Key.KP6 Keyboard.005f --> Key.KP7 Keyboard.0060 --> Key.KP8 Keyboard.0061 --> Key.KP9 Keyboard.0062 --> Key.KP0 Keyboard.0063 --> Key.KPDot Keyboard.0064 --> Key.1... |

| Style Consistency = 1 | Style Consistency = 2 | Style Consistency = 3 | Style Consistency = 4 | Style Consistency = 5 |
|---|---|---|---|---|
| ...0C x1b x8f x1aC x00 x00 x00 x00 x00 x00 x00 x00 x00 x00 x00 x00 x00 x00 x00 x00 x00 x00 x00 x00 x00 x00 x00 x00 x80 x89A x03 x00 x00 x00 x06 x03 x00 x15t x84?P4 x00 x00 x00 x00 x00 x00 x00 xff x00 x01 x01 x... | ...n followers:[ key: jenny ,text: 'Jenny ank Hess',value: 'Jenny Hess' ], n repeats: , n reminders:[], n tags:[], n tasktags:[ id: 0, text: Thailand , id: 1, text: In... | ...Q: How to generate random numbers and distribute them randomly to some buttons? I am rookie in android programming and want to generate some numbers randomly in a specific range. Provided that, the sum of two of them equals a certain number, I wrote ... | ... Desktop larissa-node larissaApp rest-s erver node modules mongoose lib query.js:1394:10) at Function.findOne (C: Users Theodosios Desktop larissa-node larissaApp res t-server node modules mongoose lib model.js... | ...Q: Fragmented MP4 not playing in ffplay/QuickTime/Safari, but in VLC I encoded a fMP4-Video (HEVC) in Swift using VideoToolbox and CoreMedia. The resulting fragmented MP4 is only playing in VLC. The library I am using to write the fMP4 is an HEVC-ada... |

| Topic Focus = 1 | Topic Focus = 2 | Topic Focus = 3 | Topic Focus = 4 | Topic Focus = 5 |
|---|---|---|---|---|
| ...C5B4 uBCF4 uC558 uB2E4. uC815 uB9D0 uB3C4 uAE68 uBE44 uC5D0 uC11C uB098 uC628 uD558 uB098 uC758 uC2DC uBFD0 uB9CC uC544 uB2C8 uB77C uB098 uB098 uBA38 uC9C0 uC2DC uB4E4 uB3C4 uD558 uB098 uAC19 uC774 uBCF4 uAE30 uC88B uC740 uC2DC uB4E4, uADF8 ... | ... <nav class= navbar > <ul class= ul > <li class= textheader ><a class= logo >teleginsmm</a></li> <li class= textheader ><a></a></li> <li class= textheader ><a></a></li> ... | ...0): ['-2'], (21.0, 305.0, 9.0): ['-2'], (27.0, 310.0, 20.0): ['2'], (18.0, 303.0, 14.0): ['2'], (28.0, 293.0, 4.0): ['-2'], (29.0, 296.0, -2.0): ['2'], (23.0, 307.0, 19.0): ['-2'], (28.0, 294.0, 10.0): ['1'], (27.0, 293.0, 0.0): ['-2'], (33.0, 307.0,... | ...Q: My actionPerformed method(Java) is not working and I have no clue why Here is my whole program, don't wonder about the words I am using, I am German. Down from l. 95 to l. 103 is the action performed method, (I only did the System.out.println() to... | ...(-base-1); /* STYLE */ .container width: 1140px; margin: 0 auto; @media (max-width: 1200px) .container width: 960px; @media (max-width: 992px) .container width: 720px; @media (max-wi... |

| Overall Score = 1 | Overall Score = 2 | Overall Score = 3 | Overall Score = 4 | Overall Score = 5 |
|---|---|---|---|---|
| ...AB4AHgAd ABsAG AbABoAHgAaA BsAGwAbABw AHgAbABsA HgAeAB8A HgAeAB4AIAAgACAAH gAdABsAHAAg ACEAIAAgAC AAIQAjACIAIAAg ACAAHwAeA B0AGwAbABsA GgAZA BkAGQA VABQAFgA WABYAFAAUA BQAFgAbABoA GQAZA... | ...FinishModel> <d3p1:actualFinish> 25/09/2015 </d3p1:actualFinish> <d3p1:baseLineStart> 12/12/2014</d3p1: baseLineStart> </d3p1:BaseStart FinishModel> <d3p1:BaseStart FinishModel> <d3p1:actualFinish> 27/03/2015 </d3p1:actualFinish> <d3p1: baseLineStart>27/03/20... | ...Q: The code works but it lags when it works. It works with foreach and loop it When the code is working so laggy it would be very good so that it is not laggy when it works. How the code works: It searches the computer for a file is then when t.find... | ...Q: How to keep a local directory automatically synced with a remote, without latency issues? I develop a git-tracked codebase that has a lot of files. This code must be run on a remote machine. So every time I make a change locally, I must then sync ... | ...Q: Concurrent Queue in Java that only retains the last item of each child thread I have 1 main thread which starts up n child threads. Each of these child threads continually produce an new event and add it to the shared queue. That event represents ... |

Table 37: Raw training examples selected to have 14 quality ratings from 1 to 5 within Wikipedia.

| Accuracy = 1 | Accuracy = 2 | Accuracy = 3 | Accuracy = 4 | Accuracy = 5 |
|---|---|---|---|---|
| ...5) 1998 VR23\|\|\|\|10 1998\|\| (-)\|\|LINEAR \|- \| (44466) 1998 VT23\|\|\|\|10 1998\|\| (-)\|\|LINEAR \|- \| (44467) 1998 VU27\|\|\|\|10 1998\|\| (-)\|\|LINEAR \|- \| (44468) 1998 VH34\|\|\|\|11 ... | ...ie, a cărui mantie roșie este observată de cuplu încă. de când fetița se oprise anterior la brutărie pentru a cumpăra pâine și dulciuri pe drumul spre casa bunicii. Urmează Rapunzel, cu părul ei blond, pe lângă al cărei turn din pădure trece nevasta ... | ...see Ibataikoku), was appointed queen regnant of Japan. Himiko died in the 240s, and the next king of Japan was a male, but civil war broke out again, and the rebellion ended when another female, Taeyeo/Ichibayo (see Taiyo), became queen of Japan. In... | ...ond hadden. Het huis werd vervolgens verhuurd aan burgemeester van Sappemeer Henk Eikema, die echter al in 1927 plotseling overleed. Mogelijk werd het huis vanaf 1928 verhuurd aan Johannes Jurgens die vanaf dat jaar werkzaam was als griffier bij het ... | .... El único autor que, por su espíritu escapista y su optimismo infantil, logró escapar al tenebrismo de la época, fue el novelista Mór Jókai, autor de más de cien volúmenes de prosa de ficción (entre novelas y cuentos) en los que se respira un roman... |

| Coherence = 1 | Coherence = 2 | Coherence= 3 | Coherence = 4 | Coherence = 5 |
|---|---|---|---|---|
| ...James Middleton Archibald Miller William Miller Norman Craig Millman Laurence Minot Hugh Fitzgerald Moore Gerald Ewart Nash Ernest Edward Owen Augustus Paget Medley Parlee Laurence Pearson Geoffrey Pidcock Sydney Pope Frederick Powell T... | .... Aided Fergusa maic Roig. , 2006. Aided Guill meic Garbada ocus Aided Gairb Glinne Ríge (Les morts violentes de Guill Mac Carbada et Gairb Glinne Rige) Aided Laegairi Buadaig (La mort violente de Loegaire Buadach), trad. Guyonvarc'h, La mort violen... | ...c'est la Verrerie-cristallerie d'Arques qui devenue Arc International fera travailler directement et indirectement jusqu' plus de dans l'Audomarois, causant une mutation spatiale et paysag re de l'Audomarois, avec une forte p riurbanisation... | ...urbaine entre Saint-Omer et Arques. De nouvelles routes sont r guli rement construites ou largies (rocade, voie nouvelle de la vall e de l'Aa (VNVA), voies expresses Saint-Omer-Dunkerque et Boulogne-Saint-Orner, desserte par l'autoroute A26, ... | ...Paul Finch is an English author and scriptwriter. He began his writing career on the British television programme The Bill. His early scripts were for children's animation. He has written over 300 short stories which have appeared in magazines, such ... |

| Creativity = 1 | Creativity = 2 | Creativity = 3 | Creativity = 4 | Creativity = 5 |
|---|---|---|---|---|
| ...87231 - \|\| \|\| 30 de juliol, 2000 \|\| Socorro \|\| LINEAR \|- \| 87232 - \|\| \|\| 30 de juliol, 2000 \|\| Socorro \|\| LINEAR \|- \| 87233 - \|\| \|\| 30 de juliol, 2000 \|\| Socorro \|\| LINEAR \|- \| 87234 - \|\| \|\| 30 de juliol, 2000 \|\| Socorro \|\| LINEAR \|- \| 87235 - \|\| \|\| ol, 2000 \|\| Socorro \|\| LINEAR \|- \| 87235 - \|\| \| ... | ...Otto Arendt (* 10. Oktober 1854 in Berlin; † 28. April 1936 ebenda) war ein deutscher Publizist und freikonservativer Politiker. Leben Arendt stammte aus einer jüdischen Familie und konvertierte später zum Christentum. Nach dem Besuch des Gymnasium... | ...nească a Flacăra, è stata definita da Lovinescu come una serie di quadri della nostra antica esistenza dai toni arcaici , e da Ion Vianu come una storia pittoresca della Valacchia. Călinescu ha notato come, in molte delle sue poesie e in partico... | ...ta a metà battuta 12 insieme alla cadenza perfetta impone che il solista riprenda a suonare dopo una doverosa pausa. Segue un ponte di collegamento al secondo tema (misure 13- 19). Se il carattere del primo tema era molto cantabile, il secondo tema ... | ...Dahingehende Deutungen hat auch Rip Van Winkle, die andere in Amerika spielende Kurzgeschichte des Skizzenbuchs, erfahren, deren Protagonist in der Kolonialzeit in einen zwanzigjährigen Zauberschlaf fällt, den Unabhängigkeitskrieg. |

| Grammatical Diversity = 1 | Grammatical Diversity = 2 | Grammatical Diversity = 3 | Grammatical Diversity = 4 | Grammatical Diversity = 5 |
|---|---|---|---|---|
| ...\|\| - \|\| \|- style=background: \| 58 \|\| June 4 \|\| Mariners \|\| - \|\| \|\| \|\| -- \|\| \|\| - \|\| \|- style=background: \| 59 \|\| June 5 \|\| Cardinals \|\| - \|\| \|\| \|\| -- \|\| \|\| - \|\| \|- style=background: \| 60 \|\| June 6 \|\| Cardinals \|\| - \|\| \|\| \|\| -- \|\| \|\| - \|\| \|- style=... | ...al Stadium Stadio Artemio Franchi Ghelamco Arena Arubaans voetbalelftal Arubaans voetbalelftal (vrouwen) Ashton Gate Asian Football Confederation Norair Aslanyan Oussama Assaidi Fernando Astengo Aston Villa FC Asturisch voetbalelftal... | ...frique de l'Est, au Brésil, aux États-Unis et en Australie. Les deux espèces descendent de l'aurochs (Bos primigenius), dont le dernier représentant européen s'est éteint en 1627, alors que la sous-espèce ayant mené au zébu aurait disparu en Inde env... | ...endencia perpetua y neutralidad del estado. Las paredes de la fortaleza fueron derribadas y la guarnición prusiana fue retirada. Los visitantes famosos a Luxemburgo en el y el incluyeron al poeta alemán Goethe, a los escritores franceses Émile Zol... | ...Czerwiec 1976 – określenie nadane fali strajków i protestów, do których doszło w PRL pod koniec czerwca 1976, po ogłoszeniu przez rząd Piotra Jaroszewicza wprowadzenia drastycznych podwyżek cen urzędowych na niektóre artykuły konsumpcyjne. Przyczyny... |

| Knowledge Novelty = 1 | Knowledge Novelty = 2 | Knowledge Novelty = 3 | Knowledge Novelty = 4 | Knowledge Novelty = 5 |
|---|---|---|---|---|
| ...60 - \|\| \|\| 1 noiembrie 2000 \|\| Socorro \|\| LINEAR\|-\| 178761 - \|\| \|\| 19 noiembrie 2000 \|\| Socorro \|\| LINEAR\|-\| 178762 - \|\| \|\| 19 noiembrie 2000 \|\| Socorro \|\| LINEAR\|-\| 178763 - \|\| \|\| 19 noiembrie 2000 \|\| Socorro \|\| LINEAR\|-\| 178764 - \|\| \|\| ... | ...š 34' – J.H. Rossi 43' Spartak Moskwa – Valencia CF 0:3 (0:1) Angulo 6', Mista 71', Juan Sánchez 85' 3. kolejka 2 października 2002 r. Liverpool F.C. – Spartak Moskwa 5:0 (3:0) Heskey 7', 89', Cheyrou 15', Hyypia 28', Diao 81' Valencia CF – FC ... | ...Vongsa (r. 1638-1690), figlio di Ton Kham. Sotto il suo regno, Lan Xang conobbe il massimo splendore. Alla sua morte scoppiarono nuovamente i dissidi interni dell'aristocrazia Tian Thala (r. 1690-1695), primo ministro di Surigna Vongsa e ... | ...is also assumed to be some sort of non-trivial medium to which one can associate certain energy. This is because the concept of absolutely empty space contradicts the postulates of quantum mechanics. According to QFT, even in absence of real particles... | ...its duration. He purchased equipment and wrote software that allowed him to record and analyze heartbeats, and began studying his own heartbeat rhythms as well as those of friends and other musicians. After decades of study, Graves used some of the ... |

Table 38: Raw training examples selected to have 14 quality ratings from 1 to 5 within Wikipedia.

| Language Consistency = 1 | Language Consistency = 2 | Language Consistency = 3 | Language Consistency = 4 | Language Consistency = 5 |
|---|---|---|---|---|
| ...\|\| \| \| \| \|\|- style=font-size: 85%;\| Tudelano\| style=background :FFB0B0\|14*\| style= background: FFB0B0\|18*\|10 \|style='background :FFE4B5;\|13\|8\| 6\|style= background: cfffff\|4\|style=background :cfffff\|4\|style=.. | ...que el destí d'Ahsoka era ambigu i una mica obert encara que la seva dobladora Eckstein creia que el personatge era viu. En el capítol Un món entre mons, de la quarta temporada el destí d'Ahsoka es revela finalment. Ezra Bridger, que ha anat e... | ...í o rekord 7 vítězství v sezóně, ve které ale nezískal titul mistra světa. Stejně dopadl v roce 1984 a 1988 Alain Prost a v roce 2006 Michael Schumacher. V roce 2005 se Kimi Räikkönen podělil o rekord 10 nejrychlejších kol v sezóně. O rekord se dělí... | ...Mir iskousstva (en , « Le Monde de l'Art ») est une association d'artistes russes fondée en 1898 dans l'idée de prôner un renouveau pictural de l'art russe en synthétisant plusieurs formes artistiques dont le théâtre, la décoration et l'art du livre.... | ...moneta cartacea debba avere una controparte adeguata: la terra o altre attività produttive. L'idea di una moneta che prende la forma di biglietti bancari e sganciata da un metallo prezioso è molto moderna. Ai nostri giorni, l'oro e l'argento hanno co... |
| Originality = 1 | Originality = 2 | Originality = 3 | Originality = 4 | Originality = 5 |
| ...15643 – \|\| \|\| \|\| Goodricke-Pigott \|\| R. A. Tucker \|– 215644 – \|\| \|\| \|\| Goodricke-Pigott \|\| R. A. Tucker \|– 215645 – \|\| \|\| \|\| Kitt Peak \|\| Spacewatch \|– 215646 – \|\| \|\| \|\| Palomar \|\| NEAT \|– 215647 – \|\| \|\| \|\| Kitt Peak \|\| Spacewatch \|– 2... | ...\|– 552174 – \|\| \|\| 12 ottobre 2007 \|\| Mount Lemmon Survey \|– \| 552175 – \|\| \|\| 9 ottobre 2013 \|\| Mount Lemmon Survey \|– \| 552176 – \|\| \|\| 9 ottobre 2013 \|\| Mount Lemmon Survey \|– \| 552177 – \|\| \|\| 31 marzo 2008 \|\| Mount Lemmon Survey \|– \| 552178 – ... | ...Tropfest Arabia () is an extension of Tropfest, the world's largest short film festival. Tropfest began in 1993 as a screening for 200 people in a cafe in Sydney but has since become the largest platform for short films in the world. Tropfest Arabia... | ...squadra. Al bar il barista si avvicina a Brooke. La ragazza pensa che voglia un autografo, ma lui le dice che gli serve una firma per il conto e che non è il tipo da autografi specialmente se non sa a chi lo chiede. Lei gli dice chi è e lui si presen... | ...producir el guaro, bastante neutro y de alta pureza, con un ligero sabor dulce resultante del azúcar de la caña. Con este destilado final, añadiendo diversos ingredientes, se elabora también el Colorado y la Extraconcha, que obtienen al concluir el p... |
| Professionalism = 1 | Professionalism = 2 | Professionalism = 3 | Professionalism = 4 | Professionalism = 5 |
| ...2777 – \|\| \|\| 24 settembre 2000 \|\| LINEAR \|– \| 122778 – \|\| \|\| 24 settembre 2000 \|\| LINEAR \|– \| 122779 – \|\| \|\| 24 settembre 2000 \|\| LINEAR \|– \| 122780 – \|\| \|\| 24 settembre 2000 \|\| LINEAR \|– \| 122781 – \|\| \|\| 24 settembre 2000 \|\| LINEAR \|– \| 122782 ... | ..., va començar mostrant que aquestes millores eren temporals. Per exemple, un estat àrab va llançar un míssil nuclear, augmentant un conflicte a petita escala i fent que les potències mundials es rearmin, i els fons d'Alternative Earth van ser malvers... | ...arbres morts. C'était à l'origine une guilde légale mais qui acceptait (Eligoal surtout) des missions d'assassinat. Ces missions ont été déclarées interdites par le Conseil de la Magie, le maître d'Eisen Wald a été arrêté et mis en prison. La guilde ... | ...Carl Christian Berner (* 20. November 1841 in Christiania; † 25. Mai 1918 ebenda) war ein norwegischer Politiker. Leben Seine Eltern waren der Richter am Stiftsobergericht Oluf Steen Julius Berner (1809–1855) und dessen Frau Marie Louise Falkenberg... | ...inamen Tjan-Schanski erhielt, beschrieben, nachdem er 1856/57 die Gegend um den Yssykköl-See besuchte. Semjonow-Tjan-Schanski konnte beweisen, dass es selbst in den Trockenwüsten Asiens große Gebirgsgletscher gibt, was er und andere Wissenschaftler... |
| Semantic Density = 1 | Semantic Density = 2 | Semantic Density = 3 | Semantic Density = 4 | Semantic Density = 5 |
| ...– \|\| \|\| \|\| Spacewatch \|– \|398397 – \|\| \|\| \|\| Spacewatch \|– \|398398 – \|\| \|\| \|\| Mt. Lemmon Survey \|– \|398399 – \|\| \|\| \|\| Spacewatch \|– \|398400 – \|\| \|\| \|\| Spacewatch \| 398401–398500 \|– \|398401 – \|\| \|\| \|\| Spacewatch \|– \|398402 – \|\| \|\| \|\| ... | ...l – Erehof Kuinre – Erehof Vollenhove – Erehof Willemsoord – Espelo F Fanfare (film) – Fanny Blankers-Koen Stadion – FC Twente – Herman Finkers Friezenberg – G Gammelke – Ganzendiep – Gelderman – Genemuiden – Genne – Genne-Overwaters – Gesch... | ...Morse (1902) Morse (1905–1906) Morse (1910–1916) Morse (1914–1916) Motor Bob (1914) Motorette (1911–1914) Moyea (1903–1904) Moyer (1911–1915) M.P.M. (1914–1915) Mueller (1896–1899; also Mueller-Benz) Mulford (1915, 1922) Multiplex (1912–19... | ...Regionale delle Arti Figurative presso l'aula magna dell'Istituto Magistrale Statale S. Satta di Nuoro. Nel 1957 a Nuoro partecipa alla Biennale Nazionale di Pittura – Premio Sardegna, promossa dall'ente provinciale per il turismo. Nel 1958 espone... | ...Callejón sangriento (título original: Blood Alley) es una película estadounidense de 1955 dirigida por William A. Wellman y protagonizada por John Wayne y Lauren Bacall. Argumento La guerra civil en China ha terminado. Los comunistas han tomado el ... |
| Sensitivity = 1 | Sensitivity = 2 | Sensitivity = 3 | Sensitivity = 4 | Sensitivity = 5 |
| ...Confessions 13 (2000) Understudy (2000) Wet Dreams 7 (2000) Wet Dreams 8 (2000) When The Boyz Are Away The Girlz Will Play 1 (2000) When The Boyz Are Away The Girlz Will Play 2 (2000) White Panty Chronicles 16 (2000) X Girls (2000) Calendar I... | ...Warum begeht Helen Koch schweren Kraftwagendiebstahl? ist ein deutscher Kurzfilm unter der Regie von Moritz Geiser aus dem Jahr 2022. Seine Uraufführung feierte der Film am 21. Januar 2022 auf dem Filmfestival Max Ophüls Preis 2022... | ...estrale dei bianchi (la cosiddetta razza caucasica) abitanti della Scandinavia, con i capelli biondi e gli occhi azzurri. Charroux sostiene inoltre che questo popolo avesse avuto un'origine extraterrestre, proveniente da un... | ...Die Antisemitenliga war eine der ersten Vereinigungen zur Sammlung von Judengegnern im Deutschen Kaiserreich und die erste, die das Schlagwort Antisemitismus zum politischen Programm erhob. Sie wurde am 26. September... | ...), österreichischer Tischtennisspieler Koller, Heinrich (1924–2013), österreichischer Historiker Koller, Heinrich (* 1941), Schweizer Jurist, Rechtskonsulent und Direktor des Bundesamtes für Justiz ... |

Table 39: Raw training examples selected to have 14 quality ratings from 1 to 5 within Wikipedia.

| Structural Standardization = 1 | Structural Standardization = 2 | Structural Standardization = 3 | Structural Standardization = 4 | Structural Standardization = 5 |
|---|---|---|---|---|
| ...avait été étonnamment ouverte ce jour-là. De même que le tunnel rétractable jusqu'au milieu du terrain pour protéger les joueurs n'avait pas été déplié. Le , le président de la JSK, Mohand Chérif Hannachi déclare que selon les médecins du club, Albe... | ...type: Point, coordinates: [ –155.5988931655884, 18.970787529076187 ] , type: Feature, properties: , geometry: type: Point, coordinates: [ 115.1675319671631, –8.726969207892507 ] , type: Feature, properties: , geometry: type: Point, coordinates: [ 72.82279014587404, ... | ...ha un'allucinazione vedendo l'assassino mascherato in lontananza ma subito dopo corre tra le braccia di Kieran, i due si baciano e finiscono poi col fare l'amore nella macchina. Al loro ritorno la zia di Kieran comunica a quest'ultimo che continuerà ... | ...The pair always had intentions of bringing Christopher Reeve onto the show, and when they found out that he enjoyed watching the show himself Gough and Millar decided that they were going to bring him on for season two. They had already crafted a character, Dr. Virgil Swann, they knew would reveal... | ...La discographie de Sheryfa Luna, une chanteuse de RnB française, se compose de quatre albums studio, dix singles et dix clips vidéo. Albums Chansons Singles | class=wikitable style=text-align:center; |+ Liste des singles et positions dans le... |

| Style Consistency = 1 | Style Consistency = 2 | Style Consistency = 3 | Style Consistency = 4 | Style Consistency = 5 |
|---|---|---|---|---|
| ...1 |13,0 |11,5 |10,4 |164 |131 |269 |274 |3,70 |3,65 |11,4 |42,3 |1,22 |22 |--- |22 |0,6438 |25,4 |0,69 |27,0 |0,322 |642,4 |14,5 |13,3 |11,8 |207 |165 |341 |346 |2,93 |2,89 |18,3 |53,6 |0,965 |23 |--- |23 |0,5733 |22,6 |0,61 |24,1 |0,255 |509,5 |16... | ...kath., 280 ref., 18 zsidó lak. Ref. anyaszentegyház, kath. kápolna. Szőlőhegy. Erdő. 192 hold szántóföld. F. u. gr. Andrásy Gyula. Borovszky Samu monográfiasorozatának Zemplén vármegyét tárgyaló része szerint: Szőlőske, bodrogmenti magyar kisközsé... | ...ajda, Malta, 1831 (prestavljen 1893) Obelisk Thomasa Jeffersona v Monticellu, 1833 Obelisk levov, Iaşi, Romunija, 1834 Villa Torlonia, Rim – dva obeliska, 1842 Obelisk pokrajine Emilija v spomin na poroko Francesca V., vojvode Modene, in princese Ad... | ...have been recovered. (Location of scores for four other songs missing this time)) (Lyricist Sean Rafferty) These are not at the BL because Players' Theatre is a private club and was not censored. Four to the Bar (1961) Diedre was included in this,... | ... | 286209 – || || 18 ottobre 2001 || NEAT |– | 286210 – || || 19 ottobre 2001 || NEAT |– | 286211 – || || 16 ottobre 2001 || LINEAR |– | 286212 – || || 17 ottobre 2001 || LINEAR |– | 286213 – || || 17 ottobre 2001 || LINEAR |– | 286214 – || || ... |

| Topic Focus = 1 | Topic Focus = 2 | Topic Focus = 3 | Topic Focus = 4 | Topic Focus = 5 |
|---|---|---|---|---|
| ...(Darlington). Yupadee Kobkulboonsiri, 51, Thai-American artist and jewelry designer, COVID-19. James Mahoney, 62, American pulmonologist and internist, COVID-19. Mark McNamara, 60, American basketball player (Philadelphia 76ers, San Antonio Spurs, Lo... | ...und Hotels in Sri Lanka kamen mindestens 253 Menschen ums Leben, 485 weitere wurden verletzt. 2. Juni: Ermordung des CDU Politikers Walter Lübcke 9. Oktober: Anschlag in Halle (Saale) Kultur und Gesellschaft 6. Januar: 76. Verleihung der Golden ... | ...class=note | |– class=vcard | class=fn org | Northover (Glastonbury) | class=adr | Somerset | class=note | | class=note | |– class=vcard | class=fn org | North Owersby | class=adr | Lincolnshire | class=note | | class=note |... | ...Easy Virtue starring Ben Barnes, Jessica Biel, Kristin Scott Thomas and Colin Firth (among others) is due for release in Europe on 7 November 2008. Jobbins has also contributed to, Breast Wishes, a comedy musical about 'breasts, and the people who support them'... | ...Sheridan Jobbins (born 2 July 1960) is an Australian journalist, television presenter and screenwriter. Life and career Jobbins was born in Melbourne, Australia. She was educated at Ascham School, Edgecliff She is a third generation Australian film maker.... |

| Overall Score = 1 | Overall Score = 2 | Overall Score = 3 | Overall Score = 4 | Overall Score = 5 |
|---|---|---|---|---|
| ...2, 57, 72, 34, 73, 85, 35, 371, 59, 196, 81, 92, 191, 106, 273, 60, 394, 620, 270, 220, 106, 388, 287, 63, 3, 6, 191, 122, 43, 234, 400, 106, 290, 314, 47, 48, 81, 96, 26, 115, 92, 158, 191, 110, 77, 85, 197, 46, 10, 113, 140, 353, 48, 120, 106, 2, 6... | ...1 krog | 15 |– ! 10 | 8 | Raph | B de las Casas | Maserati 6CM | 38 | +2 kroga | 25 |– ! 11 | 6 | Armand Hug | Privatnik | 'Maserati 4CM | 33 | +7 krogov | 21 |– ! Ods | 18 | Clemente Biondetti | Alfa Corse | Alfa Romeo Tipo 308 | | | 4 |– ! Ods... | ...120101)||2003 FP5|| align=right|14,8|| align=right|3,028|| align=right|0,054|| align=right|8,07|| align=right|5,268 || MBA || 26. března 2003||Campo Imperatore||CINEOS |– |(120102)||2003 FU5|| align=right|15,1|| align=right|2,657|| align=right|0,047|| al... | ...5–2007) Adoum Younousmi, Acting Prime minister (2007) Delwa Kassiré Koumakoye, Prime minister (2007–2008) Youssouf Saleh Abbas, Prime minister (2008–2010) Emmanuel Nadingar, Prime minister (2010–2013) Djimrangar Dadnadji, Prime minister (2013) Kalzeu... | ...El término latino (en latín: Latini) hace referencia a una de las etnias de origen indoeuropeo y del grupo itálico que se asentaron a lo largo de la costa tirrénica del Latium, en Italia, en el curso del II milenioa.C., durante la Edad del Bronce. ... |

