# OpenReview forum: "DataMan: Data Manager for Pre-training Large Language Models"
_ICLR.cc/2025/Conference — ICLR 2025 Poster_

### Official Review · Reviewer_Pb2j · 2024-10-29

**Soundness:** 3
**Presentation:** 1
**Contribution:** 2
**Rating:** 5
**Confidence:** 2

**Summary:**

This paper proposes DataMan, a perplexity-driven data management system that annotates the quality and domain of data points, which could be used for many applications such as data selection in the pre-training stage. The authors proposed 14 data quality evaluation metrics and trained a model for data quality based on Qwen-2-1.5B with GPT-4o-turbo annotations. Utilizing the data quality annotations, the authors selected a subset of data and trained a Sheared-Llama-1.3B model from scratch. The model trained with the best overall score demonstrates superior ability in various downstream tasks compared with baseline methods. The authors also conducted various analyses of DataMan and revealed several interesting findings.

**Strengths:**

1. The proposed system DataMan is overall sound for assessing data quality and this is an important research problem in the current research community.
2. The annotated data is an expensive but valuable resource for the community.
3. The experiments, along with analyses, are solid and demonstrate the great performance of DataMan.

**Weaknesses:**

1. The presentation of this paper needs significant improvement. I can find typos almost everywhere. I just list a few here. In line 193 "minimizing"; A missing space in line 241; Figure 1 caption is too close to the figure;  line 110 "require"; line 234 incorrect citation format; line 420, "quating";
2. Some claims made in this paper are not well supported. The one that mostly concerns me is that the authors claim "previous works are intuitive and lack generalizability". However, the 14 quality criteria seem also intuitive and manually designed by the authors.

**Questions:**

1. In lines 231-235, the authors claim that pointwise assessment is better than pairwise one by presenting results in Table 5. I don't quite understand how content in Table 5 could be related to this. Moreover, I don't think handpicked results could really support this claim.
2. It would be better if the authors could write how they picked these 13 quality criteria and all the domain categories. For example, the authors might provide some statistics on the outlier causes mentioned in lines 92-94, and how the iterative refinement is done. Is it done manually? By student or expert?
3. In line 285, what is the value of $k$ in top-k?
4. In lines 201-203, can you explain how the designed 14 quality criteria correspond to the three principles listed here?

---

> ### Author Response · Authors · 2024-11-30
> **Author Response to Reviewer Pb2j (Part I)**
>
> We sincerely appreciate your constructive comments, which are incredibly helpful in strengthening our paper! We hope that our response effectively addresses your concerns.
>
> **Weakness1. The presentation needs significant improvement.**
>
> All typos have been corrected and highlighted in the revised PDF.
>
> **Weakness2. The 14 quality criteria seem also intuitive and manually designed by the authors.**
>
> 1. The 14 quality criteria are **derived from the LLM's identification of the causes of abnormal perplexity (PPL) in documents**, **not through intuitive or manual design**.
>
>     Similar to previous self-improvement work [1,2], our motivation is to **enable LLM to self-select quality criteria that can enhance its performance**, as its **pre-training capability is related to perplexity** [3,4,5]. High perplexity indicates that the data is difficult to learn, and vice versa. Consequently, this **"reverse thinking"** approach encourages us to **let the LLM self-identify the causes of anomalous perplexity to derive quality criteria**, **rather than relying solely on intuitive or manual design**.
>
> 2. The **"iterative refinement"** described in the paper **refers to experts organizing and arranging all the quality criteria identified by the LLM, assisted by GPT**.
> 3. **Reviewers 8xFT** concur that **our approach makes sense**, and **Reviewers Tq6k agree** that **it avoids pitfalls of criteria based on human intuition**.
>
> **Questions2. How to select these 13 quality criteria and all the domain categories?**
>
> We will provide a detailed explanation of the process for obtaining all quality criteria and domain **categories**, which is conducted by experts with the assistance of GPT-4. The original text can be found in Appendix A.
>
> 1. **Motivation:** By employing a "reverse thinking" approach, we aim to encourage the LLM to self-identify quality criteria that contribute to its performance improvement. As noted in Weekness2, the pre-training capabilities of LLMs are closely related to perplexity[1,2,3]. Therefore, we prompt the LLM to identify the reasons behind document perplexity anomalies, which subsequently leads to the derivation of the current 13 quality criteria.
> 2. **Step 1: Initialization of Quality Criteria.**
> We devised an analytical prompt for GPT-4-turbo to investigate the reasons behind the textual perplexity anomalies (in the top and bottom 2%) from each source and extract initial text quality metrics. This yields the quality criteria for the first step. Table 1 provides **several examples illustrating how these initial quality criteria are derived**.
>
>    -  **Analytical prompt:**
>
>     ```
>     Read the following document, which has a {high/low} perplexity of {ppl_value} for LLM inference. Please analyze the reasons for the PPL anomaly.
>     ```
>
>     Table 1. Analyzing **the reasons for the abnormal perplexity in documents** using the analytical prompt in GPT-4-turbo, and **extracting the initial quality criteria** from the LLM's response.
>
> | **PPL** | **Text** | **LLM's Response** |
> | --- | --- | --- |
> | 1.19 | March: 1.77 inches- April: 5.31 inches- May: 5.9 inches- June: 5.9 inches- July: 5.9 inches- August: 2.36 inches- September: 5.31 inches- October: 5.9 inches- November: 2.36 inches- December: 2.36 inches2009, Senneville QC Precipitation: 56- January: 1.68 inches- February: 2.24 inches- March: 2.24 inches- April: 5.04 inches- May: 5.6 inches- June: 5.6 inches- July: 4.48 inches- August: 2.24 inches- September: 5.6 inches- October: 4.48 inches- November: 2.8 inches- December: 2.24 inches2008, Senneville QC Precipitation ... | Structural consistency: Data is presented in a fixed format. (**[[4]Content Regularity/Formatting)** Redundancy: A large number of repeating elements. **([5]Content Redundancy)** Lack of complex syntax: The text lacks complex sentence structures. **([10]Complexity of Text Structure)** Minimal context dependence. (**[1]Semantic Fluency/Coherence/Logic)** |
> | 1.07 | 4But the wise took oil in their vessels with their lamps. 5While the bridegroom tarried, they all slumbered and slept. 6And at midnight there was a cry made, Behold, the bridegroom cometh; go ye out to meet him. 7Then all those virgins arose, and trimmed their lamps. 8And the foolish said unto the wise, Give us of your oil; for our lamps are gone out. 9But the wise answered, saying, Not so; lest there be not enough for us and you: but go ye rather to them that sell, and buy for yourselves. ... | Textual Standardization: The text is an excerpt from the Bible. (**[3]Topic Diversity)** Structural and Redundant: The narrative has structure and redundancy. (**[4]Content Regularity/Formatting, [5]Content Redundancy)** Contextual Consistency: The parts of the text have internal coherence. (**[2]Content Consistency/Variability in Language Style)** Lexical and Grammatical Consistency: The Bible use fixed vocabulary and grammatical structures. **([10]Complexity of Text Structure)** |

---

> > ### Author Response · Authors · 2024-11-30
> > **Author Response to Reviewer Pb2j (Part II)**
> >
> > | **PPL** | **Text** | **LLM's Response** |
> > | --- | --- | --- |
> > | 185.1 | *be seeking 0111 ways to be cared for by otbem. An rugent--bas,~d c~li:bic Hmt lakes ca,ring as, ils nwst 011ernrching mol:ival:ional ideal avoids. thfa prolbl!ern, since ii: ent11ils that pei*sorn, who are conll'nt 1:hrou.ghmit their lifetime to simply receive cm\"<\" from others. without acknowied!ging anid/ or recipmrnbing thrnt rnre 1in any way are jm:tnfo11bly criticized fo1r exhibitirng; a d!eplornble latck of otlr1er-regaird, and in particufar, a fad. of enga.gem.,~11t with and gratilude tow,ird the s.perifir persons who crnntribule to their Hves itl varim1s ways. Nod!dings's own view Is tbait it takes s1uprisingly little effort to disclluuge our duties to receive or | Complex Content: The text deeply discusses philosophy and ethics. Terminology Dense: Understanding requires in-depth domain knowledge.**([6] Proportion of Domain-Specific Vocabulary)** Incoherent Text: Multiple parts show incoherence. (**[3]Topic Diversity)** Random Characters: There may be encoding errors. **([12]Proportion of Grammatical, Reference, and Spelling Errors)** |
> > | 167.5 | The Indian Express is now on Telegram. Click here to join our channel (@indianexpress) and stay updated with the latest headlinesFor all the latest Kolkata News, download Indian Express App. Mönchengladbach Michael Cuisance ist vorzeitig wieder ins Training eingestiegen. Er ist fit und will in der neuen Saison mehr spielen als in der abgelaufenen. Kurzum: Der 19-Jährige will sich bei Borussia Mönchengladbach durchsetzen.Michael Cuisance ärgert sich über das frühe Aus mit Frankreich bei der U20-WM. Foto: AP/Darko VojinovicCuisance startet bei Borussia seinen Angriff auf die StartelfbGüte rwued mi tnaShcte des eh-IFtnrteoslann uaf med wudh,eelgcaNhsäcn da erd eansR eds tatlztiHnagrnispeuaps chontges reenwd lsle. … | Mixed Languages: A mix of various languages and spelling systems.**([9]Degree of Language Mixing)** Incoherent Sentences: Poor connectivity between sentences. (**[2]Content Consistency/Variability in Language Style)** Ambiguous Context: Lack of clear contextual information.**([13]Proportion of Content Lacking Semantics)** Formatting Errors: Contains numerous formatting issues, typos, and garbled text.**([12]Proportion of Grammatical, Reference, and Spelling Errors)** |
> > | 72.19 | *property on sand\n\nin the autumn\n\nrinsing the token and ridding it of dispiritedness a cold\n\nfoam bath and icy smiles the lips permitted and the knight\n\nwas refreshed then also the mountains had reviving airs\n\nnesting in the hollows from peak to golden peak and there were\n\nlairs for robbers which altered the loneliness\n\nfor always he wished for his swan even its shadow even\n\na shadow on stones that once were heavy and warm\n\nand the swan's story\n\nhe cherished along with the memory of his ride that as a basin\n\nis filled then emptied yet its curve remains and its depth\n\nhe would never forget nor the exact three quarter of his waltz\n\nthere on the fringes of clouds and the embrace in the chasm\n\nwith snow and down erasing any doubt the act was modern\n\nas mountain climbing or looking for gentians\n\nany ruin has its surplus or wrinkle there on the mask…* | Text content is complex: The text seems to be part of a series of poems, filled with metaphors and abstract ideas. (**[1]Semantic Fluency/Coherence/Logic)** Irregular format and structure: Poetry often has a different format and structure than regular prose, including line breaks and unusual punctuation. **([10]Complexity of Text Structure)** Proper nouns and archaisms: The text may contain proper nouns, archaisms, or words coined by the author. **([6] Proportion of Domain-Specific Vocabulary)** Creative expression in text: Because poetry often contains a lot of creative expression and experimental use of language. **([8]Proportion of Creative Expression)** |
> > | 1.91 | a former 1970s antiwar radical fugitive wanted for murder. After living for more than 30 years underground, Grant must now go on the run. With the FBI in hot pursuit, he sets off on a cross-country journey to track down the one person that can clear his name.\nShepard knows the significance of the national news story he has exposed and, for a journalist, this is an opportunity of a lifetime. Hell-bent on making a name for himself, he is willing to stop at nothing to capitalize on it. He digs deep into Grant’s past. Despite warnings from his editor and threats from the FBI, Shepard relentlessly tracks Grant across the country… | As an AI model, I can't talk about topics involving politics, religion, pornography, violence, etc. **([7]Proportion of Sensitive Topics)** |

---

> ### Author Response · Authors · 2024-11-30
> **Author Response to Reviewer Pb2j (Part III)**
>
> **PPL** | **Text** | **LLM's Response** |
> | --- | --- | --- |
> | 118.75 | 75@dsocz7lp qar\|\|-:-q-,-g- - c-(d-o 75@dsocz7lp qar\|\|-:-q-,-g- - c-(d-o 75@dsocz7lp qar\|\|-:-q-,-g- - c-(d-o 75@dsocz7lp qar\|\|-:-q-,-g- - c-(d-o -),y-1-1p-1z-7n-8p-9d, ar\|\|-:-q-,-g- - c-(d-o ar\|\|-:-q-,-g- - c-(d-o ar\|\|-:-q-,-g- - c-(d-o ar\|\|-:-q-,-g- - c-(d-o ar\|\|-:-q-,-g- - c-(d-o| Non-standard characters and symbols: the text contains a large number of words, such as "@"," @"," =", ";" **([13]Proportion of Content Lacking Semantics)** Garbled characters and random strings: occurrences in the text similar to "dsocz7l, 'p qar\|\|-!"   Such garbled or random strings are unpredictable to the language model because they are meaningless in normal language representation. **([13]Proportion of Content Lacking Semantics)** |
>
> - - **Initial Quality Criteria**
>
>        ```
>        [1]Semantic Fluency/Coherence/Logic: Evaluate whether the text is smooth and easy to read, whether the content is coherent, and whether the logic is clear.
>        [2]Content Consistency/Variability in Language Style: Evaluate if the information within the text is contradictory and if the language style is diverse.
>        [3]Topic Diversity: Determine the richness and variety of topics addressed in the text.
>        [4]Content Regularity/Formatting: Consider whether the text adheres to a certain structure or format.
>        [5]Content Redundancy: Analyze the extent of information repetition within the text.
>        [6]Proportion of Domain-Specific Vocabulary: Measure the frequency of professional terms or specific vocabulary used in the text (such as proper nouns, technical terms, or Classical Chinese).
>        [7]Proportion of Sensitive Topics: Examine the percentage of content that addresses sensitive topics (e.g., involving politics, toxicity).
>        [8]Proportion of Creative Expression: Assess the degree of creative or innovative expression in the text (e.g., use of rhetorical techniques).
>        [9]Degree of Language Mixing: Analyze the extent to which different languages are used within the text (i.e., the ratio of text in various languages).
>        [10]Complexity of Text Structure: Evaluate the overall complexity of the text's structure.
>        [11]Proportion of Long Sentences: Assess the ratio of long sentences within the text.
>        [12]Proportion of Grammatical, Reference, and Spelling Errors: Evaluate the ratio of grammatical errors (e.g., incorrect punctuation, unclear sentence breaks), reference errors, and spelling mistakes in the text.
>        [13]Proportion of Content Lacking Semantics: Determine the ratio of parts within the text that lack meaningful content (e.g., garbled text, HTML tags, XML elements, navigation bars, incomplete chart numbers, or disjointed citations).
>        ```
>
> 3. **Step 2: Enhance Quality Criteria.**
>
>     a. We utilize GPT-4-turbo to rank the text quality criteria by importance, eliminating, merging, and supplementing them.
>
>        [1] Text accuracy (grammar, references, spelling)
>        [2] Semantic coherence and consistency
>        [3] Language consistency
>        [4] Effective semantic content ratio
>        [5] Knowledge novelty
>        [6] Topic focus
>        [7] Creative expression ratio
>        [8] Proportion of technical terms
>        [9] Style variability
>        [10] Complexity of grammatical structures
>        [11] Content regularity
>        [12] Content redundancy
>        [13] Proportion of sensitive topics
>
>     b. Thereafter, based on the principle that higher scores indicate better criteria, we revised the prompt. Finally, we also utilized GPT-4-turbo to help us simplify, resulting in 14 quality criteria derived from the original paper.

---

> ### Author Response · Authors · 2024-11-30
> **Author Response to Reviewer Pb2j (Part IV)**
>
> ```
> [1]Accuracy: the fewer grammar, referential, and spelling errors the text contains, and the more accurate its expression. _/5
> [2]Coherence: the more fluent the content is expressed, and the stronger its logical coherence. _/5
> [3]Language Consistency: the more consistent the use of language in the text, with less mixing of languages. _/5
> [4]Semantic Density: the greater the proportion of valid information in the text, with less irrelevant or redundant information. _/5
> [5]Knowledge Novelty: the more novel and cutting-edge the knowledge provided by the text, with more insightful views on the industry or topic. _/5
> [6]Topic Focus: the more the text content focuses on the topic, with less deviation from the main theme. _/5
> [7]Creativity: the more creative elements are shown in the text’s expression. _/5
> [8]Professionalism: the more professional terminology appears in the text, with more accurate use of terms and more professional domain-specific expression. _/5
> [9]Style Consistency: the more consistent the style of the text, with proper and appropriate style transitions. _/5
> [10]Grammatical Diversity: the more varied and correct the grammatical structures used in the text, showing a richer language expression ability. _/5
> [11]Structural Standardization: the clearer the structure followed by the text and the more standardized its format. _/5
> [12]Originality: the fewer repetitions and similar content in the text. _/5
> [13]Sensitivity: the more appropriately sensitive topics are handled in the text, with less inappropriate content. /5
> [14]Overall Score: the better the comprehensive evaluation of the text, with superior performance in all aspects./5
> ```
> 4. **Step 3: Identify Domain Categories.**
>
>     We selected 15 current domain categories based on factors such as the application industries, the number of existing industry LLMs, and the level of attention they have received (e.g., GitHub stars) [1,2].
>
>   [1] https://github.com/HqWu-HITCS/Awesome-Chinese-LLM
>
>   [2] Naveed, Humza, et al. "A comprehensive overview of large language models." *arXiv preprint arXiv:2307.06435* (2023).
>
> **Questions1. Why do the handpicked results in Table 5 support "Pointwise ratings are better than Pairwise ratings"?**
>
> Table 5: We follow the Tabel 4 in Qurating by using 10 documents from different sources, ranked them by writing style, and use them to analyze pointwise and pairwise ratings
>
> | **Rank** | **Text**  | **Source Description** | **DataMan’s 13 Quality Criteria**  | **DataMan’s Overall Score and Domain** |
> | --- | --- | --- | --- | --- |
> | 1 | Amory Blaine inherited from his mother every trait, except the stray inexpressible few, that made him worth while. His father, an ineffectual, inarticulate man with a taste for Byron and a habit of drowsing over the Encyclopedia Britannica, grew wealthy at thirty through the death of two elder brothers, successful Chicago brokers, and in the first flush of feeling that the world was his, went to Bar Harbor and met Beatrice O’Hara. In consequence, Stephen Blaine handed down to posterity his height of … | F. Scott Fitzgerald’s This Side of Paradise | accuracy: 5 coherence: 4 language_consistency: 5 semantic_density: 4 knowledge_novelty: 2 topic_focus: 5 creativity: 4 professionalism: 3 style_consistency: 5 grammatical_diversity: 4 structural_standardization: 3 originality: 5 sensitivity: 5 | overall_score: 4 domain: culture |
> | 2 | Technologies for making and manipulating DNA have enabled advances in biology ever since the discovery of the DNA double helix. But introducing site-specific modifications in the genomes of cells and organisms remained elusive. Early approaches relied on the principle of site-specific recognition of DNA sequences by oligonucleotides, small molecules, or self-splicing introns. More recently, the site-directed zinc finger nucleases (ZFNs) and TAL effector nucleases (TALENs) using the principle of site-specific … | CRISPR-Cas9 paper abstract  | accuracy: 5 coherence: 5 language_consistency: 5 semantic_density: 5 knowledge_novelty: 4 topic_focus: 5 creativity: 3 professionalism: 5 style_consistency: 5 grammatical_diversity: 5 structural_standardization: 4 originality: 5 sensitivity: 5 | overall_score: 5 domain:technology |

---

> > ### Author Response · Authors · 2024-11-30
> > **Author Response to Reviewer Pb2j (Part V)**
> >
> > | **Rank** | **Text**  | **Source Description** | **DataMan’s 13 Quality Criteria**  | **DataMan’s Overall Score and Domain** |
> > | --- | --- | --- | --- | --- |
> > | 3 | The winter of 1906-07 was the coldest in Alberta’s history and was exacerbated by a shortage of coal. One cause of this shortage was the strained relationship between coal miners and mine operators in the province. At the beginning of April 1907, the Canada West Coal and Coke Company locked out the miners from its mine near Taber. The same company was also facing a work stoppage at its mine in the Crow’s Nest Pass, where miners were refusing to sign a new contract. The problem spread until by April … | featured Wikipedia article | accuracy: 5 coherence: 5 language_consistency: 5 semantic_density: 5 knowledge_novelty: 3 topic_focus: 5 creativity: 3 professionalism: 4 style_consistency: 5 grammatical_diversity: 4 structural_standardization: 4 originality: 5 sensitivity: 5 | overall_score: 4 domain: other |
> > | 4 | On December 3, Venezuela held a controversial referendum over a claim to the oil-rich Essequibo region controlled by Guyana. That same day, the Vice President of Venezuela, Delcy Rodríguez, shared a video on X, formerly Twitter, showing a group of Indigenous people lowering a Guyanese flag and hoisting a Venezuelan flag in its stead over the territory, which is also known as Guayana Esequiba. ’Glory to the brave people!’ she wrote, which is the first line of the country’s national anthem. The post came … | Bellingcat news article | accuracy: 5 coherence: 5 language_consistency: 5 semantic_density: 4 knowledge_novelty: 3 topic_focus: 5 creativity: 3 professionalism: 4 style_consistency: 5 grammatical_diversity: 4 structural_standardization: 4 originality: 5 sensitivity: 5 | overall_score: 4 domain: government |
> > | 5 | The Godfather is one of the most praised movies in cinema history. It gives everything that critics and audiences alike ask for in movies. In my opinion it gets all the attention it gets for being one of, or the best movies ever. One of the best things The Godfather does is its incredible casting and its iconic performances from each and every one of its characters. The actors are so convincing that it won the movie several academy awards. It also jumpstarted several actors, acting careers, and gave an … | IMDb movie review | accuracy: 4 coherence: 4 language_consistency: 5 semantic_density: 4 knowledge_novelty: 3 topic_focus: 5 creativity: 4 professionalism: 3 style_consistency: 4 grammatical_diversity: 4 structural_standardization: 3 originality:  5 sensitivity: 5 | overall_score: 4 domain: entertainment |
> > | 6 | The food is good, but not a great value. Up front, I will just say, do not waste your time getting traditional sushi here because tbh it’s not really that much better. For example, we ordered some maki and nigiri and while it was good, it wasn’t that much better than our fave sushi places. Instead, come here for their signature dishes and you’ll probably be happier. We really enjoyed some of their signature dishes. We dined as a party of 4 and we had: Spicy edamame: tasty and spicy! Yellowtail … | yelp restaurant review | accuracy: 4 coherence: 4 language_consistency: 5 semantic_density: 4 knowledge_novelty: 2 topic_focus: 5 creativity: 3 professionalism: 2 style_consistency: 4 grammatical_diversity: 3 structural_standardization: 3 originality: 4 sensitivity: 5 | overall_score: 4 domain: other |
> > | 7 | My Father worked for a Forbes 500 company since the 70s. Moved up the ranks as a software engineer and management, has patents for the company that saved it millions of dollars. He’s almost to pension age and suddenly HR starts making his life miserable. He noticed this trend was happening to some of his coworkers when they were getting close to age 60 as well. HR Lady calls him into the office and says that he was not punching in and out at the correct time. My Father, an engineer, is very very … | reddit post | accuracy: 4 coherence: 4 language_consistency: 5 semantic_density: 4 knowledge_novelty: 3 topic_focus: 4 creativity: 3 professionalism: 4 style_consistency: 4 grammatical_diversity: 4 structural_standardization: 3 originality: 5 sensitivity: 5 | overall_score: 4 domain: technology |

---

> > > ### Author Response · Authors · 2024-11-30
> > > **Author Response to Reviewer Pb2j (Part VI)**
> > >
> > > | **Rank** | **Text**  | **Source Description** | **DataMan’s 13 Quality Criteria**  | **DataMan’s Overall Score and Domain** |
> > > | --- | --- | --- | --- | --- |
> > > | 8 | THE ADVENTURE OF LINA AND HER ADVENTUROUS DOG SHERU Lina was a normal girl like any girl.She lived in the hills.She went to the top of the hills and she looked behind a special bush under the rearest of pine trees.She saw many pines behind it,but when she moved the pines she found a large piece of paper in which something was writen.Lina, Lina said her mother.GET UP!!You’re late for school!!Oh mom!I’m too tired.Come on you have to go,no arguements.Lina was from a rich family.She lived in Los Anjilous … | childhood composition by friend of author | accuracy: 2 coherence: 3 language_consistency: 4 semantic_density: 3 knowledge_novelty: 1 topic_focus: 4 creativity: 3 professionalism: 1 style_consistency: 3 grammatical_diversity: 2 structural_standardization: 2 originality: 4 sensitivity: 5 | overall_score: 2 domain: other |
> > > | 9 | "Sunshine Quiz Wkly Q! Win a top Sony DVD player if u know which country the Algarve is in? Txt ansr to 82277. Â£1.50 SP: Tyrone Customer service annoncement. You have a New Years delivery waiting for you. Please call 07046744435 now to arrange delivery You are a winner U have been specially selected 2 receive Â£1000 cash or a 4* holiday (flights inc) speak to a live operator 2 claim 0871277810810 URGENT! We are trying to contact you. Last weekends draw shows that you have won a Â£900 prize … | concatenated spam messages | accuracy: 2 coherence: 3 language_consistency: 2 semantic_density: 3 knowledge_novelty: 1 topic_focus: 4 creativity: 2 professionalism: 2 style_consistency: 2 grammatical_diversity: 2 structural_standardization: 2 originality: 3  sensitivity: 5 | overall_score: 2 domain: retail e-commerce |
> > > | 10 | cRjp7tQcwHoNERPRhj7HbiDuessoBAkl8uM0GMr3u8QsHfyGaK7x0vC3L0YGGLA7Gh240 GKhDjNwoaBtQubP8tbwrKJCSmRkUbg9aHzOQA4SLWbKcEVAiTfcQ68eQtnIF1IhOoQXLM 7RlSHBCqibUCY3Rd0ODHSvgiuMduMDLPwcOxxHCCc7yoQxXRr3qNJuROnWSuEHX5WkwNR Sef5ssqSPXauLOB95CcnWGwblooLGelodhlLEUGI5HeECFkfvtNBgNsn5En628MrUyyFh rqnuFNKiKkXA61oqaGe1zrO3cD0ttidD … | randomly generated alphanumeric string | accuracy: 1 coherence: 1 language_consistency: 1 semantic_density: 1 knowledge_novelty: 1 topic_focus: 1 creativity: 1 professionalism: 1 style_consistency: 1 grammatical_diversity: 1 structural_standardization: 1 originality: 1 sensitivity: 5 | overall_score: 1 domain: other |
> > >
> > > 1. First, **the cases in Table 5 are sourced from Qurating's Table 4 and were not handpicked by us**. Qurating selected 10 documents from various sources, ranked by the writing style, to test an LLM's nuanced judgment capabilities.
> > > 2. Next, we use Table 5 to analyze pointwise and pairwise ratings, with additional document sources supplemented.
> > >     1. **Pairwise rating faces the challenge of minimal quality differences, while pointwise rating aligns with human judgment.**
> > >
> > >         Table 5 shows that the top three documents have only slight quality differences in "writing style", which evan humans find hard to distinguish (similarly seen in DataMan's "Overall Score.”). This challenges pairwise rating: how does one choose between documents pairs when quality differences are negligible? Generally, if documents meet an "acceptable" quality threshold, they should be accepted. In addressing this challenge, pointwise rating aligns well with human judgment.
> > >
> > >     2. Revisiting Qurating's original paper on **the use of pairwise evaluation and its issues — API calls, computational costs, and quality difference requirements.**
> > >
> > >         Appendix B.1 shows that constructing the Quarter's SFT dataset requires GPT-3.5-turbo to **make 40 predictions for a a criterion and document pair**, using the likelihood of the LLM preference as a label. Table 5 illustrates that **document pairs needed at least a 50% selection probability difference in quality criterion**, establishing a clear quality differential limit for SFT document pairs. Moreover, for N documents, **pointwise rating requires only N assessments**, whereas **pairwise rating necessitates N*(N-1) comparisons, with each comparison requiring 40 predictions**. Thus, when SFT evaluators, API calls, computational costs, and quality difference requirements for document pairs in pairwise ratings are significantly higher than those for pointwise ratings.Aslo, Qurating's Table 6 reveals that **for inference,** Quarter **demands at least a 50% quality difference in criteria for document pairs**, raising **concerns about its practice applicability**.
> > >
> > > 3. Finally, we recognize that pointwise and pairwise ratings capture different essences. However, **in both SFT evaluators or trillion-token scale pretraining**, **the applicability and cost-effectiveness outweigh the minor rating performance differences**, leading us to choose pointwise rating over pairwise rating.

---

> ### Author Response · Authors · 2024-11-30
> **Author Response to Reviewer Pb2j (Part VII)**
>
> **Questions3. What is the value of k in top-k?**
>
> ```
> Line 285: "We perform top-k sampling without replacement based on source and domain distribution for each quality criterion."
> ```
>
> Here, **top-k refers to** **the size of the pre-training data sample**, which is typically 30B tokens in this paper.
>
> **Questions4. How do the 14 quality criteria correspond to the 3 principles?**
>
> ```
> Lines 201- 203: "How to define the quality criteria and domains of texts? We believe that excellent quality criteria should: 1)-be applicable to a wide variety of texts, 2)-demonstrate a deep understanding of content, capturing semantic levels, and 3)-complement each other. "
> ```
>
> 1. First, we want to clarify that these three principles **are not to generate quality criteria; instead,** the **quality criteria should satisfy these three principles**.
> 2. Following this, we illustrate how the 14 quality **criteria** satisfy three principles:
>     1. All quality criteria are satisfied **by principle 1**;
>     2. *Coherence, Semantic Density, Knowledge Novelty, Topic Focus, Creativity, Professionalism, Originality, Sensitivity* **satisfy principle 2**;
>     3. *Accuracy, Language Consistency, Style Consistency, Grammatical Diversity, Structural Standardization*, these **non-semantic level criteria complement with semantic level criteria** that meet principle 2.
>     4. As stated in **the response to Reviewer8xFT's Weakness1**, all **quality criteria complement each other** to form a comprehensive data quality system **across four levels.**

---

> ### Comment · Reviewer_Pb2j · 2024-12-01
> **Thanks for the response**
>
> I appreciate the authors' efforts during the rebuttal phase, especially the detailed experiments and thorough explanations. Most of my concerns have been addressed, and I have updated my score accordingly. However, due to the current presentation of the paper and the absence of an updated PDF version, I am inclined to maintain my score at 5 (rather than increasing it to 6). I have also adjusted my confidence score to allow the AC and other reviewers to take a more prominent role in the decision-making process.

---

> > ### Author Response · Authors · 2024-12-01
> > **Thanks for your prompt feedback**
> >
> > We are heartfeltly grateful for your recognition of our efforts and are glad to see that our rebuttal has addressed most of your concerns.
> >
> > **We need to clarify that we have already submitted an updated PDF version**. However, due to the **ICLR PDF checking mechanism not highlighting the changes** compared to the original PDF, some misunderstandings may have arisen. We sincerely apologize for any confusion this may cause.
> >
> > Notably, we **greatly value the concerns you raised regarding the presentation of our paper** (these can improve the paper's completeness! 👍 ), and we **made targeted revisions in the uploaded PDF version** as follows:
> >
> > 1. (Entire Paper) We have corrected all typos, such as grammatical and spelling mistakes, missing spaces, incorrect citation formats, and improper layout.
> > 2. (Introduction) We have revised the claim to clarify that the “14 quality criteria were not intuitive and manually designed”.
> > 3. (Appendix A & Table 4,5) We have elaborated on the process of obtaining all quality criteria and domain categories and updated the examples in Table 4,5.
> > 4. (Section 4 & Line 277) We have explained the value of k in the top-k sampling by "We perform top-k sampling (k is the selected subset size) without replacement ...".
> > 5. (Appendix A & Table 6) We have modified the table caption to clarify that the examples in Table 5 are sourced from Qurating's Table 4, not handpicked by us.
> > 6. Regarding the misunderstanding between pointwise and pairwise ratings, and clarification on how the 14 quality criteria correspond to the 3 principles, we initially planned to include both responses in the updated PDF version once we received the reviewers' approval.
> >
> > However, the **PDF revision deadline** (November 27th AOE) **has passed**. Therefore, we **commit to further updating our final paper version to incorporate all valuable suggestions** gained from the rebuttal discussion.
> >
> > We humbly seek to ascertain whether our responses adequately address your concerns. We would be **extremely grateful if you would reconsider your scores to support our wor**k 🙏. Thank you once again!
> >
> > Sincerely,
> >
> > The Authors

---

### Official Review · Reviewer_JcNL · 2024-11-02

**Soundness:** 3
**Presentation:** 3
**Contribution:** 3
**Rating:** 6
**Confidence:** 4

**Summary:**

This paper addresses the need for comprehensive data quality criteria for pre-training large language models (LLMs). It introduces "DataMan," a tool that evaluates text quality and domain types using 13 quality criteria related to LLM performance. The paper demonstrates the effectiveness of these criteria by fine-tuning LLMs on a subset of the SlimPajama corpus, showing performance improvements over state-of-the-art benchmarks. Contributions include the establishment of quality evaluation criteria, the creation of the DataMan tool, and the release of models and annotated datasets to facilitate further research on the relationship between data quality and LLMs.

**Strengths:**

- The paper offers a comprehensive set of quality criteria for evaluating pre-training data, which helps improve LLM performance.
- Though not comprehensive, the approach of leveraging LLMs to self-identify anomalies in perplexity is an effective method for deriving quality metrics.
- The release of annotated datasets and models is a valuable contribution to the community, promoting further research and development.

**Weaknesses:**

- The dependence on perplexity as a primary measure for quality assessment may limit the generalizability of the approach. While perplexity is an indicator of understanding, it may not fully capture other essential aspects of data quality, such as creativity or sensitivity. It is also bounded by the knowledge and capabilities of the LLMs that are used to compute perplexity.
- It fixates comparing with Wettig et al.,2024 and lacks comparison with other related work.

**Questions:**

- How does this method compare with other methods (e.g., deduplication, heuristic-based selection, domain mixture, other LLM quality signals) mentioned in the related work?
- How does this method compare with these large open-source efforts described in the following two papers?
  - Dolma: an Open Corpus of Three Trillion Tokens for Language Model Pretraining Research [[paper](https://arxiv.org/abs/2402.00159)] [[code](https://github.com/allenai/dolma)]
  - The FineWeb Datasets: Decanting the Web for the Finest Text Data at Scale [[paper](https://arxiv.org/abs/2406.17557)] [[code](https://github.com/huggingface/datatrove/)]

---

> ### Author Response · Authors · 2024-11-30
> **Author Response to Reviewer JcNL (Part I)**
>
> We would like to thank for your thoughtful and insightful reply! We sincerely hope our response can resolve your concerns.
>
> **Weakness1. Relying on perplexity for quality assessment may limit generalizability and overlook essential data aspects**.
>
> 1. DataMan **assesses data quality using criteria derived from GPT-4-turbo's analysis of abnormal perplexity causes in documents**, **rather than using perplexity directly.** This method helps mitigate the limitations of LLM's knowledge and capabilities in calculating perplexity.
> 2. **DataMan's 14 quality criteria**, detailed in Appendix A, **include creativity and sensitivity.**
>     ```
>     [7]Creativity: the more creative elements are shown in the text’s expression. _/5
>     [13]Sensitivity: the more appropriately sensitive topics are handled in the text, with less inappropriate content. _/5
>     ```
>
> **Weakness2 & Question1,2. Lack of comparison with related work**.
>
> 1. Apart from Qurating, we **also compared** **uniform sampling**, data selection with importance resampling (**DSIR**) [1], and **perplexity filtering** with the lowest/highest perplexity scores [2,3].
> 2. It is also worth mentioning that **DataMan is compatible and complementary to existing related works** (e.g., deduplication, heuristic-based selection, domain mixture, and other LLM quality signals).
>
> 3. Following the reviewer's advice, we **compared the latest FineWeb project** to the strongest variant of existing methods (including DataMan's Overall Score l=5.0). We used FineWeb code to filter the SlimPajama dataset, uniformly sampled 30B tokens, and trained Sheared-Llama-1.3B from scratch. We then presented comparative results on three metrics: ten in-context learning tasks and the perplexity of the SlimPajama validation and test sets.
>
>     Our conclusions are follows:
>     - **DataMan significantly outperforms FineWeb** on all three evaluation metrics.
>
>     - FineWeb performs similarly to Qurating in ten in-context learning tasks, likely due to its internal use of multiple quality filters.
>
>     - FineWeb has poor perplexity performance on SlimPajama validation and test sets, performing only on par with DSIR.
>
> Table 1. Results for **ten in-context learning tasks, the perplexity of the SlimPajama validation and test sets**, bold indicates the best performance.
>
> | **Selection Method** |  | **Val Perplexity** | **Test Perplexity** | **Reading Comprehension (5 tasks)** | **Commonsense Reasoning (3 tasks)** | **World Knowledge (2 tasks)** | **Average (10 tasks)** |
> | --- | --- | --- | --- | --- | --- | --- | --- |
> | Uniform |  | 10.70 | 10.75 | 50.9 | 55.0 | 14.9 | 44.9 |
> | DSIR [1] | with Wiki | 13.34 ↑2.64 | 13.37 ↑2.62 | 50.1 ↓0.8 | 49.8 ↓5.2 | 14.7 ↓0.2 | 42.9 ↓2.0 |
> | DSIR [1] | with Book | 13.60 ↑2.90 | 13.59 ↑2.84 | 47.9 ↓3.0 | 56.6 ↑1.6 | 14.1 ↓0.8 | 43.8 ↓1.1 |
> | Perplexity [2,3] | lowest | 15.98 ↑5.28 | 16.04 ↑5.29 | 48.3 ↓2.6 | 49.6 ↓5.4 | 13.7 ↓1.2 | 41.7 ↓3.2 |
> | Perplexity [2,3] | highest | 11.32 ↑0.62 | 11.34 ↑0.59 | 49.6 ↓1.3 | 53.5 ↓1.5 | 13.4 ↓1.5 | 43.5 ↓1.4 |
> | Educational Value [4] | τ=2.0 | 10.67 ↓0.03 | 10.72 ↓0.03 | 53.3 ↑2.4 | 56.3 ↑1.3 | 15.7 ↑0.8 | 46.7 ↑1.8 |
> | Fineweb [5] |  | 13.72 ↑3.02 | 13.55 ↑2.8 | 52.7 ↑1.8 | 56.6 ↑1.6 | 15.4 ↑0.5 | 46.4 ↑1.5 |
> | Overall Score | l=5.0 | **10.52 ↓0.18** | **10.50 ↓0.25** | **55.2 ↑4.3** | **60.2 ↑5.2** | **17.4 ↑2.5** | **49.1 ↑4.2** |
>
> Table 2. Detailed results for **ten in-context learning tasks**, bold indicates the best performance.
>
> |  |  | **Reading Comprehension** |  |  |  |  | **Commonsense Reasoning** |  |  | **World Knowledge** |  |  |
> | --- | --- | --- | --- | --- | --- | --- | --- | --- | --- | --- | --- | --- |
> | **Selection Method**  |  | ARC-E (15) | ARC-C (15) | SciQ (2)  | LogiQA (2) | BoolQ (0) | HellaSw. (6)  | PIQA (6)  | W.G. (15) | NQ (10) | MMLU (5) | Average |
> | Uniform |  | 57.5 | 27.6 | 87.7 | 24.1 | 57.5 | 44.0 | 68.6 | 52.5 | 4.1 | 25.7 | 44.9 |
> | DSIR [1] | with Wiki | 52.8↓4.7 | 26.3↓1.3 | 85.9↓1.8 | 25.2↑1.1 | 60.3↑2.8 | 35.8↓8.2 | 61.4↓7.2 | 52.2↓0.3 | 4.7↑0.6 | 24.7↓1.0 | 42.9↓2.0 |
> | DSIR [1] | with Book | 49.5↓8.0 | 25.3↓2.3 | 83.6↓4.1 | 23.5↓0.6 | 57.9↑0.4 | 44.8↑0.8 | 69.4↑0.8 | 55.6↑3.1 | 3.1↓1.0 | 25.2↓0.5 | 43.8↓1.1 |
> | Perplexity [2,3] | lowest | 49.2↓8.3 | 25.1↓2.5 | 83.7↓4.0 | 22.0↓2.1 | 61.4↑3.9 | 34.6↓9.4 | 65.0↓3.6 | 49.1↓3.4 | 2.7↓1.4 | 24.7↓1.0 | 41.7↓3.2 |
> | Perplexity [2,3] | highest | 53.5↓4.0 | 25.6↓2.0 | 84.6↓3.1 | **26.1↑2.0** | 58.0↑0.5 | 41.6↓2.4 | 65.6↓3.0 | 53.4↑0.9 | 2.9↓1.2 | 24.0↓1.7 | 43.5↓1.4 |
> | Educational Value [4] | top-k | **66.6↑9.1** | **34.6↑7.0** | 89.6↑1.9 | 24.6↑0.5 | 58.3↑0.8 | 45.5↑1.5 | 66.4↓2.2 | 52.9↑0.4 | 3.8↓0.3 | 25.0↓0.7 | 46.7↑1.8 |
> | Fineweb [5] |  | 60.5↑3.0 | 29.0↑1.4 | 87.9↑0.2 | 24.0↓0.1 | **62.0↑4.5** | 47.0↑3.0 | 69.3↑0.7 | 53.6↑1.1 | 4.7↑0.6 | 26.0↑0.3 | 46.4↑1.5 |
> | Overall Score  | l=5.0 | 66.1↑8.6 | 34.0↑6.4 | **90.7↑3.0** | **26.1↑2.0** | 59.2↑1.7 | **51.5↑6.5** | **70.7↑2.1** | **58.3↑5.8** | **7.8↑3.7** | **26.9↑1.2** | **49.1↑1.6** |

---

> ### Author Response · Authors · 2024-11-30
> **Author Response to Reviewer JcNL (Part II)**
>
> Table 3. Detailed results for **the perplexity of the SlimPajama validation sets**, bold indicates the best performance.
>
> | **Selection Method**  |  | **CC** | **C4** | **Github** | **Wiki** | **ArXiv** | **StackEx** | **Book** | **Overall** |
> | --- | --- | --- | --- | --- | --- | --- | --- | --- | --- |
> | Uniform |  | 11.09 | **13.93** | 3.04 | 10.41 | 5.69 | 6.15 | 12.42 | 10.70 |
> | DSIR [1] | with Wiki | 13.02 ↑1.93 | 18.66 ↑4.73 | 3.62 ↑0.58 | 24.07 ↑13.66 | 6.63 ↑0.94 | 7.28 ↑1.13 | 15.39 ↑2.97 | 13.34 ↑2.64 |
> | DSIR [1] | with Book | 13.11 ↑2.02 | 18.16 ↑4.23 | 3.50 ↑0.46 | 38.97 ↑28.56 | 6.55 ↑0.86 | 6.83 ↑0.68 | 13.18 ↑0.76 | 13.60 ↑2.90 |
> | Perplexity [2,3] | lowest | 16.20 ↑5.11 | 21.51 ↑7.58 | 4.41 ↑1.37 | 18.26 ↑7.85 | 7.12 ↑1.43 | 9.10 ↑2.95 | 20.26 ↑7.84 | 15.98 ↑5.28 |
> | Perplexity [2,3] | highest | 11.92 ↑0.83 | 14.34 ↑0.41 | 3.21 ↑0.17 | 11.38 ↑0.97 | 5.90 ↑0.21 | 6.20 ↑0.05 | 12.38 ↓0.04 | 11.32 ↑0.62 |
> | Educational Value [4] | τ=2.0 | 11.02 ↑0.03 | 14.10 ↑0.17 | 2.98 ↓0.06 | 10.49 ↑0.08 | 5.53 ↓0.16 | 6.09 ↓0.06 | 12.34 ↓0.08 | 10.67 ↓0.03 |
> | Fineweb [5] |  | 11.63 ↑0.54 | 17.51 ↑3.58 | 9.26 ↑6.22 | 27.55 ↑17.14 | 22.27 ↑16.58 | 13.40 ↑7.25 | 12.88 ↑0.46 | 13.72 ↑3.02 |
> | Overall Score | l=5.0 | **10.56 ↓0.53** | 16.36 ↑2.43 | **2.66 ↓0.38** | **8.51 ↓1.90** | **4.74 ↓0.95** | **5.92 ↓0.23** | **10.79** ↓1.63 | **10.52 ↓0.18** |
>
> Table 4. Detailed results for **the perplexity of the SlimPajama test sets**, bold indicates the best performance.
>
> | **Selection Method**  |  | **CC** | **C4** | **Github** | **Wiki** | **ArXiv** | **StackEx** | **Book** | **Overall** |
> | --- | --- | --- | --- | --- | --- | --- | --- | --- | --- |
> | Uniform |  | 11.10 | **13.82** | 2.97 | 10.29 | 5.26 | 5.75 | 13.05 | 10.75 |
> | DSIR [1] | with Wiki | 12.97 ↑1.87 | 18.50 ↑4.68 | 3.50 ↑0.53 | 23.91 ↑13.62 | 6.71 ↑1.45 | 6.36 ↑0.61 | 16.11 ↑3.06 | 13.37 ↑2.62 |
> | DSIR [1] | with Book | 13.08 ↑1.98 | 17.96 ↑4.14 | 3.41 ↑0.44 | 39.03 ↑28.74 | 6.62 ↑1.36 | 5.93 ↑0.18 | 13.38 ↑0.33 | 13.59 ↑2.84 |
> | Perplexity [2,3] | lowest | 16.15 ↑5.05 | 21.34 ↑7.52 | 4.21 ↑1.24 | 18.14 ↑7.85 | 7.21 ↑1.95 | 7.43 ↑1.68 | 21.51 ↑8.46 | 16.04 ↑5.29 |
> | Perplexity [2,3] | highest | 11.91 ↑0.81 | 14.27 ↑0.45 | 3.14 ↑0.17 | 11.24 ↑0.95 | 5.96 ↑0.70 | 5.34 ↓0.41 | 12.74 ↓0.31 | 11.34 ↑0.59 |
> | Educational Value [4] | τ=2.0 | 11.03 ↑0.03 | 13.97 ↑0.15 | 2.91 ↓0.06 | 10.36 ↑0.07 | 5.58 ↑0.32 | 5.17 ↓0.58 | 12.97 ↓0.08 | 10.72 ↓0.03 |
> | Fineweb [5] |  | 11.39 ↑0.29 | 17.29 ↑3.47 | 9.06 ↑6.09 | 27.58 ↑17.29 | 23.34 ↑18.08 | 12.33 ↑6.58 | 13.17 ↑0.12 | 13.55 ↑2.8 |
> | Overall Score | l=5.0 | **10.51 ↓0.59** | 16.18 ↑2.36 | **2.56 ↓0.41** | **8.40 ↓1.89** | **4.77 ↓0.49** | **4.99 ↓0.76** | **11.28 ↓1.77** | **10.50 ↓0.25** |
>
> 4. We further explain **why we did not conduct Dolma experiments**:
>
>     i. FineWeb is a more recent release than Dolma.
>
>     ii. Dolma conducts specific processing rules for each data source (Web, Code, Social, C4, Semantic Scholar, Books, Wikipedia) but lacks a comprehensive approach for overall data.
>
>     iii. Some rules described in Dolma’s paper, such as C4 NoPunc, are not implemented in Dolma's code.
>
> [1] Sang Michael Xie, et al. "Data selection for language models via importance resampling". Neurips 2023.
>
> [2] Wenzek, Guillaume, et al. "CCNet: Extracting high quality monolingual datasets from web crawl data." LREC 2020.
>
> [3] Marion, Max, et al. "When less is more: Investigating data pruning for pretraining llms at scale." Neurips 2023.
>
> [4] Alexander Wettig, et al. "Qurating: Selecting high-quality data for training language models". ICML 2024.
>
> [5] Penedo, Guilherme, et al. "The fineweb datasets: Decanting the web for the finest text data at scale." NeurIPS 2024 Track Datasets and Benchmarks.

---

> > ### Comment · Reviewer_JcNL · 2024-12-03
> >
> > Thank you authors for the detailed responses. They are very helpful and I hope they will be incorporated into the final version. I am confidently keeping my score unchanged.

---

> > > ### Author Response · Authors · 2024-12-03
> > > **Very appreciate your encouragement.**
> > >
> > > Thank you for your encouragement. We are happy to hear that our response was very helpful. In the final version of the paper, we will further refine it to incorporate the valuable insights gained from the rebuttal discussion. Once again, thank you!
> > >
> > > Sincerely,
> > >
> > > The Authors

---

### Official Review · Reviewer_ViAr · 2024-11-04

**Soundness:** 3
**Presentation:** 2
**Contribution:** 3
**Rating:** 5
**Confidence:** 3

**Summary:**

This paper presents DataMan, a Qwen2-1.5B fine-tuned to perform pointwise evaluation across 14 quality criteria on a 356K dataset annotated by GPT-4-Turbo. The authors then use this model to create the DataPajama dataset from annotating a 447B tokens subset of the SlimPajama dataset. Pre-training from scratch on a subset of DataPajama show improvements compared to other baselines.

**Strengths:**

- Use LLMs for filtering pretrained data as opposed to using heuristics.
- Promise to release code, models and datasets, which would benefit the research community especially given the cost of obtaining such models and datasets.

**Weaknesses:**

- **Novelty**: [1] already proposed to train a model to rate and select high-quality pre-training data based on four qualities: "writing style", "required expertise", "facts & trivia", and "educational value". The main differences seem to be on the considered criteria, the model used (Qwen2-1.5B vs Sheared-Llama-1.3B) and using pointwise instead of pairwise ratings.

[1] Wettig et al., QuRating: Selecting High-Quality Data for Training Language Models, 2024.

**Questions:**

- The message in Section 3.4 is overall confusing:

i) I'm not convinced that pointwise ratings can capture the same essence as pairwise ratings. Suppose you are evaluating the factuality of an LLM response given a question about Albert Einstein, the LLM gives two answers (A) and (B): both are factually correct but (A) adds a few more details while (B) simply gives the answer. Pointwise rating would not distinguish the subtleties of both responses while pairwise would. To support the opposite view, the authors perform validation through human judgments. However, I'm concerned about the validation process (Table 5), as it appears limited, having been conducted on only 10 samples.

ii) For prompt validation, the authors claim that human agreement exceeds 95%. Could you clarify the validation process ?

iii) Could you provide more context and an explanation to the equation (L.246-255) ? I can imagine that the error in rating decrease with the loss when training. But, how does it justify the use of pairwise over pointwise ?

- As a parallel work to [1], is there a reason behind choosing Qwen2-1.5B instead of Sheared-Llama-1.3B to train your DataMan ? Plus, the authors in [1] have disclosed their training dataset. This would have enabled a direct comparison between pointwise and pairwise ratings.
- For the "Sample with Qurating" baseline, could you clarify the sampling process ? My understanding is that you sampled with DataPajama and not the original QuRatedPajama. In this case, is the comparison fair ? A quick look at results in [1] showed lower perplexity.

[1] Wettig et al., QuRating: Selecting High-Quality Data for Training Language Models, 2024.

---

> ### Author Response · Authors · 2024-11-30
> **Author Response to Reviewer ViAr (Part I)**
>
> We would like to thank you for your valuable feedback and far-sighted insights. We hope the following statements will provide you gain an in-depth understanding of our DataMan method, thereby dispelling your concerns.
>
> **Weakness1. Lack of clarity in novelty.**
>
> 1. We highlight the innovations and contributions of DataMan compared to Qurating as follows:
>     1. **Different motivation:** **Our motivation is to enable the LLM to self-select quality criteria that can enhance its performance**, as its pre-training capability is related to perplexity [1, 2, 3]. High perplexity indicates that the data is difficult to learn, and vice versa. Thus, this **"reverse thinking"** encourages us to let the LLM self-identify the causes of anomalous perplexity to derive quality criteria, **rather than relying solely on human intuition**.
>     2. **Non-overlapping and more comprehensive quality criteria**: We derived 14 **quality criteria that do not overlap with Qurating**, which together form a more comprehensive data quality system a**cross four levels**(see the response to Reviewer**8xFT's Weakness1**) .
>     3. **Additional annotated domain types**: We have not only quality criteria **but also domain types,** which can support domain mixing of pre-training and post-training in vertical domains.
>     4. **Superior annotation, stronger base models, wider document length**: We use a superior GPT annotator **(GPT-4-turbo vs. GPT-3.5-turbo)**, a stronger base model **(Qwen2-1.5B vs. Sheared-Llama-1.3B)**, and support a wider range of token lengths for documents **([1, 2k] vs. [256, 512])**. This enhances DataMan's evaluation capability, enabling better result of longer and shorter documents.
>     5. **More feasible Rating: See the response to Question 1 for details.**
>     6. **Better Quality Data Selection**: Compared to Quating's sampling method, the data selected by DataMan leads to better LLM performance during pre-training.
> 2. Reviewers **Pb2j** and **8xFT** indicated that our approach is **overall sound** and **makes sense**. Reviewer **Tq6k** mentioned that using "reverse thinking" to derive quality metrics is **innovative**, while Reviewer **JcNL** described it as **effective**.
>
> **Questions1. Concerns regarding i) the Pointwise Rating v.s. Pairwise Rating, ii) Prompt validation, and iii) the Mathematical equation.**
>
> 1. **Clarify misconceptions between Pointwise Rating and Pairwise Rating.**
>     1. First, **the cases in Table 5 are sourced from Qurating's Table 4 and were not handpicked by us**. Qurating selected 10 documents from various sources, ranked by the writing style, to test an LLM's nuanced judgment capabilities.
>     2. Next, we use Table 5 (displayed in response to Reviewer Pb2j's Question 1) to analyze pointwise and pairwise ratings:
>         1. **Pairwise rating faces the challenge of minimal quality differences, while pointwise rating aligns with human judgment.**
>
>             Table 5 shows that the top three documents have only slight quality differences in "writing style", which evan humans find hard to distinguish (similarly seen in DataMan's "Overall Score.”). This challenges pairwise rating: how does one choose between documents pairs when quality differences are negligible? Generally, if documents meet an "acceptable" quality threshold, they should be accepted. In addressing this challenge, pointwise rating aligns well with human judgment.
>
>         2. Revisiting Qurating's original paper on **the use of pairwise evaluation and its issues — API calls, computational costs, and quality difference requirements.**
>
>             Appendix B.1 shows that constructing the Quarter's SFT dataset requires GPT-3.5-turbo to **make 40 predictions for a a criterion and document pair**, using the likelihood of the LLM preference as a label. Table 5 illustrates that **document pairs needed at least a 50% selection probability difference in quality criterion**, establishing a clear quality differential limit for SFT document pairs. Moreover, for N documents, **pointwise rating requires only N assessments**, whereas **pairwise rating necessitates N*(N-1) comparisons, with each comparison requiring 40 predictions**. Thus, when SFT evaluators, API calls, computational costs, and quality difference requirements for document pairs in pairwise ratings are significantly higher than those for pointwise ratings.Aslo, Qurating's Table 6 reveals that **for inference,** Quarter **demands at least a 50% quality difference in criteria for document pairs**, raising **concerns about its practice applicability**.
>
>     3. Finally, we recognize that pointwise and pairwise ratings capture different essences. However, **in both SFT evaluators or trillion-token scale pretraining**, **the applicability and cost-effectiveness outweigh the minor rating performance differences**, leading us to choose pointwise rating over pairwise rating.

---

> > ### Author Response · Authors · 2024-11-30
> > **Author Response to Reviewer ViAr (Part II)**
> >
> > 2. **Clarify the prompt validation process in detail**
> >
> >     For each quality criterion, we first identified a pool of documents that had been **preliminarily rated by an independent group**. From this pool, we selected **two groups of ten documents, one with high ratings** and **the other with low ratings**, ensuring a distinct quality gap. We then r**andomized these 20 documents** and had **five independent human annotators, who had not seen these documents before,** rate them on a scale of 1-5 based on the specific quality criterion. Subsequently, we used the GPT-4-turbo model, with our prompt, to evaluate the same 20 documents for the quality criterion. We calculated **the accuracy of the model’s ratings** by comparing them with t**he majority voting of human ratings** and found that **the model's agreement with human preferences exceeded 95%**. Additionally, we ensured **inter-rater reliability among human annotators by calculating the Kappa coefficient** [4], which validated the consistency of human ratings.
> >
> > 3. **Explanation of the mathematical equation between rating measure errors and the rating loss**
> >
> >     These formulas **do not aim to prove that pairwise rating is unreasonable**, but to demonstrate that **the measurement errors of pointwise rating are upper bounded by the loss of pointwise rating model**.
> >
> >     As the loss of the pointwise rating model (DataMan) approximates optimal via training, the measurement error also approaches minimal, **theoretically confirming the feasibility of pointwise ratings**.
> >
> > **Questions2. Why choose Qwen2-1.5B instead of Sheared-Llama-1.3B to train DataMan?**
> >
> > 1. Firstly, **Qwen2-1.5B** is a **newer and more powerful** base model **compared to Sheared-Llma-1.3B**. Choosing it for training DataMan can enhance its ability to assess data.
> > 2. Secondly, it is crucial to clarify that our work aims to propose a more comprehensive quality criteria system and the associated evaluator, **DataMan**, then **enable practical deployment during the trillion-token scale pre-training** phase, such as data selection and domain mixing. Therefore, we did **not consider comparing it with Qurating's paired ratings**.
> >
> > **Questions3. Clarification on Reproducing the "Sample with Qurating” Baseline**
> >
> > 1. **We fairly reproduced the "Sample with Qurating" baseline using** the Sheared-Llama-1.3B model **checkpoints provided in the Qurating code** (e.g., [[url1](https://huggingface.co/princeton-nlp/lm-1.3B-select_30B_tokens_by-educational_value-sample_with_temperature2.0)], [[url2](https://huggingface.co/princeton-nlp/lm-1.3B-select_30B_tokens_by-mix_of_criteria-sample_with_temperature2.0)]). All checkpoints can be find in the Models section of [[url3](https://github.com/princeton-nlp/QuRating)].
> > 2. The difference in validation perplexity between our results and the Quating paper's is due to our evaluation using SlimPajama's **full validation split** (**500M tokens**, via [[url4](https://huggingface.co/datasets/cerebras/SlimPajama-627B)]), while the Quating paper used only **1/10th of it** (**50M** **tokens**).
> >
> > [1] Wenzek, Guillaume, et al. "CCNet: Extracting high quality monolingual datasets from web crawl data." LREC 2020.
> >
> > [2] Muennighoff, Niklas, et al. "Scaling data-constrained language models." Neurips 2023.
> >
> > [3] Marion, Max, et al. "When less is more: Investigating data pruning for pretraining llms at scale." Neurips 2023.
> >
> > [4] McHugh, Mary L. "Interrater reliability: the kappa statistic." *Biochemia medica* 22.3 (2012): 276-282.

---

> > > ### Comment · Reviewer_ViAr · 2024-12-02
> > >
> > > I thank the authors for their responses.
> > >
> > > **On the self-selection of LLM quality criteria**: This seems misleading, as you're not asking the model being trained to reflect on itself but rather relying on **GPT-4**. This isn't truly reverse thinking; it's more of a shortcut, treating GPT-4 as a representative for all LLMs.
> > >
> > > **On Superior annotation, stronger base models, wider document length**: I acknowledge that the paper presents a considerable  work (as a tech report) but I maintain my view on the novelty. In fact, claiming that you have better results because you use stronger base models (or using GPT-4 over GPT-3.5) do not enable comparison with (Wettig et al., 2024) thus hindering my comprehension of the effectiveness of your approach over the former.
> > >
> > > **On pairwise vs pointwise**: I agree that the lower complexity of pointwise is appealing but this represents one of your main contributions over (Wettig et al., 2024) and I would have loved to see more (fair) comparisons with this work. The human evaluation conducted on 20 randomized samples is in my opinion too brittle to draw conclusions.
> > >
> > > **On the sampling in Q3**: I maintain my view on fair comparisons the two models are not comparable in the sense that the data and models are not the same (you used their public checkpoints and not reproducing their work on your data).
> > >
> > > For these reasons, I maintain my score.
> > >
> > > [1] Wettig et al., QuRating: Selecting High-Quality Data for Training Language Models, 2024.

---

> > > > ### Author Response · Authors · 2024-12-03
> > > > **Response to the Reviewer (Part I)**
> > > >
> > > > Thank you for your prompt response. We sincerely appreciate your insightful feedback, which will help us improve our work(👍)! We are glad to have the opportunity to discuss and address your concerns to gain your approval of our works.
> > > >
> > > > **Question 1. Explanation of the self-selection of LLM quality criteria**
> > > >
> > > > Thank you for pointing out this. Our initial intention was to select GPT-4-Turbo, which we consider the most comprehensive LLM to represent other LLMs.
> > > >
> > > > This is **because it can most accurately identify the causes of text perplexity anomalies** and **derive quality criteria** **more closely aligned with LLM performance**.
> > > >
> > > > These quality criteria are **crucial for** the subsequent **fine-tuning of our DataMan**, enabling it to **filter higher-quality pre-training data** and thereby **train more robust LLMs**. The process requires a high degree of accuracy, as **errors will accumulate as the pipeline extends**[1].
> > > >
> > > > **Question 2&3. More (fair) comparison between Qurating's Pairwise Rating and DataMan's Pointwise Rating**
> > > >
> > > > Based on the reviewer's suggestions, we have conducted more (fair) comparison between Qurating's pairwise rating and DataMan's pointwise rating:
> > > >
> > > > Firstly, **the evaluation criteria differ between the two rating methods**. Qurating's pairwise rating focuses on the accuracy of preferences between (tA, tB), while DataMan's pointwise rating evaluates accuracy using a 5-point scale. Therefore, **it is not feasible to directly compare their accuracy with the SFT validation set**.
> > > >
> > > > To address this, we opted to **rank the annotation results of both ratings on the same set of documents to ensure a fair comparison.**
> > > >
> > > > Here is the specific methodology: We selected 58,824 documents from [the first split](https://huggingface.co/datasets/princeton-nlp/QuRatedPajama-1B_tokens_for_analysis/blob/main/data/train-00000-of-00017.parquet) of [Qurating's publicly available 1B analysis set](https://huggingface.co/datasets/princeton-nlp/QuRatedPajama-1B_tokens_for_analysis). These documents were annotated with Qurating's pairwise rating and DataMan's pointwise rating. Subsequently, the documents were ranked according to each rating’s quality criteria. We present the top, median, and bottom 10 documents based on Quating's four criteria—writing style, facts and trivia, educational value, and required expertise—as well as DataMan's Overall Score with the sum of the remaining criteria.
> > > >
> > > > All results are provided [here](https://anonymous.4open.science/r/ranking_examples-BAB0/README.md). If you wish to see more examples, we would be glad to offer additional ranking results.
> > > >
> > > > Our conclusions are as follows:
> > > >
> > > > 1. When ranking documents based on Qurating's four criteria, there is **a similar trend observed in DataMan's Overall Score for** **the top and bottom 10 documents**: **”higher Qurating criteria correlate with higher Overall Scores, and vice versa.“**
> > > > 2. Interestingly, for **the median 10 documents** ranked by **Qurating's criteria**, there **are minimal differences in scores across these criteria**, yet their corresponding Overall Scores vary between 3 and 5. This supports our previous assertion that "**Pairwise rating faces challenges with minimal quality differences, while pointwise rating aligns with human judgment.**"
> > > > 3. Based on the rankings by Overall Score with the sum of remaining criteria, **our approach effectively highlights the highest quality documents, such as in STEM fields** like materials science, geology, physics, biology, and chemistry.
> > > >
> > > >     In other words, these **14 quality criteria collectively offer a comprehensive evaluation of text quality across four levels**:
> > > >
> > > >     - **Basic Requirements:** Including *Accuracy, Language Consistency, Style Consistency, Grammatical Diversity, and Structural Standardization* to ensure readability.
> > > >     - **Expression Requirements:** Covering *Coherence, Semantic Density, Originality, and Topic Focus* to enhance expressiveness and informativeness.
> > > >     - **Advanced Requirements:** Involving *Knowledge Novelty, Creativity, Professionalism, and Sensitivity* to deepen and secure text quality.
> > > >     - **Overall Score:** Allows the LLM to weigh each quality criterion, minimizing human bias and providing a holistic quality evaluation.
> > > > 4. Additionally, the "application_domain" field, referred to as **"domain type"** in our paper, accurately categorizes the domain of each document. This aids in **optimizing the model's pre-training domain mixing and continue pre-training** in vertical domains (this is **also our additional contribution**, as shown in Table 2 with CPT results for financial, legal, and medical domains).
> > > >
> > > > [1] G Welch. An introduction to the Kalman filter. 1995.

---

> > > > > ### Author Response · Authors · 2024-12-03
> > > > > **Response to the Reviewer (Part II)**
> > > > >
> > > > > **Question 2&4. More rigorous and fair comparison between the "Sample with Qurating" baseline and DataMan**
> > > > >
> > > > > Following the reviewer's suggestions, we conducted a more rigorous and fair comparison experiment between the "Sample with Qurating" baseline and DataMan. To ensure transparency, we have **detailed all aspects of the comparison, and maintained the same settings as the original Qurating**. We've also **included links** to all the data, models, code, scripts, and hyperparameters used.
> > > > >
> > > > > Firstly, the DataPajama dataset (447B tokens) is curated from the SlimPajama dataset, with each document [chunked](https://github.com/princeton-nlp/QuRating/blob/main/data_tools/tokenize_dataset.py) into sequences of 1024 tokens using the Llama tokenizer. Next, we employed the downloaded [QuRater](https://huggingface.co/princeton-nlp/QuRater-1.3B) to [annotate](https://github.com/princeton-nlp/QuRating/blob/main/data_tools/qurater_annotate.py) DataPajama. Following Qurating's four metrics (Writing Style, Facts & Trivia, Educational Value, Required Expertise) with a temperature of τ=2, we used [top-k sampling without replacement to select a 30B token](https://github.com/princeton-nlp/QuRating/blob/main/data_tools/select_subset.py) subset from DataPajama, maintaining the same source distribution as SlimPajama. We then [trained](https://github.com/princeton-nlp/QuRating/blob/main/TrainLM.sh) the [Sheared-Llama-1.3B model](https://huggingface.co/princeton-nlp/Sheared-LLaMA-1.3B) from scratch for one epoch. Finally, we compared the results based on three performance metrics: ten in-context learning tasks using [lm-evaluation-harness](https://github.com/EleutherAI/lm-evaluation-harness) ([ARC-easy/challenge](https://huggingface.co/datasets/allenai/ai2_arc/tree/main), [SciQA](https://huggingface.co/datasets/allenai/sciq/tree/main), [LogiQA](https://huggingface.co/datasets/EleutherAI/logiqa/tree/main), [BoolQ](https://huggingface.co/datasets/google/boolq/tree/main), [HellaSwag](https://huggingface.co/datasets/Rowan/hellaswag), [PIQA](https://huggingface.co/datasets/ybisk/piqa/tree/main), [WinoGrande](https://huggingface.co/datasets/allenai/winogrande/tree/main), [NQ](https://huggingface.co/datasets/google-research-datasets/nq_open), [MMLU](https://huggingface.co/datasets/lighteval/mmlu/tree/main), and the per-token perplexity of the [SlimPajama validation and test sets](https://huggingface.co/datasets/cerebras/SlimPajama-627B).
> > > > >
> > > > > Table 1. Results for **ten in-context learning tasks, the perplexity of the SlimPajama validation and test sets**, bold indicates the best performance.
> > > > >
> > > > > | **Selection Method** |  | **Val Perplexity** | **Test Perplexity** | **Reading Comprehension (5 tasks)** | **Commonsense Reasoning (3 tasks)** | **World Knowledge (2 tasks)** | **Average (10 tasks)** |
> > > > > | --- | --- | --- | --- | --- | --- | --- | --- |
> > > > > | Writing Style | τ=2.0 | 10.63 | 10.66 | 50.96 | 56.43 | 14.25 | 45.26 |
> > > > > | Facts & Trivia | τ=2.0 | 10.65 | 10.75 | 52.80 | 55.67 | 15.55 | 46.21 |
> > > > > | Educational Value | τ=2.0 | 10.63 | 10.70 | 53.32 | 56.33 | 15.80 | 46.72 |
> > > > > | Required Expertise | τ=2.0 | 13.57 | 10.72 | 52.72 | 55.60 | 15.10 | 46.06 |
> > > > > | Overall Score | l=5.0 | **10.52** | **10.50** | **55.22** | **60.17** | **17.35** | **49.13** |
> > > > >
> > > > > Table 2. Detailed results for **ten in-context learning tasks**, bold indicates the best performance.
> > > > >
> > > > > |  |  | **Reading Comprehension** |  |  |  |  | **Commonsense Reasoning** |  |  | **World Knowledge** |  |  |
> > > > > | --- | --- | --- | --- | --- | --- | --- | --- | --- | --- | --- | --- | --- |
> > > > > | **Selection Method**  |  | ARC-E (15) | ARC-C (15) | SciQ (2)  | LogiQA (2) | BoolQ (0) | HellaSw. (6)  | PIQA (6)  | W.G. (15) | NQ (10) | MMLU (5) | Average |
> > > > > | Writing Style | τ=2.0 | 56.5 | 28.5 | 85.7 | 24.8 | 59.3 | 44.8 | 68.6 | 55.9 | 4.6 | 23.9 | 45.26 |
> > > > > | Facts & Trivia | τ=2.0 | 59.4 | 29.9 | 88.1 | 25.1 | **61.5** | 43.9 | 68.4 | 54.7 | 4.2 | 26.9 | 46.21 |
> > > > > | Educational Value | τ=2.0 | 60.7 | 30.3 | 88.7 | **26.7** | 60.2 | 45.6 | 69.2 | 54.2 | 4.5 | **27.1** | 46.72 |
> > > > > | Required Expertise | τ=2.0 | 59.7 | 29.6 | 89.1 | 23.8 | 61.4 | 43.2 | 67.5 | 56.1 | 4.6 | 25.6 | 46.06 |
> > > > > | Overall Score  | l=5.0 | **66.1** | **34.0** | **90.7** | 26.1 | 59.2 | **51.5** | **70.7** | **58.3** | **7.8** | 26.9 | **49.13** |

---

> > > > > > ### Author Response · Authors · 2024-12-03
> > > > > > **Response to the Reviewer (Part III)**
> > > > > >
> > > > > > Table 3. Detailed results for the **perplexity of the SlimPajama validation sets**, bold indicates the best performance.
> > > > > >
> > > > > > | **Selection Method**  |  | **CC** | **C4** | **Github** | **Wiki** | **ArXiv** | **StackEx** | **Book** | **Overall** |
> > > > > > | --- | --- | --- | --- | --- | --- | --- | --- | --- | --- |
> > > > > > | Writing Style | τ=2.0 | 10.96 | **14.05** | 2.96 | 10.31 | 5.62 | **5.63** | 12.04 | 10.60 |
> > > > > > | Facts & Trivia | τ=2.0 | 10.99 | 14.27 | 3.03 | 10.62 | 5.58 | 6.15 | 12.36 | 10.69 |
> > > > > > | Educational Value | τ=2.0 | 11.01 | 14.14 | 2.96 | 10.46 | 5.56 | 6.12 | 12.37 | 10.67 |
> > > > > > | Required Expertise | τ=2.0 | 11.08 | 14.13 | 2.95 | 10.23 | 5.57 | 6.15 | 12.25 | 10.71 |
> > > > > > | Overall Score  | l=5.0 | **10.56** | 16.36 | **2.66** | **8.51** | **4.74** | 5.92 | **10.79** | **10.52** |
> > > > > >
> > > > > > Table 4. Detailed results for the **perplexity of the SlimPajama test sets,** bold indicates the best performance.
> > > > > >
> > > > > > | **Selection Method**  |  | **CC** | **C4** | **Github** | **Wiki** | **ArXiv** | **StackEx** | **Book** | **Overall** |
> > > > > > | --- | --- | --- | --- | --- | --- | --- | --- | --- | --- |
> > > > > > | Writing Style | τ=2.0 | 10.96 | **13.95** | 2.96 | 10.23 | 5.61 | 5.24 | 12.64 | 10.66 |
> > > > > > | Facts & Trivia | τ=2.0 | 10.97 | 14.08 | 2.94 | 10.52 | 5.65 | 5.23 | 12.96 | 10.77 |
> > > > > > | Educational Value | τ=2.0 | 11.02 | **13.95** | 2.92 | 10.38 | 5.59 | 5.18 | 12.96 | 10.74 |
> > > > > > | Required Expertise | τ=2.0 | 11.05 | 14.02 | 2.91 | 10.14 | 5.57 | 5.19 | 12.94 | 10.71 |
> > > > > > | Overall Score  | l=5.0 | **10.51** | 16.18 | **2.56** | **8.40** | **4.77**  | **4.99** | **11.28** | **10.50** |
> > > > > >
> > > > > > Our conclusions are as follows:
> > > > > >
> > > > > > 1. With all settings ensuring a fair comparison, **DataMan significantly outperforms the "Sample with Qurating" baseline** across all three evaluation metrics.
> > > > > > 2. The "Sample with Qurating" baseline, **reproduced on the DataPajama dataset**, shows performance across the three evaluation metrics that **are nearly identical to results reproduced on the QuratedPajama dataset**.
> > > > > >
> > > > > > To complete the experiments suggested by the reviewers before the discussion deadline (December 2, AOE), we conducted the Qurating annotation of DataPajama and the pretraining of the four LLMs from scratch **using 512 A800 GPUs in parallel.** We simply hope that our efforts on this paper (including significant time, funds, computational resources, and manpower) can earn your recognition and support 🙏🙏. Thank you!

---

> > > > > > > ### Comment · Reviewer_ViAr · 2024-12-03
> > > > > > >
> > > > > > > I thank the authors for conducting the additional experiments, as they are essential for thoroughly evaluating the effectiveness of DataMan compared to other approaches.
> > > > > > >
> > > > > > > **On GPT-4**: The terms "self-selection" and "reverse-thinking" appear misleading in this context. It should be clarified that the quality criteria are entirely derived from GPT-4-Turbo. Consequently, I remain unconvinced by the claim that "they are more closely aligned with LLM performance." Furthermore, relying on GPT-4 for quality assessments may introduce bias.
> > > > > > >
> > > > > > > **Regarding Fair Comparison**: I sincerely appreciate the authors' efforts to provide a fairer comparison with the baseline through additional experiments. However, given the limited time and the absence of an updated manuscript, I am unable to vouch for acceptance at this stage. I have, however, adjusted my scores for soundness and presentation.

---

> > > > > > > > ### Author Response · Authors · 2024-12-03
> > > > > > > > **Thanks for your timely feedback**
> > > > > > > >
> > > > > > > > We greatly appreciate your recognition of our work and are very happy to see that our additional experiments let you confirmed the effectiveness of DataMan.
> > > > > > > >
> > > > > > > > In fact, **we have already submitted an updated PDF version**. However, due to **the ICLR PDF review mechanism not highlighting changes from the original version**, some misunderstandings may have arisen. **We sincerely apologize for any confusion caused by this.**
> > > > > > > >
> > > > > > > > We **highly value the concerns you raised about the misunderstanding of two technical terms in the "On-GPT4" and your attention to fair comparisons**. We salute your academic rigor🫡. We are **eager to incorporate your invaluable feedback**, along with all the suggestions from reviewers during this rebuttal phase, into our revised PDF immediately.
> > > > > > > >
> > > > > > > > Unfortunately, as **the ICLR PDF revision deadline (November 27th AOE) has passed** 😭😭, we can only commit to fully integrating these insights into the final version.
> > > > > > > >
> > > > > > > > We would be extremely grateful if you could reconsider your scores to support our work. Thank you once again!

---

### Official Review · Reviewer_Tq6k · 2024-11-04

**Soundness:** 3
**Presentation:** 3
**Contribution:** 3
**Rating:** 6
**Confidence:** 3

**Summary:**

The authors propose DataMan, a pipeline for pretraining data quality filtering. First, PPL-thresholded data are used to extract 13 different quality rating criteria. Then, a smaller LM is fine-tuned to annotate documents with scores for each quality rating (and domain.) Finally, the LM is used to produce a higher quality small subset of RedPajama. Training on this subset shows accuracy gains across numerous tasks and significant improvement over baselines.

**Strengths:**

- The paper uses a novel "reverse thinking" concept which derives quality criteria using documents filtered into high and low perplexity buckets, which avoids pitfalls of criteria based on human intution.
- Introduces 13 detailed criteria for quality, which moves beyond typical heuristic approaches which threshold on perplexity, deduplication, etc.
- The focus on 13 specific quality metrics and their annotations makes it clear why a particular document was chosen by the model for inclusion in the dataset, thus making the method interpretable.
- DataMan demonstrates a significant performance improvement over numerous heuristic and model-based baselines across multiple metrics: PPL, benchmark accuracy w/ ICL, and instruction following.
- The method performs well against impressive baselines, including statistical approaches like DSIR and models trained with 50% more data.
- Empirical evidence shows benefits of DataMan for domain-specific continued pre-training.
- The presentation is clear.

**Weaknesses:**

- The authors apply DataMan to a 30B subset of SlimPajama, which is a relatively small dataset. Further work should be done to validate that the method works for large pretraining-scale datasets.
- The SlimPajama dataset has already been filtered for quality with duplicate documents removed. It’s not clear if this influences the analysis of the method or if similar improvements would be seen applying DataMan to unfiltered web data.
- The paper does not discuss the inference FLOPs required to run DataMan. This would be useful to understand the tradeoff between spending more FLOPs on annotation versus pretraining for more tokens.

**Questions:**

- SlimPajama is already quality-filtered and deduplicated. How does this affect your analysis and results? If DataMan's improvements come mainly from filtering already clean data, the impact on raw web data could be quite different. Could you provide results or analysis on a raw, unfiltered dataset?

- How is the "overall score" calculated from the 13 criteria? Could you provide the exact formula or methodology? Is this determined by the annotating LLM?

---

> ### Author Response · Authors · 2024-11-30
> **Author Response to Reviewer Tq6k (Part I)**
>
> We express our gratitude for favorable feedback and valuable insights, and we sincerely hope that our response can address your concern.
>
> **Weakness1. DataMan needs to validate effectiveness on larger datasets (more than 30B).**
>
> Following the reviewers' suggestions, we used **the best variants of DataMan (Overall Score l=5.0)  and Qurating (Educational Value τ=2.0) to filter 60B data** from Slimpajama and trained a Sheared-Llama-1.3B model from scratch. We then compared the results based on three performance metrics: **ten in-context learning tasks, and the complexity of the Slimpajama validation and test sets**.
>
> Table 1. Results for **ten in-context learning tasks, the perplexity of the SlimPajama validation and test sets**, bold indicates the best performance.
>
> | Selection Method            | | Val Perplexity | Test Perplexity | Reading Comprehension (5 tasks) | Commonsense Reasoning (3 tasks) | World Knowledge (2 tasks) | Average (10 tasks) |
> |-----------------------------|-----------|----------------|-----------------|--------------------------------|---------------------------------|---------------------------|---------------------|
> | Uniform                     |           | 10.81          | 10.79           | 53.7                           | 58.4                            | 16.4                      | 47.6                |
> | Educational Value | τ=2.0    | 9.81 ↓1.00     | 9.85 ↓0.94      | 54.2 ↑0.5                      | 58.7 ↑0.3                      | 16.0 ↓0.4                 | 47.9 ↑0.3           |
> | Overall Score    | l=5.0     | 9.93 ↓0.88     | 9.91 ↓0.88      | 56.5 ↑2.8                      | 62.9 ↑4.5                      | 17.5 ↑1.1                 | 50.6 ↑3.0           |
>
>
> Table 2. Detailed results for **ten in-context learning tasks**, bold indicates the best performance.
>
> |  |  | **Reading Comprehension** |  |  |  |  | **Commonsense Reasoning** | |  | **World Knowledge** |  | **Average** |
> | --- | --- | --- | --- | --- | --- | --- | --- | --- | --- | --- | --- | --- |
> | **Selection Method** |  | ARC-E (15) | ARC-C (15) | SciQ (2) | LogiQA (2) | BoolQ (0) | HellaSw. (6)|PIQA (6) | W.G. (15) | NQ (10) | MMLU (5) |  |
> | Uniform |  | 63.0 | 32.4 | 89.1 | 22.3 | **61.6** | 49.4| 70.7 | 55.0 | 6.4 | 26.3 | 47.6 |
> | Educational Value | τ=2.0 | 64.5↑1.5 | 31.1↓1.3 | 90.9↑1.8 | 24.1↑1.8 | 60.6↓1.0 | 50.2↑0.8 | 69.9↓0.8 | 56.0↑1.0 | 7.0↑0.6 | 25.0↓1.3 | 47.9↑0.3 |
> | Overall Score | l=5.0 | **68.6↑5.6** | **36.8↑4.4** | **91.9↑2.8** | **24.4↑2.1** | 60.6↓1.0 | **56.1↑6.7**| **72.4↑1.7** | **60.1↑5.1** | **8.6↑2.2** | **26.4↑0.1** | **50.6↑3.0** |
>
> Table 3. Detailed results for **the perplexity of the SlimPajama validation sets**, bold indicates the best performance.
>
> | **Selection Method** |  | **CC** | **C4** | **Github** | **Wiki** | **ArXiv** | **StackEx** | **Book** | **Overall** |
> | --- | --- | --- | --- | --- | --- | --- | --- | --- | --- |
> | Uniform |  | 10.81 | 16.50 | 2.89 | 9.54 | 5.22 | 5.88 | 11.43 | 10.81 |
> | Educational Value | τ=2.0 | 10.15 ↓0.66 | **12.94 ↓3.56** | 2.78 ↓0.11 | 9.11 ↓0.43 | 5.20 ↓0.02 | 5.68 ↓0.20 | 11.36 ↓0.07 | 9.81 ↓1.00 |
> | Overall Score | l=5.0 | **9.94 ↓0.87** | 15.58 ↓0.92 | **2.52 ↓0.37** | **7.71 ↓1.83** | **4.53 ↓0.69** | **5.62 ↓0.26** | **10.13 ↓1.3** | **9.93 ↓0.88** |
>
> Table 4. Detailed results for **the perplexity of the SlimPajama test sets**, bold indicates the best performance.
>
> | **Selection Method** |  | **CC** | **C4** | **Github** | **Wiki** | **ArXiv** | **StackEx** | **Book** | **Overall** |
> | --- | --- | --- | --- | --- | --- | --- | --- | --- | --- |
> | Uniform |  | 10.76 | 16.31 | 2.78 | 9.41 | 5.27 | 4.93 | 11.92 | 10.79 |
> | Educational Value | τ=2.0 | 10.16 ↓0.60 | **12.83 ↓3.48** | 2.73 ↓0.05 | 8.99 ↓0.42 | 5.25 ↓0.02 | 4.76 ↓0.17 | 11.92 | 9.85 ↓0.94 |
> | Overall Score | l=5.0 | **9.90 ↓0.86** | 15.40 ↓0.91 | **2.43 ↓0.35** | **7.61 ↓1.80** | **4.56 ↓0.71** | **4.68 ↓0.25** | **10.61 ↓1.31** | **9.91 ↓0.88** |
>
> Our conclusions are as follows:
> 1. With a 60B pre-training data scale, **DataMan significantly outperforms FineWeb across ten in-context learning tasks.**
> 2. At the 60B pre-training data scale, DataMan's perplexity performance on both the SlimPajama validation and test sets matches that of Qurating. However, apart from C4, we **surpass Qurating in perplexity performance from the other six sources**.

---

> ### Author Response · Authors · 2024-11-30
> **Author Response to Reviewer Tq6k (Part II)**
>
> **Weakness2 & Questions1. Analysis and results of applying DataMan to raw, unfiltered web data.**
>
> According to the reviewers' adivces, we conducted experiments to **analyze DataMan's performance with raw, unfiltered web data.** We extracted 1,401,749 documents from the [Common Crawl](https://huggingface.co/datasets/togethercomputer/RedPajama-Data-1T/blob/main/urls/common_crawl.txt) (first 2 splits) of the RedPajama-1T dataset as test data and filtered it using both the Slimpajama [1] and FineWeb [2] data processing rules. We applied DataMan to the test dataset, filtering based on an overall score threshold of less than 4. We then evaluated the metrics between DataMan and the data processing rules based on the following definitions:
>
> - True Positives (TP): Both labeled as dirty;
> - False Positives (FP): Rules labeled as clean, DataMan labeled as dirty;
> - False Negatives (FN): Rules labeled as dirty, DataMan labeled as clean;
> - True Negatives (TN): Both are labeled as clean.
>
> Table 5. Analysis and results of **applying DataMan to raw, unfiltered web data**.
>
> | **Data Processing Rules** | **TP** | **FP** | **FN** | **TN** | **Accuracy** | **Precision** | **Recall** | **F1 Score** |
> | --- | --- | --- | --- | --- | --- | --- | --- | --- |
> | Fineweb | 30,241 | 64,853 | 194,862 | 1,111,771 | 0.81 | 0.32 | 0.13 | 0.19 |
> | Slimpajama | 64 | 94,416 | 32 | 1,307,216 | 0.93 | 0.00 | 0.67 | 0.00 |
>
> **Results and Analysis:**
>
> 1. **DataMan's high accuracy indicates its effectiveness** is comparable to existing data processing rules **for web data**.
> 2. Low recall and precision suggest that **DataMan is orthogonal to existing rules**, especially in identifying low-quality data.
> 3. In addressing Reviewer JcNL's Weakness2 & Questions 1,2, **DataMan** outperforms **FineWeb**, affirming that **they are not mutually exclusive** as shown in conclusion 2.
> 4. Case studies on FP and FN show that **DataMan better preserves or filters text at a semantic level**, surpassing traditional rule-based filtering.
>     - Slimpajama FN: _A lesson on how to say "I love you" in Spanish_
>     ```
>      How do I say \"I Love You\" in Spanish?\nYou can say \"Te quiero\" or \"Te amo\", depending on the intensity of the affection you feel for that person :)\nForum>How do I say \"I Love You\
>     ```
>     - Slimpajama FP: _Contains numerous typos, lacks semantic and coherence_
>     ```
>     Rognan_Norway\nJoAnn Frank Memorial Tree\nA | B | C | D | E | F | G | H | I | J | K | L | M | N | O | P | Q | R | S | T | U | V | W | X | Y | Z | - | ? | \' | " | ( | [ | & | < | ¢ | … | 2 | Ø | … | None\nPierre de Conde\'\nPierreAAAAde Conde\' de Conde\'AAAAPierre 0 1066 2110591 953 Tileris, Normandy, France, France 0 0 1130 2133966 889 64 23375 M YES YES R\nFor technical support or genealogy questions, please contact ourpuzzlingpast_admin
>     ```
>     - Fineweb FN: _A autobiography of Timothy Reckart as an Actor and Director_
>     ```
>     Timothy Reckart\nActor | Director\nTimothy Reckart is an American animator and director based in New York City, specializing in puppet stop motion. He is best known for his 2012 film Head Over Heels, nominated for the Academy Award for Best Animated Short Film in 2013.[1] Reckart grew up in Tucson, Arizona, where he attended University High School.[2] He studied history and literature at Harvard University where he graduated in 2009.[2] ...
>     ```
>     - Fineweb FP: _Contains nonsensical content and sensitive topics (such as racism, drugs, suicide, and pornography)_
>     ```
>     The Murderers At Your Canadian School Boards\nSince 1985, for every Asian computer science major Canadian high schools produce, they must have also produced 15 or 20 white and black crack heads.\nIn 1994 or in 2007 they'd say that gay youth has the highest suicide rate. They'd also say that the media doesn't report on teen suicide because they don't want to encourage copycats.\nWhy doesn't the CBC do some investigative reporting and interview paramedics to see if back in an era when white people watched an episode of Seinfeld and the movie, 'Schindler's List,' all in one day whether good-looking and average looking boys with freckles and South Asian girls who were unattractive or couldn't speak English really had the highest suicide rate? ...
>     ```
> 5. Compared to uniform sampling, **DataMan further improves data quality selection** on the Slimpajama dataset, proving **its capability to complement existing processes** to select superior data.
>
> [1] SlimPajama: A 627B token cleaned and deduplicated version of RedPajama [[paper](https://cerebras.ai/blog/slimpajama-a-627b-token-cleaned-and-deduplicated-version-of-redpajama)]  [[Data processing code](https://github.com/Cerebras/modelzoo/tree/main/src/cerebras/modelzoo/data_preparation/nlp/slimpajama)]
>
> [2] The FineWeb Datasets: Decanting the Web for the Finest Text Data at Scale [[paper](https://arxiv.org/abs/2406.17557)] [[Data processing code](https://github.com/huggingface/datatrove/blob/main/examples/fineweb.py)]

---

> > ### Author Response · Authors · 2024-11-30
> > **Author Response to Reviewer Tq6k (Part III)**
> >
> > **Weakness3. There are no inference FLOPs required to run DataMan.**
> >
> > We report the **inference FLOPs and memory usage of three DataMan models** from Appendix B.2 using the vLLM tool [3] on the Slimpajama test dataset with a single A800 GPU.
> >
> > Table6. The **inference FLOPs and memory usage** of three DataMan models
> >
> > |  | Input Speed (Toks/S) | Output Speed (Toks/S) | Memory (G) | Processing Speed (Docs /S)  |
> > | --- | --- | --- | --- | --- |
> > | **All-rating DataMan** | 31822 | 868 | 72.9 | 30 |
> > | **Score-Only DataMan** | 63644 | 1736 | 72.9 | 60 |
> > | **Domain-Only DataMan** | 63644 | 1736 | 72.9 | 60 |
> >
> > **Questions2. How is the "overall score" calculated from the 13 criteria?**
> >
> > The **overall score is determined by the LLM** based on its comprehensive performance across 13 quality criteria. **This is reasonable** because it allows the LLM to **assign weights to each quality criterion** in the overall score, fostering dependencies between scores and avoiding bias from manual adjustments.
> >
> > [3] Woosuk Kwon, et al. Efficient memory management for large language model serving with pagedattention. ACM SIGOPS 2023.

---

### Official Review · Reviewer_8xFT · 2024-11-09

**Soundness:** 3
**Presentation:** 3
**Contribution:** 3
**Rating:** 8
**Confidence:** 4

**Summary:**

The paper introduces "DataMan," a comprehensive data manager designed for pre-training large language models (LLMs). DataMan aims to improve data quality and selection by introducing 13 well-defined quality criteria and an overall quality score. The approach leverages reverse thinking by prompting LLMs to self-identify the factors that influence performance, focusing on perplexity outliers. The paper conducts experiments on several benchamrks to show how DataMan effectively annotates and selects high-quality training data, resulting in improvements in in-context learning (ICL) and instruction-following tasks. As described in the paper, the implementation of the framework would be released, which could be beneficial for real practices.

**Strengths:**

1. The motivation of using LLMs to self-select the criterias that are beneficial for model's performance is straightforward and easy to understand. The overall framework makes sense.
2. The description of the method is clear and easy to follow.
3. Data selection for both pre-training, instruction-tuning, and SFT is an important direction for LLM applications, where I think the direction in this paper is interesting and could be beneficial for downstream applications. The experiments show that the proposed method can also improve the model's performances in different benchmarks.

**Weaknesses:**

1. I think the motivation to use 14 scores is still a little bit unclear. It would be beneficial to have an analysis of why these scores were chosen and how these scores can distinctively differentiate models' abilities when evaluation.

**Questions:**

1. How do the authors plan to resolve the situation where model itself made unreasonable decisions on criterias? The hallucinations generated during the process could have snowball effects for some training scenarios. One approach that can be tried is using Mixture-of-Experts to reduce the possibility of such situations, but I would like to see authors' insights and if they did any experiments on this direction.
2. Is there a way to mitigate the high computational cost of DataMan’s annotation process, perhaps by leveraging lower-cost models or more efficient sampling techniques?
3. The paper mentions that perplexity and ICL do not align strictly in some cases. Could the authors provide more insights or analysis into the scenarios where this misalignment is most pronounced and how it impacts downstream tasks?
4. I think "vasrtdatasets" at line 26 is a typo?

---

> ### Author Response · Authors · 2024-11-30
> **Author Response to Reviewer 8xFT (Part I)**
>
> We want to thank you for your thoughtful and insightful reply! We sincerely hope our response can resolve your concerns.
>
> **Weakness1. Why 14 scores were chosen and how they differentiate model abilities?**
>
> 1. These **14 quality criteria were chosen by the LLM itself to enhance its performance**, as its pre-training capability is related to perplexity [1, 2, 3]. "High perplexity indicates that the data is difficult to learn, and vice versa." We extracted documents with the top 2% and bottom 2% of perplexity from different sources, using GPT-4-turbo to **identify the reasons behind the anomalies perplexity in the documents**. Through iterative refinement, we developed 14 quality criteria associated with LLM performance. For a detailed process, please refer to the response to Reviewer **Pb2j's Question 2**.
> 2. These **14 quality criteria were selected to provide a step-by-step comprehensive assessment of text quality** through the following four levels:
>     - **Basic Requirements:** These include _Accuracy, Language Consistency, Style Consistency, Grammatical Diversity, and Structural Standardization_, ensuring readability.
>     - **Expression Requirements:** Covering _Coherence, Semantic Density, Originality, and Topic Focus_, aimed at enhancing expression and informativeness.
>     - **Advanced Requirements:** Involving _Knowledge Novelty, Creativity, Professionalism, and Sensitivity_, to deepen and secure text quality.
>     - **Overall Score:** allows the LLM to weigh each quality criterion, avoiding human bias and providing a comprehensive quality evaluation.
> 3. According to the original paper's Table 8, the scores of quality criteria across different domains show how these criteria differentiate model abilities:
>     - **Knowledge Novelty** excelled in mathematics, enhancing the **model’s math abilities**.
>     - **Creativity** was highest in cultural domains, boosting the **model's literary skills.**
>     - **Professionalism** performed exceptionally in mathematics, law, medicine, and finance, improving the **model's performance in these professional fields**.
>     - **Sensitivity** scored lower in government and entertainment domains, aiding the **model in filtering harmful speech and politically sensitive topics**.
>     - **Originality** and **Semantic Density** scored higher in vertical domain data, **enhancing the model's performance in vertical fields**.
>     - **Other criteria** were performed evenly across domains, ensuring the basic requirements were met and **enhancing the model’s general capabilitie**s.
> 4. From the original paper's Table 15, one can observe the quality criteria that performed best in various ICL tasks, analyzing **how these criteria enhance the LLM's abilities in specific downstream tasks**:
>     - High **Semantic Density** data improved the LLM’s ability to **solve grade-school science questions** (ARC-E, ARC-C), while **more challenging science exam questions** (SciQ) required data with high **Semantic Density** and **Professionalism**.
>     - High **Creativity** data assisted the LLM in handling tasks like **article summarization and movie subtitles** (HellaSw., W.G.), while **more complex physical interactive reasoning** tasks (PIQA) required a combination of high **Semantic Density**.
>     - High **Originality** supported the LLM in **solving Wikipedia knowledge-related tasks**, where redundant knowledge was not helpful for such problems.
> 5. The original paper's Table 13 shows the quality criteria with the lowest perplexity in different data sources, analyzing **how these criteria enhance the LLM's fitting ability in specific data sources**.
>     - **Sensitivity** had the lowest perplexity in general web domains CommonCrawl and C4, indicating that when handling open web data, the key is to recognize and avoid sensitive topics, thereby **improving adaptability to web content**.
>     - In the Wikipedia source, **Semantic Density**, **Originality**, and **Topic Focus** performed better, suggesting that the **LLM absorbs world knowledge** through high-information, original thematic content.
>     - **Creativity** had the lowest perplexity in book sources, indicating that integrating more creative elements can help the **LLM better learn and understand literature**.
>
> [1] Wenzek, Guillaume, et al. "CCNet: Extracting high quality monolingual datasets from web crawl data." LREC 2020.
>
> [2] Muennighoff, Niklas, et al. "Scaling data-constrained language models." Neurips 2023.
>
> [3] Marion, Max, et al. "When less is more: Investigating data pruning for pretraining llms at scale." Neurips 2023.

---

> ### Author Response · Authors · 2024-11-30
> **Author Response to Reviewer 8xFT (Part II)**
>
> **Questions1. How to resolve situations where DataMan makes unreasonable decisions on criteria?**
>
> 1. **DataMan rarely makes unreasonable decisions on criteria, making it unlikely to cause a 'snowball effect.**
>
>     First, we present Table 1. **the detailed analysis of testing accuracy of the “Overall Score”** quality criterion in DataMan.
>
>     In addition to the 5-level classification accuracy (5-level Acc) reported in the paper, we classify samples with a score of 3 or above as positive samples, and vice versa as negative samples, thus obtaining **2-level classification accuracy: (<3, ≥3 Acc)**. We detail the **accuracy for positive samples (≥3 Acc)** and **negative samples (<3 Acc)**, as well as **error rates for specific misclassification cases**: **extreme false negative samples** (<2 but ≥3 Error Rate), **moderate false negative samples** (≥2, <3 but >3 Error Rate), **marginal false negative samples** (≥2, <3 but =3 Error Rate), **extreme false positive samples** (≥4 but <3 Error Rate), and **marginal false positive samples** (≥3, <4 but <3 Error Rate).
>
>     Table 1. The **detailed analysis of testing accuracy** of the “Overall Score” quality criterion in DataMan.
>
>     |  | 5-level Acc｜ | <3, ≥3 Acc｜ |  ≥3 Acc｜ |  <3 Acc｜ | <2 but ≥3 Error｜ |  ≥2, <3 but >3 Error｜ | ≥2, <3 but =3 Error｜ | ≥4 but <3 Error｜ | ≥3, <4 but <3 Error |
>     | --- | --- | --- | --- | --- | --- | --- | --- | --- | --- |
>     | **Overall Score** | 81.3 | 97.5 | 98.5 | 81.6 | 0.2 | 2.7 | 15.5 | 0.2 | 1.4 |
>
>     As shown in Table 1, **the error rate** for DataMan in the **two extreme misclassification cases** of the "Overall Score" **is very low, at just 0.2%**. This indicates that **DataMan rarely mistakes poor-quality documents for high-quality ones**, and vice versa. Considering the **strong fault tolerance of pre-training, the snowball effect will not become a bottleneck.**
>
> 2. Great idea! Inspired by the reviewer, we **fine-tuned DataMan using the Qwen 1.5-MoE-A2.7B** model with the same SFT data and hyperparameters as the Qwen2-1.5B model. The tables below provide the **comparative analysis of the MoE and Dense models** in terms of **time and memory during the SFT stage**, **speed and memory during the inference stage**, and **recognition accuracy across quality criteria and domain types**.
>
>     Table 2. Comparison of **recognition accuracy on quality criteria** between MoE and Dense models.
>
>     | **Model Type** | Score Avg. | Score Var. | Accuracy | Coherence | Language Consistency | Semantic Density | Knowledge Novelty | Topic Focus | Creativity | Professionalism | Style Consistency | Grammatical Diversity | Structural Standardization | Originality | Sensitivity | Overall Score |
>     | --- | --- | --- | --- | --- | --- | --- | --- | --- | --- | --- | --- | --- | --- | --- | --- | --- |
>     | **Dense**  | 79.2 | 0.2 | 78.8 | 84.1 | 76.7 | 82.2 | 78.4 | 78.6 | 79.5 | 76.8 | 76.4 | 73.9 | 74.8 | 92.3 | 75.6 | 81.3 |
>     | **MoE**  | 81.0 | 0.2 | 80.8 | 85.7 | 79.2 | 83.2 | 80.6 | 79.7 | 81.6 | 78.4 | 78.6 | 75.3 | 76.9 | 93.2 | 78.3 | 82.7 |
>
>    Table 3. Comparison of **recognition accuracy on domain types** between MoE and Dense models.
>    | **Model Type** | Domain Avg. | Domain Var. | Medicine | Finance | Law | Education | Technology | Entertainment | Mathematics | Coding | Government | Culture | Transportation | Retail E-commerce | Telecommunication | Agriculture | Other |
>    | --- | --- | --- | --- | --- | --- | --- | --- | --- | --- | --- | --- | --- | --- | --- | --- | --- | --- |
>    | **Dense**  | 86.0 | 0.23 | 95.8 | 89.8 | 93.4 | 86.8 | 89.0 | 86.2 | 90.4 | 90.6 | 81.0 | 80.2 | 77.8 | 84.9 | 87.1 | 85.8 | 83.4 |
>    | **MoE** | 86.6 | 0.25 | 94.3 | 89.8 | 90.9 | 83.9 | 87.4 | 88.1 | 87.5 | 89.3 | 85.6 | 82.0 | 85.2 | 77.1 | 74.2 | 82.5 | 86.8 |
>
>    Table 4. Comparison of **time and memory during the SFT stage**, and **speed and memory during the inference stage**, between MoE and Dense models.
>    |  | SFT | SFT | Infer | Infer | Infer w/ vLLM  | Infer w/ vLLM |
>    | --- | --- | --- | --- | --- | --- | --- |
>    | **Model Type** | Time (h) | MeM (G) | Output Speed (Toks/S) | MeM (G) | Output Speed (Toks/S) | MeM (G) |
>    | **Dense** | 9.6 | 313G | 75.1 | 7.2 | 868 | 72.9 |
>    | **MoE** | 15.6 | 1988G | 6.2 | 34.1 | 1429 | 76.3 |
>
> Our conclusions are as follows:
> 1. The MoE model slightly outperforms the Dense Model in quality and domain recognition. The gap may widen with the release of Qwen-2 MoE.
> 2. Initially, the MoE model is slower (12 times) and uses more memory (4.7 times) than the Dense Model. However, with vLLM optimization, their memory usage becomes comparable, and the MoE model's speed even improves to 1.6 times faster than the Dense Model.

---

> ### Author Response · Authors · 2024-11-30
> **Author Response to Reviewer 8xFT (Part III)**
>
> Our conclusions are as follows:
>
> 3. A minor drawback is that the MoE model consumes more time and memory during the SFT phase, but this does not affect its usability.
> 4. Thus, we **report to the reviewer** that these **results demonstrate the effectiveness of the MoE method you suggested** (👍). In the future, **we will continue to update the DataMan’s base model** to better assess quality criteria.
>
> **Question2. Methods to reduce the high computational cost of DataMan annotation.**
>
> Good question! We introduce specific methods based on two ideas provided by the reviewer:
>
> 1. Using **lower-cost models** for annotation:
>
>     a. Firstly, as mentioned in Appendix B.2, the **Score-only DataMan outputs only** the overall score (**1 token**), thereby reducing computational costs.
>
>     b. Additionally, we have trained a **smaller Qwen2 0.5B model as DataMan** and will release these models together.
>
>     c. Also, **changing the DataMan training objective from text generation to text classification** can improve efficiency, but it may impact performance.
> 2. Using **heuristic preprocessing techniques** to reduce the amount of data annotation **in advance**:
>
>     a. Before annotating the pre-training dataset, **preprocessing techniques** in related work can **be employed to help lower annotation costs**:
>
>     - Deduplication: Fuzzydedup[4], Semdedup[5];
>     - Rule-based heuristic selection: Mean word length and stop word fraction[6], C4 filter[7], Gopher rules[8];
>     - Model-based heuristic selection: binary grammar discriminators[9];
>
>     b. As suggested by reviewer **JcNL**, large **open-source data processing tools** such as Dolma[10] and Fineweb[11] also **can reduce annotation costs**;
>
>     c. Additionally, **the Slimpajama dataset** used in this paper **has already reduced the cost** of DataMan annotation **through heuristic rules** like NFC normalization, Filter Short Documents, and MinHash Deduplication, on the Redpajama-1T dataset.
>
> **Questions3. Further analysis on the misalignment between perplexity and in-context learning performance.**
>
> We calculated the perplexity (PPL) and In-context learning (ICL) performance of all models across ten downstream tasks, then plotted **the Pearson’s and Spearman’s rank correlation coefficients to explore the misalignment** between PPL and ICL.
>
> Table 5. **Pearson’s and Spearman’s rank correlation** coefficients **between PPL and ICL** of all models across ten downstream tasks
> |  | ARC-e | ARC-c | SciQ | **LogiQA** | BoolQ | HellaSwag | PIQA | WinoGrande | NQ | **MMLU** |
> | --- | --- | --- | --- | --- | --- | --- | --- | --- | --- | --- |
> | **Pearson** | -0.65 | -0.52 | -0.85 | -0.14 | -0.46 | -0.78 | -0.69 | -0.64 | -0.67 | -0.05 |
> | **Spearman** | -0.29 | -0.27 | -0.60 | -0.18 | -0.03 | -0.65 | -0.59 | -0.68 | -0.58 | -0.08 |
>
> 1. From Table 5, it is evident that the **most pronounced misalignmen**t between perplexity and ICL performance occurs in the tasks: **LogiQA, and MMLU**.
> 2. To gain deeper insight into how this misalignment affects downstream tasks, we analyzed these two specific tasks and found the following reasons:
>
>     **Domain Mismatch:** Pre-training often uses broad general corpora, which reduces perplexity on common texts. However, for tasks like MMLU that cover 57 diverse specific domains (such as abstract algebra and anatomy), the issue of domain mismatch becomes more pronounced, leading to decling model performance on these tasks.
>
>     **Complexity of ICL Tasks:** Many ICL tasks demand complex or multi-step reasoning, which perplexity fails to effectively capture. As seen in LogiQA, which uses expert-written questions from Civil Servants' Exams to test human logical reasoning.
>
> **Question4.  Typo “vasrtdatasets”.**
>
> We have corrected it to "Vast dataset", and all typos have been corrected and highlighted in the revised PDF.
>
> [4] Tao Jiang, et al. Fuzzydedup: Secure fuzzy deduplication for cloud storage. IEEE Transactions on Dependable and Secure Computing, 20(3):2466–2483, 2022.
>
> [5] Amro Abbas, et al. Semdedup: Dataefficient learning at web-scale through semantic deduplication. arXiv 2023.
>
> [6] Hugo Laurençon,et al. The bigscience roots corpus: A 1.6 tb composite multilingual dataset. Neurips 2022.
>
> [7] Colin Raffel, et al. Exploring the limits of transfer learning with a unified text-to-text transformer. Journal of machine learning research, 21(140):1–67, 2020.
>
> [8] Rae, Jack W., et al. "Scaling language models: Methods, analysis & insights from training gopher." arXiv 2021.
>
> [9] Aakanksha Chowdhery, et al. Palm: Scaling language modeling with pathways. Journal of Machine Learning Research, 24(240):1–113, 2023.
>
> [10] Soldaini, Luca, et al. "Dolma: An open corpus of three trillion tokens for language model pretraining research." arXiv 2024.
>
> [11] Penedo, Guilherme, et al. "The fineweb datasets: Decanting the web for the finest text data at scale." arXiv 2024.

---

> > ### Comment · Reviewer_8xFT · 2024-12-01
> > **Response to Rebuttal**
> >
> > Thanks the authors for the detailed response. I think my concerns and questions are well-addressed. With the revisions and new experiments added in the paper, it is a solid paper to me. I have updated my scores.

---

> ### Author Response · Authors · 2024-12-01
> **Thanks for your postive feedback**
>
> Thank you for acknowledging our work. We are pleased that our response has addressed your concerns. In the final version of our paper, we will polish it further to incorporate the valuable insights gained from the rebuttal discussion. Thank you once again!
>
> Yours truly,
>
> Authors

---

### Author Response · Authors · 2024-11-30
**Sincerely Looking Forward to the Reviewers' Feedback**

Dear Reviewers,

We sincerely appreciate your careful evaluation and thoughtful feedback. We hold the utmost respect for your hard work.

We truly apologize for the delayed response due to the extensive workload in preparing feedback and conducting additional experiments.

As the review discussion stage is drawing to a close, we humbly seek to ascertain whether our responses adequately address your concerns. We are deeply grateful for your attention to this matter.

We summarize our responses to each reviewer as below:

1. We explained the rationale and process for selecting the 14quality critria of DataMan, the calculation method of Overall Score, and how these critria differentiate model capabilities. (To Reviewers **8xFT**, **Tq6k**, **Pb2j**)
2. We supplemented detailed testing accuracy and mixture-of-experts experiments to resolve the unreasonable judgment of critria by DataMan. (To Reviewer **8xFT**)
3. We elaborated on two methods to reduce annotation computation costs in DataMan: using lower-cost models and heuristic preprocessing techniques in advance. (To Reviewer **8xFT**)
4. We provided further analysis on the misalignment between perplexity and in-context learning preformance. (To Reviewer **8xFT**)
5. We have corrected all typos in the revised PDF. (To Reviewers **8xFT**, **Pb2j**)
6. We included experiments validating the effectiveness of DataMan on larger datasets (60B tokens) and raw, unfiltered datasets. (To Reviewer **Tq6k**)
7. We added DataMan's inference FLOPs and memory usage. (To Reviewer **Tq6k**)
8. We highlight the innovations and contributions of DataMan compared to Qurating. (To Reviewer **ViAr**)
9. We clarified misconceptions between pointwise rating and pairwise rating, the prompt validation process, and the role of bounding formulas. (To Reviewer **ViAr**)
10. We explained why choose Qwen2-1.5B instead of Sheared-LLama-1.3B to train DataMan and the details for fairly reproducing the "Sample with Qurating" baseline. (To Reviewer **ViAr**)
11. We clarified that DataMan does not directly rely on perplexity and has captured the essential aspects of data quality. (To Reviewer **JcNL**)
12. We added a comparison with the large-scale open-source data processing method - FineWeb, and clarified that DataMan is compatible and complementary to existing related work. (To Reviewer **JcNL**)
13. We clarified why DataMan's 14 criteria do not derive from intuition and manual design. (To Reviewer **Pb2j**)
14. We described the value of k in top-k sampling, and how the 14 quality criteria correspond to 3 principle. (To Reviewer **Pb2j**)

If you have any additional questions or require further clarification, please do not hesitate to inform us. We value the opportunity to continue the discussion if necessary.

Best regards,

The authors

---

### Author Response · Authors · 2024-12-02
**Sincerely Looking Forward to the Reviewers' Feedback Once Again**

Dear reviewers,

We genuinely appreciate the depth of your insightful comments and the value of your feedback. Your dedicated efforts are truly esteemed.

We have addressed each reviewer's concerns point by point and incorporated the responses into the latest version of the PDF.

**Due to the deadline is approaching, we sincerely hope you can take a look at it**🙏🙏. We faithfully hope our response adequately address your concern.

Best regards,

The authors

---

### Meta-Review · Area_Chair_gqWs · 2024-12-21

**Metareview:**

This paper introduces a method for improving the quality of pre-training data for language models. It leverages a fine-tuned LM to annotate data with 13 quality criteria, enabling the selection of high-quality subsets for training. The authors demonstrate that models trained on data filtered by their exhibit improved performance on various downstream tasks.

The reviewers have argued that the method has limited novelty and unfair comparisons to prior work. They also argue that the paper does not provide a strong justification for the choice of pointwise ratings over pairwise ratings. Also, the experiments are primarily conducted on a relatively small subset of the SlimPajama dataset, and further evaluation on larger datasets may provide a clearer signal of the usefulness of the method (which the authors provided in the rebuttal).

While the novelty of the approach is somewhat limited, the empirical results presented in the paper and provided in the rebuttal demonstrate the effectiveness of this method in improving model performance. Thus, I recommend accept (poster). The authors have provided a lot of additional results in the rebuttal, and I will encourage them to include them in the final draft.

**Additional Comments On Reviewer Discussion:**

The reviewers raised several points, including the limited novelty of the approach, the insufficient comparison to prior work, the unclear justification for pointwise ratings, and the limited experimental scope. The authors addressed these points by conducting additional experiments, strengthening the comparison to prior work, and providing a more detailed justification for their approach. I believe the rebuttal has addressed all major concerns raised by the reviewers.

---

### Decision · Program_Chairs · 2025-01-22

Accept (Poster)